# Hypergraph Neural Networks through the Lens of Message Passing: A Common Perspective to Homophily and Architecture Design

## Abstract

Most of the current hypergraph learning methodologies and benchmarking datasets in the hypergraph realm are obtained by *lifting* procedures from their graph analogs, simultaneously leading to overshadowing hypergraph network foundations. This paper attempts to confront some pending questions in that regard: **Q1** Can the concept of homophily play a crucial role in Hypergraph Neural Networks (HGNNs), similar to its significance in graph-based research? **Q2** Is there room for improving current hypergraph architectures and methodologies? (e.g. by carefully addressing the specific characteristics of higher-order networks) **Q3** Do existing datasets provide a meaningful benchmark for HGNNs? Diving into the details, this paper proposes a novel conceptualization of homophily in higher-order networks based on a message passing scheme; this approach harmonizes the analytical frameworks of datasets and architectures, offering a unified perspective for exploring and interpreting complex, higher-order network structures and dynamics. Further, we propose MultiSet, a novel message passing framework that redefines HGNNs by allowing hyperedge-dependent node representations, as well as introduce a novel architecture –MultiSetMixer– that leverages a new hyperedge sampling strategy. Finally, we provide an extensive set of experiments that contextualize our proposals and lead to valuable insights in hypergraph representation learning.

## 1    Introduction

Hypergraph learning techniques have rapidly grown in recent years, demonstrating their effectiveness in processing higher-order interactions in numerous fields, spanning from recommender systems (Yu et al., 2021; Zheng et al., 2018; La Gatta et al., 2022), to bioinformatics (Zhang et al., 2018; Yadati et al., 2020; Klamt et al., 2009) and computer vision (Li et al., 2022; Xu et al., 2022; Gao et al., 2012; Yin et al., 2017; Kim et al., 2011). However, so far, the development of HyperGraph Neural Networks (HGNNs) has been largely influenced by the well-established Graph Neural Network (GNN) field. In fact, most of the current methodologies and benchmarking datasets in the hypergraph realm are obtained by *lifting* procedures from their graph counterparts.

The advancement of hypergraph research has been significantly propelled by drawing inspiration from graph-based models (Feng et al., 2019; Yadati et al., 2019; Chien et al., 2022), but it has simultaneously led to overshadowing hypergraph network foundations. We argue that it is now the time to address fundamental questions in order to pave the way for further innovative ideas in the field. In that regard, this study explores some of these open questions to understand better current HGNN architectures and benchmarking datasets. **Q1** Can the concept of homophily play a crucial role in HGNNs, similar to its significance in graph-based research? **Q2** Given that current HGNNs are predominantly extensions of GNN architectures adapted to the hypergraph domain, are these extended methodologies suitable, or should we explore new strategies tailored specifically for handling hypergraph-based data? **Q3** Are the existing hypergraph benchmarking datasets truly *meaningful* and representative enough to draw robust and valid conclusions?

To begin with, we explore how the concept of homophily can be characterized in complex, higher-order networks. Notably, there are many ways of characterizing homophily in hypergraphs –such as the distribution of node features, the analogous distribution of the labels, or the group connectivity

similarity (as already discussed in (Veldt et al., 2023)). In particular, this work places the *node class distribution* at the core of the analysis, and introduces a novel definition of homophily that relies on a Message Passing (MP) scheme. Interestingly, this enables us to analyze both hypergraph datasets and architecture designs from the same perspective. In fact, we reckon that this unified message passing framework has the potential to inspire the development of meaningful contributions for processing higher-order relationships more effectively, as well as to successfully describe HGNN model performances (see Section 3 and Appendix A).

Next, we study state-of-the-art HGNN architectures and introduce a new framework called MultiSet. We demonstrate that MultiSet generalizes most existing frameworks for HGNNs, including AllSet (Chien et al., 2022) and UniGCNII (Huang & Yang, 2021). Our framework presents an innovative approach to message passing, where multiple hyperedge-dependent representations of nodes are enabled. Then, we introduce novel methodologies to process hypergraphs –including MultiSetMixer, a new HGNN architecture based on a particular implementation of a MultiSet layer. In these implementations, we introduce a novel connectivity-based mini-batching strategy capable of processing large hyperedges and discuss the intriguing property of natural connectivity-based distribution shifts.

Last, but not least, we provide an extensive set of experiments that, driven by the general questions stated above, aim to gain a better understanding on fundamental aspects of hypergraph representation learning. In fact, the obtained results not only help us contextualize the proposals introduced in this work, but indeed offer valuable insights that might help improve future hypergraph approaches.

**Summary of contributions**:

- We introduce a novel definition of the MP homophily for hypergraphs capable of effectively describing HGNN model performances (**Q1** and **Q3**).

- We present the novel MultiSet framework, which generalizes previous AllSet (Chien et al., 2022) and UniGCNII (Huang & Yang, 2021) formulations and allows for hyperedge-dependent node representations (**Q2**).

- We implement a novel MultiSetMixer model, a straightforward implementation of MultiSet framework that incorporates a novel hyperedge processing methodology based on hyperedge mini-batching sampling. Our proposed strategy addresses some scalability issues of current hypergraph models, and reveals a natural connectivity-based distribution shift with relevant implications in our experimental results (**Q2**).

- We perform a large set of experiments assessing the meaningfulness of benchmarking datasets, studying different MP propagation schemes and finally connecting homophily with models' performance (**Q1**, **Q2**, **Q3**).

## 2 RELATED WORKS

**Homophily in hypergraphs.** Homophily measures are typically defined for graph models and consider only pairwise relationships. In the context of Graph Neural Networks (GNNs), many of the current models implicitly use the homophily assumption, which is shown to be crucial for achieving a robust performance with relational data (Zhou et al., 2020; Chien et al., 2020; Halcrow et al., 2020). Nevertheless, despite the pivotal role that homophily plays in graph representation learning, its hypergraph counterpart mainly remains unexplored. In fact, to the best of our knowledge, Veldt et al. (2023) is the only work that faces the challenge of defining homophily in higher-order networks. Veldt et al. (2023) introduces a framework in which hypergraphs are used to quantify homophily from group interactions; however, the definition of homophily is restricted to uniform hypergraphs –i.e. where all hyperedges have exactly the same size (more details in Section 3). This represents a hard assumption that complicates its applicability to most of the current hypergraph datasets.

**Hypergraph Neural Networks.** The work of Chien et al. (2022) introduced AllSet, a general framework to describe HGNNs through a two-step message passing based mechanism, and demonstrated that most of the current hypergraph models are special instances of their formulation, based on the composition of two learnable permutation invariant functions that transmit information from nodes to hyperedges, and back from hyperedges to nodes. In particular, AllSet can be seen as a generalization of the most commonly used HGNNs, including all clique expansion based (CE) methods, HGNN (Feng et al., 2019), HNHN (Dong et al., 2020), HCHA (Bai et al., 2021), HyperSAGE (Arya et al.,

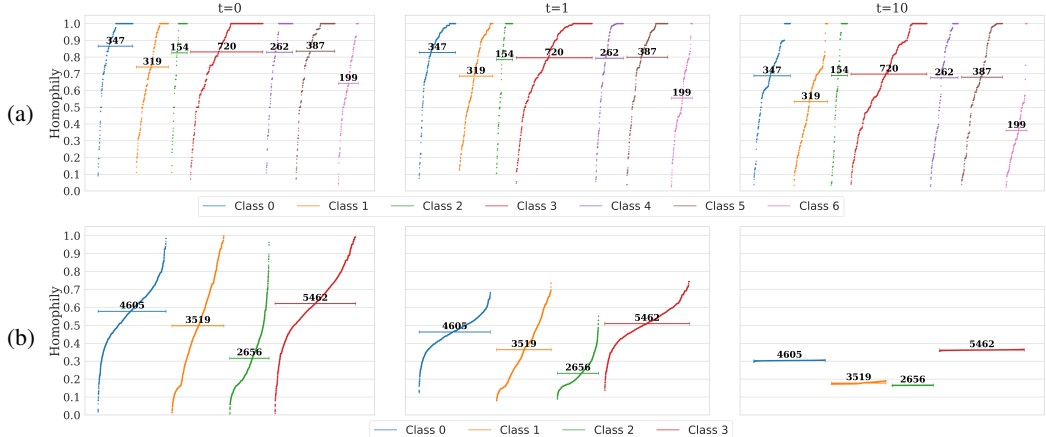

Figure 1: Node Homophily Distribution Scores for CORA-CA (a) and 20Newsgroups (b) using Equation 2 at $t = 0, 1$, and $10$ (left, middle, and right plots correspondingly). Horizontal lines depict class mean homophily, with numbers above indicating the number of visualized points per class.

2020) and HyperGCN(Yadati et al., 2019). Chien et al. (2022) also proposes two novel AllSet-like learnable layers: the first one –AllDeepSet– exploits Deep Set (Zaheer et al., 2017), and the second one –AllSetTransformer– Set Transformer (Lee et al., 2019), both of them achieving state-of-the-art results in the most common hypergraph benchmarking datasets. Concurrent to AllSet, the work of Huang & Yang (2021) also aimed at designing a common framework for graph and hypergraph NNs, and its more advanced UniGCNII method leverages initial residual connections and identity mappings in the hyperedge–to-node propagation to address over-smoothing issues; notably, UniGCNII do not fall under AllSet notation due to these residual connections. With Chien et al. (2022) and Huang & Yang (2021) being the most relevant ones to our work, we extend this review in Appendix D.

---

**Notation.** A hypergraph is an ordered pair of sets $\mathcal{G} = (\mathcal{V}, \mathcal{E})$, where $\mathcal{V}$ is the set of nodes and $\mathcal{E}$ is the set of hyperedges. Each hyperedge $e \in \mathcal{E}$ is a subset of $\mathcal{V}$, i.e., $e \subseteq \mathcal{V}$. A hypergraph is a generalization of the concept of a graph where (hyper)edges can connect more than two nodes. A vertex $v$ and a hyperedge $e$ are said to be incident if $v \in e$. For each node $v$, we denote its class by $y_v$, and by $\mathcal{E}_v = \{e \in \mathcal{E} : v \in e\}$ the subset of hyperedges in which it is contained, with $d_v = |\mathcal{E}_v|$ depicting the node degree. The set of classes of the hypergraph is represented by $\mathcal{C} = \{c_i\}_{i=1}^{|\mathcal{C}|}$.

---

## 3 DEFINING AND MEASURING HOMOPHILY IN HYPERGRAPHS

As previously stated in Section 2, the only rigorous work that faces the challenge of defining homophily in hypergraph networks is Veldt et al. (2023); however, it is restricted to $k$–uniform hypergraphs, which hugely limits its applicability to real-world higher-order datasets (a detailed description can be found in Appendix L). In this Section, we present a novel propagation-based homophily measure which is applicable for general, non-uniform hypergraphs. In essence, the score proposed in Veldt et al. (2023) tends to primarily assess the composition of hyperedges within the graph by quantifying the distribution of classes among hyperedges. In contrast, our definition places a greater emphasis on capturing the interconnections between different hyperedges by the exchange of information between nodes following the message passing scheme. Our introduced formulation, as well as the related findings described below, play a pivotal role on our attempt to answer the fundamental question **Q1** raised in the Introduction.

**Message Passing Homophily**    We present a novel two-step message passing homophily measure that, unlike the one proposed by Veldt et al. (2023), does not assume a $k$-uniform hypergraph structure. Furthermore, the proposed measure enables the definition of a score for each node and hyperedge for any neighborhood resolution, i.e., the connectivity of the hypergraph can be explicitly investigated. Our homophily definition follows the two-step message passing mechanism starting from the hyperedges of the hypergraph. Thus, given an edge $e$, we define the $0$-level hyperedge

homophily $h_e^0(c)$ as the fraction of nodes within each hyperedge that belong to class $c$, i.e.

$$h_e^0(c) = \frac{1}{|e|} \sum_{v \in e} \mathbb{1}_{y_v = c}. \tag{1}$$

This score describes how homophilic the initial connectivity is with respect to class $c$. By computing the score for every class $c_i \in \mathcal{C}$ we obtain a categorical distribution for each hyperedge $e \in \mathcal{E}$, i.e. $h_e^0 = (h_e^0(c_0), \ldots, h_e^0(c_{|C|}))$. We can then use this 0-level homophily information as a starting point to calculate higher-level homophily measurements for both nodes and hyperedges through the two-step message passing approach. Formally, we define the $t$-level homophily score as

$$h_v^{t-1} = \text{AGG}_{\mathcal{E}}\big(\{h_e^{t-1}(y_v)\}_{e \in \mathcal{E}_v}\big), \quad (2) \qquad\qquad h_e^t(c) = \text{AGG}_{\mathcal{V}}\big(\{h_v^{t-1}\}_{v \in e, y_v = c}\big), \qquad (3)$$

where $\text{AGG}_{\mathcal{E}}$ and $\text{AGG}_{\mathcal{V}}$ are functions that aggregate edge and node homophily scores, respectively. In our implementation, we considered the mean operation for both aggregations.

**Qualitative Analysis**   In this paragraph, we are taking a closer look at the qualitative analysis of the node homophily measure we introduced. One of the most straightforward ways to make use of the message passing homophily measure is to visualize how the node homophily score, as described in Eq. 2, changes dynamically. We've depicted this process in Figure 1, focusing on the CORA-CA and 20NewsGroup datasets. Note that in the figure, we are only showing non-isolated nodes. Looking at Figure 1 (a), we can observe several notable trends. First, in the initial node distribution ($t = 0$), every class, except class 6, has a significant number of fully homophilic nodes. As we move to the 1-hop neighborhood ($t = 1$), the corresponding classes either exhibit a moderate decrease in homophily or show no decrease at all. It's worth noting that at $t = 0, 1$, and 10, class 2 maintains a stable homophily distribution, hinting at an isolated subnetwork within. Furthermore, at $t = 10$, some points still maintain a node homophily score of 1, indicating the presence of multiple small subnetworks. Class 6 consistently displays the lowest average homophily measure at every step, with an average score of approximately 38% at $t = 10$. The node homophily distribution for the 20Newsgroups dataset is visualized in Figure 1 (b). At time step $t = 0$, we observe a wide range of homophily scores from 0 to 1 for each class. This suggests that the network is highly irregular with respect to connectivity. Moving to time step $t = 1$, there is a significant decrease in the homophily scores for every class, indicating a high degree of heterophily within the 1-hop neighborhood, which is not surprising considering step zero node homophily distribution. Finally, at time step $t = 10$, we can observe that all the classes converge to approximately the same homophily values within each class. This convergence suggests that the network is highly interconnected. More insights regarding node homophily measure and related HGNNs performances are described in Section 5 while the rest of the plots for the datasets can be found in Appendix L.

## 4 METHODS

Current HGNNs aim to generalize GNN concepts to the hypergraph domain, and are specially focused on redefining graph-based propagation rules to accommodate higher-order structures. In this regard, the work of Chien et al. (2022) introduced a general notation framework, called AllSet, that encompasses most of the currently available HGNN layers, including CEGCN/CEGAT, HGNN (Feng et al., 2019), HNHN (Dong et al., 2020), HCHA (Bai et al., 2021), HyperGCN (Yadati et al., 2019), and the AllDeepSet and AllSetTransformer presented in the same work (Chien et al., 2022).

The first part of this Section revisits the original AllSet formulation. Then, we introduce a new framework –termed MultiSet– which extends AllSet by allowing multiple hyperedge-dependent representations of nodes. Finally, we present some novel methodologies to process hypergraphs –including MultiSetMixer, a new HGNN architecture within the MultiSet framework. In contrast to previous formulations and models, our proposed framework and implementations are inspired by hypergraph needs and features, and motivated by the raised fundamental question **Q2**.

### 4.1 ALLSET PROPAGATION SETTING

For a given node $v \in \mathcal{V}$ and hyperedge $e \in \mathcal{E}$ in a hypergraph $\mathcal{G} = (\mathcal{V}, \mathcal{E})$, let $\boldsymbol{x}_v^{(t)} \in \mathbb{R}^f$ and $\boldsymbol{z}_e^{(t)} \in \mathbb{R}^d$ denote their vector representations at propagation step $t$. We say that a function $f$ is a multiset function if it is permutation invariant w.r.t. each of its arguments in turn. Typically, $\boldsymbol{x}_v^{(0)}$ and $\boldsymbol{z}_e^{(0)}$ are initialized based on the corresponding node and hyperedge original features, if available. The vectors $\boldsymbol{x}_v^{(0)}$ and $\boldsymbol{z}_e^{(0)}$ represent the initial node and hyperedge features, respectively. In this context, the AllSet framework (Chien et al., 2022) consists in the following two-step update rule:

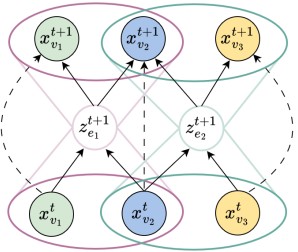

Figure 2: AllSet layout

$$\boldsymbol{z}_e^{(t+1)} = f_{\mathcal{V} \to \mathcal{E}}(\{\boldsymbol{x}_u^{(t)}\}_{u:u \in e}; \boldsymbol{z}_e^{(t)}), \tag{4}$$

$$\boldsymbol{x}_v^{(t+1)} = f_{\mathcal{E} \to \mathcal{V}}(\{\boldsymbol{z}_e^{(t+1)}\}_{e \in \mathcal{E}_v}; \boldsymbol{x}_v^{(t)}), \tag{5}$$

where $f_{\mathcal{V} \to \mathcal{E}}$ and $f_{\mathcal{E} \to \mathcal{V}}$ are two permutation invariant functions with respect to their first input. Equations 4 and 5 describe the propagation from nodes to hyperedges and vice versa, respectively. We extend the original AllSet formulation to accommodate UniGCNII (Huang & Yang, 2021), by modifying the node update rule (Eq. 5) in order to allow residual connections, i.e.:

$$\boldsymbol{x}_v^{(t+1)} = f_{\mathcal{E} \to \mathcal{V}}(\{\boldsymbol{z}_e^{(t+1)}\}_{e \in \mathcal{E}_v}; \{\boldsymbol{x}_v^{(k)}\}_{k=0}^t). \tag{6}$$

There is no requirement for the function to be permutation invariant with respect to this second set.

**Proposition 1.** *UniGCNII Huang & Yang, 2021 is a special case of AllSet considering 4 and 6.*

In the practical implementation of a model, $f_{\mathcal{V} \to \mathcal{E}}$ and $f_{\mathcal{E} \to \mathcal{V}}$ are parametrized and learnt for each dataset and task, and particular choices of these functions give rise to the different HGNN layer architectures considered in this paper; more details in Appendix E.

### 4.2 MULTISET FRAMEWORK

In this Section, we introduce our proposed MultiSet framework, which can be seen as an extension of AllSet where nodes can have multiple co-existing hyperedge–based representations. For a given hyperedge $e \in \mathcal{E}$ in a hypergraph $\mathcal{G} = (\mathcal{V}, \mathcal{E})$, we denote by $\boldsymbol{z}_e^{(t)} \in \mathbb{R}^d$ its vector representation at step $t$. However, for a node $v \in \mathcal{V}$, MultiSet allows for as many representations of the node as the number of hyperedges it belongs to. We denote by $\boldsymbol{x}_{v,e}^{(t)} \in \mathbb{R}^f$ the vector representation of node $v$ in a hyperedge $e \in \mathcal{E}_v$ at propagation time $t$, and by $\mathbb{X}_v^{(t)} = \{\boldsymbol{x}_{v,e}^{(t)}\}_{e \in \mathcal{E}_v}$ the set of all $d_v$ hidden states of that node in the specified time-step. Accordingly, the hyperedge and node update rules of MultiSet are formulated to accommodate hyperedge–dependent node representations:

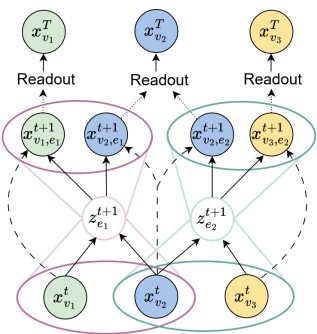

Figure 3: MultiSet layout

$$\boldsymbol{z}_e^{(t+1)} = f_{\mathcal{V} \to \mathcal{E}}(\{\mathbb{X}_u^{(t)}\}_{u:u \in e}; \boldsymbol{z}_e^{(t)}), \tag{7}$$

$$\boldsymbol{x}_{v,e}^{(t+1)} = f_{\mathcal{E} \to \mathcal{V}}(\{\boldsymbol{z}_e^{(t+1)}\}_{e \in \mathcal{E}_v}; \{\mathbb{X}_v^{(k)}\}_{k=0}^t), \tag{8}$$

where $f_{\mathcal{V} \to \mathcal{E}}$ and $f_{\mathcal{E} \to \mathcal{V}}$ are two multiset functions with respect to their first input.

After $T$ iterations of message passing, MultiSet also considers a last readout-based step with the idea of obtaining a unique final representation $x_v^T \in \mathbb{R}^{f'}$ for each node from the set of its hyperedge–based representations:

$$\boldsymbol{x}_v^{(T)} = f_{\mathcal{V} \to \mathcal{V}}(\{\mathbb{X}_v^{(k)}\}_{k=0}^T) \tag{9}$$

where $f_{\mathcal{V} \to \mathcal{V}}$ is also a multiset function.

**Proposition 2.** *AllSet 4-5, as well as its extension 4-6, are special cases of MultiSet 7-8-9.*

### 4.3 TRAINING MULTISET NETWORKS

This Section describes the main characteristics of our MultiSet layer implementation, termed Multi-SetMixer, and presents a novel sampling procedure that our model incorporates.

**Learning MultiSet Layers** Following the mixer-style block designs (Tolstikhin et al., 2021) and standard practice, we propose the following MultiSet layer implementation for HGNNs:

$$z_e^{(t+1)} = f_{\mathcal{V} \to \mathcal{E}}(\{x_{u,e}^{(t)}\}_{u:u \in e}; z_e^{(t)}) := \frac{1}{|e|} \sum_{v \in e} x_{u,e}^{(t)} + \text{MLP}\left(\text{LN}\left(\frac{1}{|e|} \sum_{v \in e} x_{u,e}^{(t)}\right)\right), \quad (10)$$

$$x_{v,e}^{(t+1)} = f_{\mathcal{E} \to \mathcal{V}}(z_e^{(t+1)}; x_{v,e}^{(t)}) := x_{v,e}^{(t)} + \text{MLP}\left(\text{LN}(x_{v,e}^{(t)})\right) + z_e^{(t+1)}, \quad (11)$$

$$x_v^{(T)} = f_{\mathcal{V} \to \mathcal{V}}(\mathbb{X}_v^{(T)}) := \frac{1}{d_v} \sum_{e \in \mathcal{E}_v} x_{v,e}^{(t)} \quad (12)$$

where MLPs are composed of two fully-connected layers, and LN stands for layer normalisation. This novel architecture, which we call MultiSetMixer, is based on a mixer-based pooling operation for *(i)* updating hyperedges from its node's representations, and *(ii)* generate and update hyperedge-dependent representations of the nodes.

**Proposition 3.** *The functions $f_{\mathcal{V} \to \mathcal{E}}$, $f_{\mathcal{E} \to \mathcal{V}}$ and $f_{\mathcal{V} \to \mathcal{V}}$ defined in MultiSetMixer are permutation invariant. Furthermore, these functions are universal approximators of multiset functions when the size of the input multiset is finite.*

**Mini-batching** The motivation for introducing a new strategy to iterate over hypergraph datasets is twofold. On the one hand, current HGNN pipelines suffer from scalability issues to process large datasets and very large hyperedges. On the other, pooling operations over relatively large sets can also lead to over-squashing the signal. To help in these directions, we propose sampling $X$ mini-batches of a certain size $B$ at each iteration. At *step 1*, it samples $B$ hyperedges from $\mathcal{E}$. The hyperedge sampling over $\mathcal{E}$ can be either uniform or weighted (e.g. by taking into account hyperedge cardinalities). Then in *step 2* $L$ nodes are in turn sampled from each sampled hyperedge $e$, padding the hyperedge with $L - |e|$ special padding tokens if $|e| < L$ –consisting of $\mathbf{0}$ vectors that can be easily discarded in some computations. Overall, the shape of the obtained mini-batch $X$ has fixed size $B \times L$. Please refer to Appendix K for additional analysis.

## 5 EXPERIMENTAL RESULTS

The questions that we introduced in the Introduction have shaped our research, leading to a new definition of higher-order homophily and novel architectural designs and sampling strategies that can potentially fit better the properties of hypergraph networks. In subsequent subsections, we set again three main questions that follow up from these fundamental inquiries and can help contextualize the technical contributions introduced in this paper.

**Dataset and Models** We use the same datasets used in Chien et al. (2022), which includes Cora, Citeseer, Pubmed, ModelNet40, NTU2012, 20Newsgroups, Mushroom, ZOO, CORA-CA, and DBLP-CA. More information about datasets and corresponding statistics can be found in Appendix I.2. We also utilize the benchmark implementation provided by Chien et al. (2022) to conduct the experiments with several models, including AllDeepSets, AllSetTransformer, UniGCNII, CEGAT, CEGCN, HCHA, HGNN, HNHN, HyperGCN, HAN, and HAN (mini-batching). Additionally, we consider vanilla MLP applied to node features and a transformer architecture and introduce three new models: MultiSetMixer, MLP Connectivity Batching (MLP CB), and Multiple MLP CB (MMLP CB). The MLP CB and MMLP CB models use connectivity information to form and process batches. Specifically, the MMLP CB model processes the top three most frequent connectivities using separate MLP encoders, while the fourth encoder is used to process the remaining connectivities. We refer to Section 4.3 for further details about all these architectures. All models are optimized using 15 splits with 2 model initializations, resulting in a total of 30 runs; see Appendix I.1 for further details.

### 5.1 HOW DOES MULTISETMIXER PERFORM?

Our first experiment directly targets our fundamental **Q2** by assessing the performance of our proposed MultiSetMixer model and the two introduced baselines, MLP CB and MMLP CB. Figure 4 shows the average rankings –across all models and datasets– of the top-3 best performing models for different training splits, exhibiting that those splits can impact the relative performance among

models. However, due to space limitations, we restrict our analysis to the 50% split results shown in Table 1,[1] and relegate to Appendix J.1 the corresponding tables for the other scenarios.

Table 1 emphasizes the MultiSetMixer model's relatively solid performance, being the best-performing model on the NTU2012, ModelNet40, and 20Newsgroups datasets. Its performance on the 20Newsgroups dataset is especially noteworthy, significantly outperforming the other models. Moreover, it is notable that MLP CB and MMLP CB exhibit similar behaviour on this dataset. In contrast, the performance of all other models achieves roughly the same performance as the MLP. This observation suggests that these models can not account for dataset connectivity; in particular, as we demonstrated in Section 3, the dispersion of the node homophily measure, with a subsequent convergence to a similar value within each class, indicates that the dataset's connectivity is notably non-

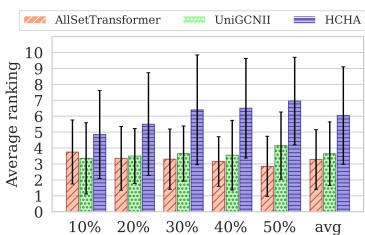

Figure 4: Average ranking and standard deviation for various training percentages.

homophilic and presents a challenge. In contrast, CORA-CA exhibits a high degree of homophily within its hyperedges and shows the most significant performance gap between the best-performing model, AllSetTransformer, and the basic MLP. A similar trend is observed for DBLP-CA (see node homophily plot in Appendix L). Please refer to Section 5.4 for additional experiments analyzing the impact of connectivity on the models. On the other hand, we can notice that CEGAT, CEGCN and our proposed model don't perform well on the Mushroom dataset. This is noteworthy because the Mushroom dataset's features are highly representative, as demonstrated by the near-perfect performance of the MLP classifier. This suggests that, in this particular case, connectivity may not play a crucial role in achieving high performance.

Table 1: Hypergraph model performance benchmarks. Test accuracy in % averaged over 15 splits.

| Model | Cora | Citeseer | Pubmed | CORA-CA | DBLP-CA | Mushroom | NTU2012 | ModelNet40 | 20Newsgroups | ZOO | avg. ranking |
|---|---|---|---|---|---|---|---|---|---|---|---|
| AllDeepSets | 77.11 ± 1.00 | 70.67 ± 1.42 | 89.04 ± 0.45 | 82.23 ± 1.46 | 91.34 ± 0.27 | 99.96 ± 0.05 | 86.49 ± 1.86 | 96.70 ± 0.25 | 81.19 ± 0.49 | 89.10 ± 7.00 | 6.00 |
| AllSetTransformer | 79.54 ± 1.02 | 72.52 ± 0.88 | 88.74 ± 0.51 | 84.43 ± 1.14 | 91.61 ± 0.19 | 99.95 ± 0.05 | 88.22 ± 1.42 | 98.00 ± 0.12 | 81.59 ± 0.59 | 91.03 ± 7.31 | 2.85 |
| UniGCNII | 78.46 ± 1.14 | 73.05 ± 1.48 | 88.07 ± 0.47 | 83.92 ± 1.02 | 91.56 ± 0.18 | 99.89 ± 0.07 | 88.24 ± 1.56 | 97.84 ± 0.16 | 81.16 ± 0.49 | 89.61 ± 8.09 | 4.15 |
| CEGAT | 76.53 ± 1.58 | 71.58 ± 1.11 | 87.11 ± 0.49 | 77.50 ± 1.51 | 88.74 ± 0.31 | 96.81 ± 1.41 | 82.27 ± 1.60 | 92.79 ± 0.44 | OOM | 44.62 ± 9.18 | 11.00 |
| CEGCN | 77.03 ± 1.31 | 70.87 ± 1.19 | 87.01 ± 0.62 | 77.55 ± 1.65 | 88.12 ± 0.25 | 94.91 ± 0.44 | 80.90 ± 1.74 | 90.04 ± 0.47 | OOM | 49.23 ± 6.81 | 11.67 |
| HCHA | 79.53 ± 1.33 | 72.57 ± 1.06 | 86.97 ± 0.55 | 83.53 ± 1.12 | 91.21 ± 0.28 | 98.94 ± 0.54 | 86.60 ± 1.96 | 94.50 ± 0.33 | 80.75 ± 0.53 | 89.23 ± 6.81 | 6.75 |
| HGNN | 79.53 ± 1.33 | 72.24 ± 1.08 | 86.97 ± 0.55 | 83.45 ± 1.22 | 91.26 ± 0.26 | 98.94 ± 0.54 | 86.71 ± 1.48 | 94.50 ± 0.33 | 80.75 ± 0.52 | 89.23 ± 6.81 | 6.85 |
| HNHN | 77.68 ± 1.08 | 73.47 ± 1.36 | 87.88 ± 0.47 | 78.53 ± 1.15 | 86.73 ± 0.40 | 99.97 ± 0.04 | 88.28 ± 1.50 | 88.28 ± 1.50 | 97.84 ± 0.15 | 81.53 ± 0.55 | 89.23 ± 7.85 | 5.05 |
| HyperGCN | 74.78 ± 1.11 | 66.06 ± 1.58 | 82.32 ± 0.62 | 77.48 ± 1.14 | 86.07 ± 3.32 | 69.51 ± 4.98 | 47.65 ± 5.01 | 46.10 ± 7.95 | 80.84 ± 0.49 | 51.54 ± 9.88 | 13.80 |
| HAN | 80.73 ± 1.37 | 73.69 ± 0.95 | 86.34 ± 0.61 | 84.19 ± 0.81 | 91.10 ± 0.20 | 91.33 ± 0.91 | 83.78 ± 1.75 | 93.85 ± 0.33 | 79.67 ± 0.55 | 80.26 ± 6.42 | 8.10 |
| HAN minibatch | 80.24 ± 2.17 | 73.55 ± 1.13 | 85.41 ± 2.32 | 82.04 ± 2.56 | 90.52 ± 0.50 | 93.87 ± 1.04 | 80.62 ± 2.00 | 92.06 ± 0.63 | 79.76 ± 0.56 | 70.39 ± 11.29 | 9.90 |
| MultiSetMixer | 79.38 ± 1.08 | 72.79 ± 1.12 | 85.71 ± 0.49 | 82.62 ± 1.20 | 89.87 ± 0.29 | 95.85 ± 3.21 | 88.73 ± 1.29 | 98.15 ± 0.19 | 87.83 ± 2.68 | 78.67 ± 9.08 | 6.40 |
| MLP CB | 74.06 ± 1.26 | 71.93 ± 1.53 | 85.83 ± 0.51 | 74.39 ± 1.40 | 84.91 ± 0.44 | 96.83 ± 2.18 | 85.43 ± 1.51 | 96.41 ± 0.32 | 86.13 ± 2.82 | 81.61 ± 10.98 | 9.80 |
| MMLP CB | 71.05 ± 2.03 | 69.26 ± 1.91 | 85.20 ± 0.54 | 71.16 ± 2.17 | 84.08 ± 0.42 | 95.71 ± 2.42 | NA | NA | 85.04 ± 4.04 | 83.89 ± 9.52 | 12.75 |
| MLP | 73.27 ± 1.09 | 72.07 ± 1.65 | 87.13 ± 0.49 | 73.27 ± 1.09 | 84.77 ± 0.41 | 99.91 ± 0.08 | 79.70 ± 1.56 | 95.31 ± 0.28 | 80.93 ± 0.59 | 85.13 ± 6.90 | 10.40 |
| Transformer | 74.15 ± 1.17 | 71.82 ± 1.51 | 87.37 ± 0.49 | 73.61 ± 1.55 | 85.26 ± 0.38 | 99.95 ± 0.08 | 82.88 ± 1.93 | 96.29 ± 0.29 | 81.17 ± 0.54 | 88.72 ± 10.25 | 9.05 |

## 5.2 CAN HOMOPHILY HELP US UNDERSTAND OUR EXPERIMENTAL RESULTS?

Our following step is to analyze whether the previously introduced message passing homophily measure (Section 3) can be useful in describing the observed results, which is totally aligned with our fundamental **Q1**. Due to space limitations, we leave to Appendix A the detailed study we perform on this relevant aspect, but highlight here the main finding: our homophily concept correlates better with HGNN models' performance (and specially our MultiSetMixer implementation) compared to classical homophily measures over the clique-expanded hypergraph. In doing so, we demonstrate the advantages of the dynamic nature of the proposed message passing score. These insights underscore the crucial role of correctly expressing homophily in hypernetworks, emphasizing the potential of our proposed homophily score in capturing higher-order dynamics.

## 5.3 WHAT IS THE IMPACT OF THE INTRODUCED MINI-BATCH SAMPLING STRATEGY?

Next, we examine the role of our proposed mini-batching sampling *(i)* in explaining Table 1 results and *(ii)* influencing other models' performance. These experiments provide valuable insights on **Q2**.

---

[1]Unless otherwise specified, all tables in the main body of the paper use a 50%/25%/25% split between training and testing. The results are shown as Mean Accuracy Standard Deviation, with the best result highlighted in bold and shaded in grey, and results within one standard deviation are displayed in blue-shaded boxes.

**Class distribution analysis**  To evaluate and motivate the potential of the proposed mini-batching sampling, we investigate the reason behind both the superior performance of MultiSetMixer, MLP CB and MMLP CB on 20NewsGroup and their poor performance on Mushroom. Framing mini-batching from the connectivity perspective presents a challenge that conceals significant potential for improvement (Teney et al., 2023). It is important to note that connectivity, by definition, describes relationships among the nodes, implying that some parts of the dataset might interconnect more densely, creating some sort of hubs within the network. Thus, mini-batching might introduce unexpected skew in training distribution. In particular, in Figure 5, we depict the class distribution of the original dataset, referred to as *Node*, while '*Step 1 and 2*' and '*Step 1*' shows the distribution after each step in our mini-batching procedure. The sampling procedure tends to rebalance class distributions in certain cases, such as the 20NewsGroup dataset, while in contrast, it introduces an imbalance that was not present in the original labels in the Mushroom dataset, where

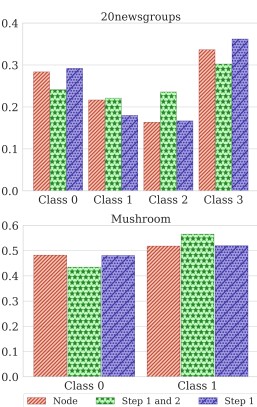

Figure 5: Distribution of classes

our model demonstrated suboptimal performance. This observation leads to the hypothesis that, in some cases, the sampling procedure produces a shift distribution that rebalances the class distributions and conducts our model to outperform the comparison models.

**Application to Other Models**  Furthermore, we explore the proposed mini-batch sampling procedure with the AllSetTransformer and UniGCNII models by implementing Step 1 of the mini-batch procedure without additional hyperparameter optimization. From Table 2, we can observe a drop in performance for most of the datasets both for AllSetTransformer and for UniGCNII; both models, on average, outperform the HAN (mini-batching) model. This suggests the substantial potential of the proposed sampling procedure. More in detail, AllsetTransformer has a substantial decrease in accuracy for the CORA-CA dataset, in contrast to the UniGCNII, which registers only marginal decreases. An analogous pattern emerges with the DBLP-CA dataset.

Table 2: Mini-batching experiment. Test accuracy in % averaged over 15 splits.

| Model | Cora | Citeseer | Pubmed | CORA-CA | DBLP-CA | Mushroom | NTU2012 | ModelNet40 | 20Newsgroups | ZOO | avg. ranking |
|---|---|---|---|---|---|---|---|---|---|---|---|
| AllSetTransformer (batched) | 74.34 ± 1.08 | 69.67 ± 1.46 | **87.75 ± 0.30** | 75.75 ± 1.46 | 86.06 ± 0.22 | 99.91 ± 0.05 | 87.55 ± 0.86 | 96.42 ± 0.17 | 81.37 ± 0.28 | **93.20 ± 5.38** | 2.70 |
| UniGCNII (batched) | 77.88 ± 0.69 | 69.51 ± 0.87 | 86.82 ± 0.33 | **83.12 ± 0.89** | 90.45 ± 0.28 | **99.95 ± 0.04** | 87.64 ± 0.99 | 97.55 ± 0.17 | 81.23 ± 0.31 | 90.00 ± 4.43 | 2.20 |
| HAN minibatch | **80.24 ± 2.17** | **73.55 ± 1.13** | 85.41 ± 2.32 | 82.04 ± 2.56 | **90.52 ± 0.50** | 93.87 ± 1.04 | 80.62 ± 2.00 | 92.06 ± 0.63 | 79.76 ± 0.56 | 70.39 ± 11.29 | 3.00 |
| MultiSetMixer | 79.38 ± 1.08 | 72.79 ± 1.12 | 85.71 ± 0.49 | 82.62 ± 1.20 | 89.87 ± 0.29 | 95.85 ± 3.21 | **88.73 ± 1.29** | **98.15 ± 0.19** | **87.83 ± 2.68** | 78.67 ± 9.08 | **2.10** |

## 5.4 How do connectivity changes affect performance?

We design two different experimental approaches, aiming to systematically modify the original connectivity of datasets. The first experiment tests the performance when some hyperedges are removed following different *drop connectivity* strategies. Then, a second experiment examines the model's performance by introducing two preprocessing strategies applied to the hypergraph connectivity. Our findings below shed some light on our fundamental questions **Q1**, **Q2** and **Q3**.

**Reducing Connectivity**  This experiment aims to investigate the significance of connectivity in datasets and the extent to which it influences the performance of the models. We divide this experiment into two parts: (i) drop connectivity and (ii) connectivity rewiring. In the first part of the experiment, we employ three strategies to introduce variations in the initial dataset's connectivity. The first two strategies involve ordering hyperedges based on their lengths in **ascending order**. In the first approach, referred to as *trimming*, we remove the initial $x\%$ of ordered hyperedges. The second approach, referred to as *retention*, involves keeping the first $x\%$ of hyperedges and discarding the remaining $100 - x\%$. Finally, the last strategy involves randomly dropping $x\%$ of hyperedges from the dataset, referred to as *random drop*. Results shown in Table 3 also indicate that connectivity minimally impacts CEGCN, and AllSetTransformer for the Citeseer and Pubmed datasets. On the other hand, MultiSetMixer performs better at the *trimming 25%* setting, although the achieved performance is on par with MLP reported in Table 1. This suggests that the proposed model was negatively affected by the distribution shift. Conversely, we observe a similar but opposite trend for the Mushroom dataset, where MultiSetMixer's performance improves due to the reduced impact of the distribution shift. Another interesting observation is that the CEGCN model gains improvement in 6 out of 9 datasets, with a doubled increase for the ZOO dataset. In the case of

Cora, CORA-CA, and DBLP-CA datasets, another interesting pattern emerges: retaining only 25% of the highest relationships (*retention 25%*) consistently results in better performance compared to retaining 50% or 75%. This is intriguing because, at the 25% level, we are preserving only a small fraction of the higher-order relationships. The opposite pattern holds for the *trimming* strategy. For the datasets mentioned above, this phenomenon remained consistent across all models. Notice that this phenomenon doesn't appear when we remove hyperedges randomly; in this case, as expected, the more hyperedges we remove, the more the performances decrease.

**Rewiring Connectivity**    In Appendix J.3, we show that the 'Label Based' strategy enhances the performance for all datasets and models, as seen in Table 15. Notably, the graph-based method CEGCN achieves similar results to HGNNs in this strategy. Additionally, on average, only CEGCN performs better with the 'k-means' strategy and mitigates distribution shifts for MultiSetMixer. These findings collectively suggest the crucial role of connectivity preprocessing, especially for graph-based models.

Table 3: Drop connectivity. Test accuracy in % averaged over 15 splits.

| Model | Type | Cora | Citeseer | Pubmed | CORA-CA | DBLP-CA | Mushroom | NTU2012 | ModelNet40 | 20Newsgroups | ZOO | avg. ranking |
|---|---|---|---|---|---|---|---|---|---|---|---|---|
| AllSetTransformer | Original | 79.54 ± 1.02 | 72.52 ± 0.88 | 88.74 ± 0.51 | 84.43 ± 1.14 | 91.61 ± 0.19 | 99.95 ± 0.05 | 88.22 ± 1.42 | 98.00 ± 0.12 | 81.59 ± 0.59 | 91.03 ± 7.31 | 1.95 |
| | Random 25% | 79.11 ± 0.99 | 72.75 ± 1.14 | 88.67 ± 0.47 | 82.36 ± 1.38 | 90.61 ± 0.29 | 99.94 ± 0.09 | 87.50 ± 1.36 | 97.98 ± 0.17 | 81.70 ± 0.52 | 89.87 ± 7.66 | 3.10 |
| | Random 50% | 77.77 ± 1.34 | 72.21 ± 1.25 | 88.50 ± 0.45 | 79.73 ± 1.58 | 89.46 ± 0.27 | 99.96 ± 0.04 | 87.34 ± 1.55 | 97.83 ± 0.17 | 81.55 ± 0.66 | 89.49 ± 6.30 | 5.85 |
| | Random 75% | 76.92 ± 1.20 | 72.40 ± 1.22 | 88.54 ± 0.47 | 77.88 ± 1.74 | 87.73 ± 0.32 | 99.76 ± 0.15 | 86.31 ± 1.34 | 97.52 ± 0.20 | 81.46 ± 0.62 | 87.69 ± 6.09 | 8.10 |
| | Retention 25% | 79.19 ± 1.11 | 72.49 ± 0.86 | 88.73 ± 0.40 | 83.58 ± 1.30 | 91.18 ± 0.17 | 99.93 ± 0.09 | 87.21 ± 1.58 | 97.82 ± 0.17 | 81.63 ± 0.48 | 86.92 ± 7.18 | 4.10 |
| | Retention 50% | 78.16 ± 0.98 | 72.55 ± 1.13 | 88.70 ± 0.37 | 82.90 ± 1.15 | 90.80 ± 0.22 | 99.89 ± 0.18 | 86.67 ± 1.64 | 97.36 ± 0.21 | 81.61 ± 0.49 | 88.08 ± 7.51 | 5.20 |
| | Retention 75% | 77.38 ± 1.35 | 72.43 ± 0.98 | 88.71 ± 0.39 | 81.07 ± 1.20 | 89.83 ± 0.25 | 99.97 ± 0.04 | 85.58 ± 1.70 | 97.27 ± 0.22 | 81.58 ± 0.48 | 88.97 ± 6.91 | 5.80 |
| | Trimming 25% | 75.83 ± 1.31 | 72.39 ± 1.50 | 88.40 ± 0.45 | 76.51 ± 1.35 | 86.38 ± 0.32 | 99.84 ± 0.13 | 86.88 ± 1.66 | 97.10 ± 0.24 | 81.55 ± 0.55 | 93.08 ± 7.79 | 8.05 |
| | Trimming 50% | 77.37 ± 1.17 | 72.32 ± 1.30 | 88.49 ± 0.40 | 77.41 ± 1.73 | 87.03 ± 0.27 | 99.91 ± 0.12 | 86.86 ± 1.53 | 97.86 ± 0.21 | 81.45 ± 0.50 | 89.74 ± 8.53 | 7.50 |
| | Trimming 75% | 78.15 ± 1.11 | 72.67 ± 1.00 | 88.48 ± 0.39 | 78.91 ± 1.54 | 88.55 ± 0.26 | 99.92 ± 0.09 | 87.68 ± 1.56 | 97.90 ± 0.23 | 81.41 ± 0.61 | 91.03 ± 7.17 | 5.35 |
| CEGCN | Original | 77.03 ± 1.31 | 70.87 ± 1.19 | 87.01 ± 0.62 | 77.55 ± 1.65 | 88.12 ± 0.25 | 94.91 ± 0.44 | 80.90 ± 1.74 | 90.04 ± 0.47 | OOM | 49.23 ± 6.81 | 4.61 |
| | Random 25% | 76.08 ± 1.55 | 71.35 ± 1.44 | 86.89 ± 0.59 | 76.51 ± 1.53 | 87.01 ± 0.39 | 93.11 ± 0.46 | 80.68 ± 1.86 | 90.36 ± 0.46 | OOM | 49.74 ± 6.22 | 6.22 |
| | Random 50% | 75.55 ± 1.63 | 71.42 ± 1.60 | 86.70 ± 0.48 | 75.27 ± 1.22 | 86.24 ± 0.35 | 93.28 ± 0.61 | 80.63 ± 1.78 | 90.69 ± 0.54 | OOM | 56.92 ± 7.24 | 6.33 |
| | Random 75% | 75.34 ± 1.62 | 71.73 ± 1.90 | 86.97 ± 0.51 | 74.53 ± 1.56 | 85.36 ± 0.26 | 93.01 ± 0.45 | 80.56 ± 1.76 | 91.91 ± 0.54 | OOM | 63.20 ± 5.59 | 6.33 |
| | Retention 25% | 76.12 ± 1.58 | 70.87 ± 1.42 | 86.94 ± 0.56 | 76.98 ± 1.53 | 87.90 ± 0.29 | 94.94 ± 0.48 | 79.20 ± 1.42 | 90.59 ± 0.59 | OOM | 49.87 ± 7.59 | 5.50 |
| | Retention 50% | 75.43 ± 1.28 | 70.83 ± 1.52 | 86.95 ± 0.54 | 76.87 ± 1.49 | 87.58 ± 0.28 | 94.97 ± 0.40 | 78.53 ± 1.90 | 90.09 ± 0.56 | OOM | 45.77 ± 6.88 | 6.89 |
| | Retention 75% | 75.53 ± 1.25 | 71.72 ± 1.42 | 87.11 ± 0.53 | 76.36 ± 1.42 | 87.03 ± 0.28 | 94.74 ± 0.39 | 79.82 ± 1.41 | 92.29 ± 0.46 | OOM | 40.38 ± 5.42 | 5.44 |
| | Trimming 25% | 75.58 ± 1.56 | 72.26 ± 1.52 | 87.36 ± 0.51 | 74.84 ± 1.31 | 84.97 ± 0.31 | 99.60 ± 0.11 | 83.10 ± 1.69 | 91.85 ± 0.42 | OOM | 87.69 ± 7.31 | 3.44 |
| | Trimming 50% | 76.57 ± 1.47 | 71.81 ± 1.44 | 87.07 ± 0.55 | 74.66 ± 1.68 | 85.24 ± 0.33 | 99.54 ± 0.18 | 80.72 ± 1.64 | 90.64 ± 0.54 | OOM | 71.28 ± 6.60 | 4.00 |
| | Trimming 75% | 76.53 ± 1.50 | 71.45 ± 1.45 | 86.75 ± 0.54 | 74.56 ± 1.32 | 85.56 ± 0.33 | 99.14 ± 0.23 | 80.38 ± 1.91 | 90.06 ± 0.37 | OOM | 58.46 ± 7.17 | 6.22 |
| MultiSetMixer | Original | 79.38 ± 1.08 | 72.79 ± 1.12 | 85.71 ± 0.49 | 82.62 ± 1.20 | 89.87 ± 0.29 | 95.85 ± 3.21 | 88.73 ± 1.29 | 98.15 ± 0.19 | 87.83 ± 2.68 | 78.67 ± 9.08 | 2.75 |
| | Random 25% | 78.63 ± 1.30 | 72.37 ± 1.50 | 85.71 ± 0.55 | 81.18 ± 1.16 | 89.11 ± 0.31 | 93.80 ± 4.69 | 87.92 ± 1.50 | 98.01 ± 0.19 | 76.65 ± 1.76 | 77.60 ± 9.00 | 4.75 |
| | Random 50% | 77.66 ± 1.18 | 72.24 ± 1.42 | 85.92 ± 0.45 | 78.51 ± 1.58 | 88.13 ± 0.34 | 94.36 ± 3.79 | 86.22 ± 2.01 | 97.92 ± 0.13 | 74.36 ± 1.23 | 75.53 ± 14.10 | 6.40 |
| | Random 75% | 76.59 ± 1.27 | 72.12 ± 1.43 | 86.10 ± 0.53 | 76.91 ± 1.43 | 86.42 ± 0.42 | 98.74 ± 0.90 | 85.31 ± 1.64 | 97.48 ± 0.21 | 76.53 ± 0.75 | 58.75 ± 17.97 | 7.25 |
| | Retention 25% | 78.99 ± 1.00 | 72.12 ± 1.28 | 85.73 ± 0.44 | 82.01 ± 1.56 | 89.61 ± 0.33 | 97.18 ± 2.01 | 86.96 ± 1.62 | 97.95 ± 0.19 | 88.17 ± 2.51 | 80.15 ± 8.87 | 3.85 |
| | Retention 50% | 77.88 ± 1.28 | 72.32 ± 1.36 | 85.89 ± 0.52 | 80.85 ± 1.14 | 89.24 ± 0.31 | 97.72 ± 1.42 | 84.56 ± 1.97 | 97.39 ± 0.24 | 85.04 ± 2.06 | 76.31 ± 12.45 | 5.20 |
| | Retention 75% | 77.44 ± 1.32 | 72.18 ± 1.32 | 85.93 ± 0.54 | 78.67 ± 1.32 | 87.86 ± 0.35 | 94.75 ± 3.86 | 83.94 ± 1.79 | 97.00 ± 0.26 | 84.65 ± 1.52 | 67.06 ± 18.55 | 7.10 |
| | Trimming 25% | 75.54 ± 1.17 | 72.57 ± 1.45 | 87.26 ± 0.38 | 76.30 ± 1.11 | 85.57 ± 0.34 | 99.97 ± 0.03 | 83.19 ± 1.55 | 96.87 ± 0.29 | 78.80 ± 0.52 | 88.51 ± 9.76 | 6.00 |
| | Trimming 50% | 76.91 ± 1.18 | 72.30 ± 1.63 | 86.79 ± 0.53 | 77.54 ± 1.44 | 86.16 ± 0.33 | 99.91 ± 0.13 | 84.20 ± 1.75 | 97.70 ± 0.27 | 72.70 ± 0.88 | 69.83 ± 14.25 | 6.60 |
| | Trimming 75% | 78.06 ± 1.16 | 72.53 ± 1.30 | 86.45 ± 0.57 | 79.03 ± 1.16 | 87.83 ± 0.29 | 98.49 ± 0.61 | 86.59 ± 1.58 | 97.86 ± 0.24 | 61.17 ± 1.32 | 76.08 ± 10.14 | 5.10 |

## 6    DISCUSSION

This last section aims to summarize some key findings from our extensive evaluation that can potentially help in improving future HGNN related research. Here, we connect our findings to each of the fundamental questions raised in Section 1, which actually drove our research.

**Q1**: We show that the introduced message passing homophily measure allows for a deeper understanding of hypernetwork dynamics and its correlation to the HGNN models' performances, representing a more meaningful measure than previous homophily concepts to further explore and develop new ways of assessing and processing hypernetworks and experimental results.

**Q2**: We argue that three main contributions presented in this paper –Message Passing Homophily, MultiSet framework with hyperedge-dependent node representations, MultiSetMixer model with mini-batch sampling– have been directly inspired from natural properties of hypernetworks and higher-order dynamics within them, thus no longer relying on extensions of graph-based approaches. Our experimental findings initiate a compelling discussion on the implications of innovative techniques for processing hypergraph data and defining HGNNs.

**Q3**: Accross our extensive evaluation, our results suggest that the expressive power of node features alone is sufficient for a decent performance in the node classification task execution; the gap between models with inductive bias and without is far shorter than one would expect. Addressing this gap presents an open challenge for future research endeavors, and we posit the necessity for additional benchmark datasets where connectivity plays a pivotal role.

For a more in-depth discussion, please refer to the extended conclusion and discussion in Appendix C.

## 7 REPRODUCIBILITY

We include all the details about our experimental setting, including the choice of hyperparameters, the specifications of our machine and environment, the training/validation/test split, in Appendix I.1 and in Section 5. To ensure the reproducibility of our results, we will provide the source code along with the camera-ready version.

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

# Supplementary Materials

## A INTERPLAY OF MESSAGE PASSING HOMOPHILY AND MODELS' PERFORMANCES

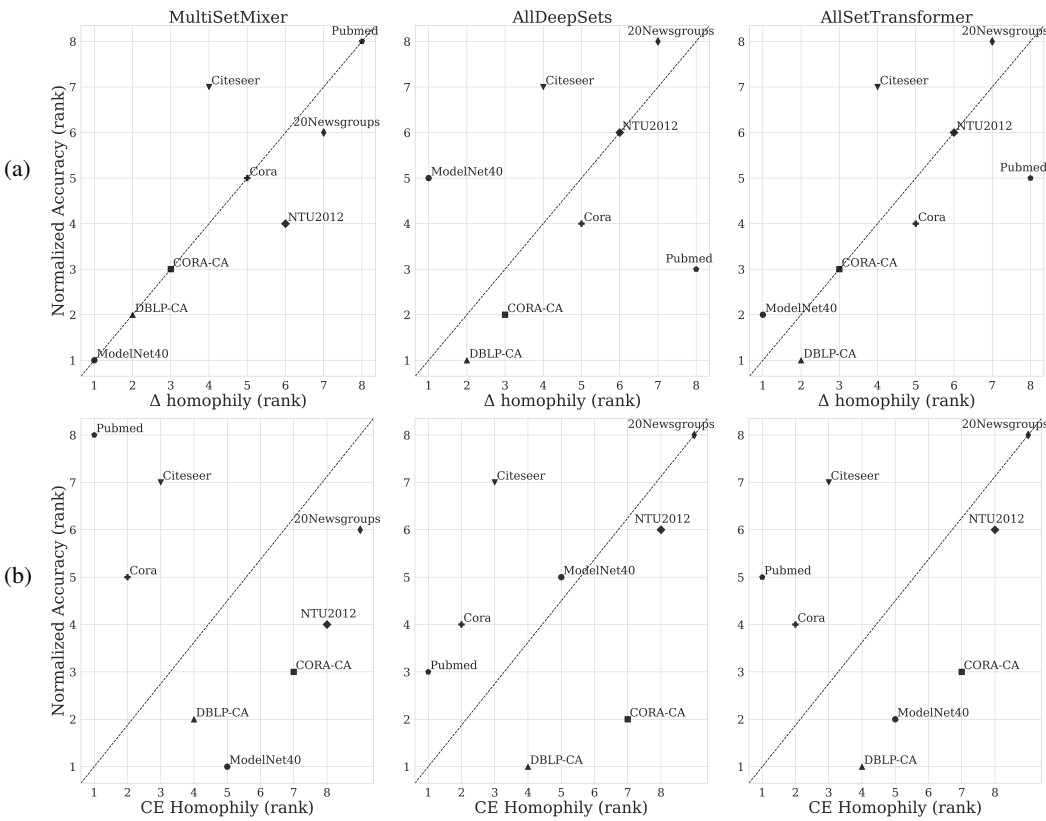

Figure 6: (a) Normalized Accuracy vs $\Delta$ homophily utilizing steps 0 and 1. (b) Normalized Accuracy vs CE Homophily (Wang et al., 2023). CE Homophily is computed as node graph homophily (Pei et al., 2020) on the clique expansion of the hypergraph. Normalized accuracy is computed using Eq. 14, utilizing different model A (depicted in the titles of the columns) while model B is 'MLP CB.' $\Delta$ homophily (Eq. 13), and CE Homophily utilized in Wang et al. (2023). Both axes on the plot represent rank values, indicating that lower values correspond to better metrics for their respective measures.

In this section, our goal is to evaluate the influence of the *inductive bias* introduced by incorporating connectivity information into the model architecture. Specifically, we aim to understand how the message passing mechanism influences performance in relation to a downstream task and to correlate this impact with the corresponding homophily measures.

We compare two measures of homophily: Clique Expanded (CE) homophily and $\Delta$ homophily. Clique Expanded (CE) homophily, employed in Wang et al. (2023), is determined by calculating node homophily Pei et al. (2020) on the graph derived from the clique-expanded hypergraphs. In addition, as detailed below, we introduce $\Delta$ homophily, derived from our dynamic definition of message passing homophily (Section 3).

$\Delta$ **homophily**    As emphasized in Section 3, extending homophily to higher-order interactions is challenging but crucial for obtaining valuable information about group compositions. In this work, we introduce the novel concept of 'message passing homophily' to capture higher-order homophily. This concept places a greater emphasis on capturing interconnections between different hyperedges through the exchange of information between nodes following the message passing scheme. In

Section 3, we define node-level 'message passing homophily' (Eq. 2), showcasing its dynamic nature, and we conduct qualitative analyses to demonstrate its applicability in studying higher-order networks. Here, we aim to illustrate that the proposed 'message passing homophily' can be leveraged to derive novel metrics for studying higher-order networks. One way to utilize 'message passing homophily' to derive novel metrics is to analyze higher-order networks from a dynamic point of view rather than simply measuring average network homophily. To achieve this, we introduce the $\Delta$ *homophily* measure. This measure is based on the assumption that if the one-step neighborhood of a node $u \in \mathcal{V}$ is predominantly homophilic (i.e., shares the same class as node $u$ itself), then the change in its homophily score at steps $t = 0$ and $t = 1$ will be around zero. Conversely, a substantial change in $u$ homophily implies that the node resides in a neighborhood characterized as heterophilic. In other words, we assess the change in node homophily following a one-step message passing iteration. Specifically, for each node, we quantify the homophily change after one-step message passing by subtracting the node homophily at step $t = 1$ from its value at previous step $t = 0$. Subsequently, after obtaining the homophily change for every node, we discretize the resulting vector with a step size of $0.1$. We then calculate the proportion of nodes falling within a bin around 0, see Eq. 13. This methodology enables us to assess, on average, the impact of one-step message passing on the hypernetwork from a dynamic point of view. It is noteworthy that $\Delta$ homophily could be defined for any steps; however, we focus on steps $t = 0$ and $t = 1$ for further analysis and comparison with AllDeepSet and AllSetTransformers, which consistently utilize one layer of two-step message passing, as suggested in the original work (see Appendix J of Chien et al. (2022)).

$$\Delta = \frac{1}{|\mathcal{V}|} \sum_{v \in \mathcal{V}} \left| h_v^0 - h_v^1 \right| < 0.1 \tag{13}$$

Here $h_v^t$ is computed according to Eq. 2.

Table 4: Relative Performance Gap: Model A and Model B represent the models with and without Inductive Bias.

| Model A | Model B | Cora | Citeseer | Pubmed | CORA-CA | DBLP-CA | Mushroom | NTU2012 | ModelNet40 | 20Newsgroups | ZOO |
|---|---|---|---|---|---|---|---|---|---|---|---|
| AllDeepSets | MLP | 3.84 | -1.40 | 1.91 | 8.96 | 6.57 | 0.05 | 6.79 | 1.39 | 0.26 | 3.97 |
| | MLP CB | 3.05 | -1.26 | 3.21 | 7.84 | 6.43 | 3.13 | 1.06 | 0.29 | -4.94 | 7.49 |
| AllSetTransformer | MLP | 6.27 | 0.45 | 1.61 | 11.16 | 6.84 | 0.04 | 8.52 | 2.69 | 0.66 | 5.90 |
| | MLP CB | 5.48 | 0.59 | 2.91 | 10.04 | 6.70 | 3.12 | 2.79 | 1.59 | -4.54 | 9.42 |
| MultiSetMixer | MLP | 6.11 | 0.72 | -1.42 | 9.35 | 5.10 | -4.06 | 9.03 | 2.84 | 6.90 | -6.46 |
| | MLP CB | 5.32 | 0.86 | -0.12 | 8.23 | 4.96 | -0.98 | 3.30 | 1.74 | 1.70 | -2.94 |

**Relative Performance Gap**   One way to quantify the impact of inductive bias in absolute value involves calculating the difference in performances between *Model A*, which leverages inductive bias, and *Model B*, which does not utilize inductive bias in its architecture. We refer to the difference between *Model A* and *Model B* as a relative performance gap (see Table 4).

The real-world datasets employed in this study span diverse domains and, as depicted in Table 1, exhibit considerable performance variations in absolute values. To alleviate the aforementioned problem, we consider the Normalized Accuracy, which can be computed as follows:

$$\text{NA} = \frac{Model\ A\ \text{Acc.} - Model\ B\ \text{Acc.}}{100 - Model\ B\ \text{Acc.}} \tag{14}$$

where Acc. accounts for the performance accuracies of the models.

**Analysis**   Figure 6 illustrates the rank dependency of normalized accuracy considering two homophily measures: (a) $\Delta$ homophily, (b) CE Homophily. The ideal correlation aligns points with the middle dashed line, indicating a positive linear relationship. We chose MLP CB since it's the strongest baseline (according to Table 1), not using connectivity within the architecture (notice that the Transformer uses positional encodings derived from connectivity). The difference between AllDeepSet and AllSetTransformer lies in the attention mechanism, highlighting its impact, while the difference between MultiSetMixer and AllSets models lies in the way message passing propagates

information (see Section 4.2). Mushroom and Zoo datasets were excluded due to Mushroom's discriminatory node features and Zoo's small hypernetwork size.

Figure 6 (a) demonstrates notably stronger correlation across all 'models A' compared to Figure 6 (b). Notably, MultiSetMixer and AllSetTransformer exhibit nearly ideal positive correlation. While Citeseer and NTU2012 deviate from the diagonal line for MultiSetMixer, it has been demonstrated in Section 5.4 that Citeseer's connectivity minimally impacts downstream task performance, in particular, we have shown that HGNN models ignore the conenctivity. In the middle and right plots of Figure 6 can be seen how the attention mechanism impacts the ModelNet40 dataset, elevating its rank from 5th to 2nd position.

At the same time, we can see that CE homophily does not exhibit any correlation with normalized accuracy, showcasing that assessing the results through this other metric does not provide any meaningful insight. In particular, CORA-CA, with a significant performance gap, attains the one of the lowest CE homophily score. Notably, Citeseer and Pubmed have the highest CE homophily scores, as demonstrated in Section 5.4, where HGNN models prioritize node features over connectivity. We hypothesize that Citeseer and Pubmed exhibits this behaviour due to a large percentage of isolated nodes. Consequently, HGNN models tend to ignore connectivity, and the presence of many self-hyperedges skews the CE homophily measure to a higher positive value.

In summary, we presented a novel way to express the homophily for higher-order networks, termed $\Delta$ homophily, based on the dynamic nature of the proposed message passing homophily (see Section 3). We demonstrated its superior correlation with HGNN models' performance compared to classical homophily measures over the clique-expanded hypergraph. Our findings underscore the crucial role of accurately expressing homophily in HGNNs, emphasizing the complexity in capturing higher-order dynamics.

## B  COMPARISON AND ANALYSIS BETWEEN MULTISET AND ALLSET FRAMEWORK PERFORMANCES

In this section, we compare the MultiSetMixer and AllSet models. Table 1 highlights MultiSet-Mixer's superior performance on three datasets: NTU2012, ModelNet40, and 20Newsgroups. The enhanced result on 20Newsgroups can be attributed to the distribution shift and the pooling operations over restricted neighbors within hyperedges. These are the results of mini-batching steps 1 and 2. We believe MultiSetMixer achieves worse performance on Mushroom and Zoo due to the negative impact of the distribution shift. We showed that this can be mitigated by applying 'k-means' to the connectivity, as demonstrated in Section 5.4 and Appendix J.3, particularly Table 15. Applying 'k-means' to the initial connectivity leads the MultiSetMixer model to achieve results like AllSetTransformer and AllDeepSets on Mushroom, Zoo, and 20Newsgroups. Furthermore, we noticed that MultiSetMixer excels in processing Computer Vision/Graphics datasets (NTU2021 and ModelNet40). This is attributed to the construction of the initial graph, which involved lifting the initial k-uniform graph by constructing hyperedges based on the one-hop neighborhood of every node. On Cora and Citeseer, MultiSetMixer outperforms AllDeepSets and performs on par with AllSet-Transformer without requiring an attention mechanism. Conversely, AllSetTransformer outperforms MultiSetMixer on CORA-CA, while MultiSetMixer achieves similar results as AllDeepSets. Finally, MultiSetMixer's lower scores on Pubmed and DBLP-CA can be attributed to mini-batching. This is due to the impossibility of processing the entire hypergraph with one forward pass, given the memory constraints for storing all hyperedge-dependent node representations.

## C  EXTENDED CONCLUSION AND DISCUSSION

This section summarizes key findings from our extensive evaluation and proposed frameworks. Here we recap the question and summarize the answers to these questions revealed by this work.

**Q1**: **Can the concept of homophily play a crucial role in HGNNs, similar to its significance in graph-based research?** We have demonstrated that the concept of homophily in higher-order networks is considerably more complicated compared to networks that exhibit only pairwise connections. We introduce a novel message passing homophily framework that is capable of characterizing homophily in hypergraphs through the distribution of node features as well as node class distribution.

In Appendix A, we present $\Delta$ homophily, based on the dynamic nature of the proposed message passing homophily (see Section 3), showing that it correlates better with HGNN models' performance than classical homophily measures over the clique-expanded hypergraph. Our findings underscore the crucial role of accurately expressing homophily in HGNNs, emphasizing its complexity and the potential in capturing higher-order dynamics. Moreover, in our experiments (see Appendix J.3) we demonstrate that rewiring hyperedges for perfect homophily leads to similar results for graph-based methods (CEGCN, CEGAT) and HGNN models, and also that simple k-means improves the performance of graph-based models. Overall, our findings potentially pave the way for new research directions in hypergraph literature, from defining new dynamic-based homophily measures to develop novel connectivity rewiring techniques.

**Q2**: **Given that current HGNNs are predominantly extensions of GNN architectures adapted to the hypergraph domain, are these extended methodologies suitable, or should we explore new strategies tailored specifically for handling hypergraph-based data?** The three main contributions presented in this paper –Message Passing Homophily, MultiSet framework with hyperedge-dependent node representations, MultiSetMixer model with mini-batch sampling– have been directly inspired from natural properties of hypernetworks and higher-order dynamics within them, thus no longer relying on extensions of graph-based approaches. Based on our experimental results and analysis, the proposed methodologies open an interesting discussion about the impact of novel ways of processing hypergraph-data and defining HGNNs. For instance, our mini-batching sampling strategy –which helps addressing scalability issues of current solutions– allowed us to realize the implicit introduction of node class distribution shifts in the process, whose study could potentially lead to the definition of meaningful connectivity rewiring techniques (as we already explore in Section 5.4). Furthermore, we show that the introduced message-passing homophily measure allows for a deeper understanding of hypernetwork dynamics and its correlation to the HGNN models' performances. Overall, and despite the fact that we also identify some common failure modes of our proposed methods (Section 5), we argue that these contributions provide a new *perspective* on dealing and processing higher-order networks that go beyond the graph domain.

**Q3**: **Are the existing hypergraph benchmarking datasets meaningful and representative enough to draw robust and valid conclusions?** In Section 5.4 and Appendices A and J.3, we demonstrate that the significant performance gap between models and MLP on Cora, CORA-CA, and DBLP-CA is primarily influenced by the largest hyperedge cardinalities. Further analysis using $\Delta$ homophily reveals that their notable improvement is strongly tied to the homophilic nature of the one-hop neighborhood. Additionally, the experimental results in Section 5.1 and 5.4 highlight challenges for current HGNNs with certain benchmark hypergraph datasets. Specifically, we find that HGNN models ignore connectivity for Citeseer, Pubmed, and 20Newsgroups, as well as for the Mushroom dataset, due to highly discriminative features. Furthermore, we observe that models that do not rely on inductive bias (i.e. do not use connectivity in the architecture), consistently exhibit good performance across the majority of datasets. This suggests that the expressive power of node features alone is sufficient for efficient task execution. Addressing this gap presents an open challenge for future research endeavors, and we posit the necessity for additional benchmark datasets where connectivity plays a pivotal role. In addition to this, we believe it would be also interesting to analyze datasets involving higher-order relationships where node classes explicitly depend on hyperedges, as introduced in Choe et al. (2023), and which can represent a insightful line of research to further exploit hyperedge-based node representations.

## D    EXTENDED RELATED WORKS ON HYPERGRAPH NEURAL NETWORKS

Numerous machine-learning techniques have been developed for processing hypergraph data. One commonly used approach in early literature is to transform the hypergraph into a graph through clique expansion (CE). This technique involves substituting each hyperedge with an edge for every pair of vertices within the hyperedge, creating a graph that can be analyzed using graph-based algorithms (Agarwal et al., 2006; Zhou et al., 2006; Zhang et al., 2018; Li & Milenkovic, 2017).

Several techniques have been proposed that use Hypergraph Neural Networks (HGNNs) for semi-supervised learning. One of the earliest methods extends the graph convolution operator by incorporating the normalized hypergraph Laplacian (Feng et al., 2019). As pointed out in Dong et al. (2020), spectral convolution with the normalized Laplacian corresponds to performing a weighted

CE of the hypergraph. HyperGCN (Yadati et al., 2019) employs mediators for incomplete CE on the hypergraph, which reduces the number of edges required to represent a hyperedge from a quadratic to a linear number of edges. The information diffusion is then carried out using spectral convolution for hypergraph-based semi-supervised learning. Hypergraph Convolution and Hypergraph Attention (HCHA) (Bai et al., 2021) employs modified degree normalizations and attention weights, with the attention weights depending on node and hyperedge features.

CE may cause the loss of important structural information and result in suboptimal learning performance (Hein et al., 2013; Chien et al., 2022). Furthermore, these models typically obtain the best performance with shallow 2-layer architectures. Adding more layers can lead to reduced performance due to oversmoothing (Huang & Yang, 2021). In the recent study Chen & Zhang (2022), an attempt was made to address oversmoothing in this type of network by incorporating residual connections; however, the method still relies on using hypergraph Laplacians to build a weighted graph through clique expansion. Another method presented in Yang et al. (2020) introduces a new hypergraph expansion called line expansion (LE) that treats vertices and hyperedges equally. The LE bijectively induces a homogeneous structure from the hypergraph by modeling vertex-hyperedge pairs. In addition, the LE and CE techniques require significant computational resources to transform the original hypergraph into a graph and perform subsequent computations, hence making the methods unpractical for large hypergraphs.

Another line of research explores hypergraph modeling involving a two-stage procedure: information is transmitted from nodes to hyperedges and then back from hyperedges to nodes (Wei et al., 2021; Yi & Park, 2020; Dong et al., 2020; Arya et al., 2020; Huang & Yang, 2021; Yadati et al., 2020). This procedure can be viewed as a two-step message passing mechanism. HyperSAGE (Arya et al., 2020) is a prominent early example of this line of research allowing transductive and inductive learning over hypergraphs. Although HyperSAGE has shown improvement in capturing information from hypergraph structures compared to spectral-based methods, it involves only one learnable linear transformation and cannot model arbitrary multiset function (Chien et al., 2022). Moreover, the algorithm utilizes nested loops resulting in inefficient computation and poor parallelism.

UniGNN (Huang & Yang, 2021) addresses some of these limitations by using a permutation-invariant function to aggregate vertex features within each hyperedge in the first stage and using learnable weights only during the second stage to update each vertex with its incident hyperedges. One of the variations of UniGNN, called UniGCNII addresses the oversmoothing problem, which is common for most of the methods described above. It accomplishes this by adapting GCNII (Chen et al., 2020) to hypergraphs. The AllSet method, proposed in Chien et al. (2022), employs a composition of two learnable multiset functions to model hypergraphs. It presents two model variations: the first one exploits Deep Set (Zaheer et al., 2017) and the second one Set Transformer (Lee et al., 2019). The AllSet method can be seen as a generalization of the most commonly used hypergraph HGNNs (Yadati et al., 2019; Feng et al., 2019; Bai et al., 2021; Dong et al., 2020; Arya et al., 2020). More implementation details and detailed drawbacks discussion can be found in Section 4.1. Although AllSet achieves state-of-the-art results, it suffers from the drawbacks of the message passing mechanism, including the local receptive field, resulting in a limited ability to model long-range interactions (Gu et al., 2020; Balcilar et al., 2021). Two additional issues are poor scalability to large hypergraph structures and oversmoothing that occurs when multiple layers are stacked.

Finally, we would like to mention two related papers that put the focus on hyperedge-dependent computations. On the one hand, EDHNN (Wang et al., 2023) incorporates the option of hyperedge-dependent messages from hyperedges to nodes; however, at each iteration of the message passing it aggregates all these messages to generate a unique node hidden representation, and thus it doesn't enable to keep different hyperedge-based node representations across the whole procedure –as our MultiSetMixer does. On the other hand, the work Aponte et al. (2022) does allow multiple hyperedge-based representations across the message passing, but the theoretical formulation of this unpublished paper is not clear and rigorous, and the evaluation is neither reproducible nor comparable to other hypergraph models. Hence, we argue that our MultiSet framework represents a step forward by rigorously formulating a simple but general MP framework for hypergraph modelling that is flexible enough to deal with hyperedge-based node representations and residual connections, demonstrating as well that it generalizes previous hypergraph and graph models.

# E  DETAILS OF THE IMPLEMENTED METHODS

We provide a detailed overview of the models analyzed and tested in this work. In order to make their similarities and differences more evident, we express their update steps through a standard and unified notation.

**Notation**  A hypergraph with $n$ nodes and $m$ hyperedges can be represented by an incidence matrix $\boldsymbol{H} \in \mathbb{R}^{n \times m}$. If the hyperedge $e_j$ is incident to a node $v_i$ (meaning $v_i \in e_j$), the entry in the incidence matrix $H_{i,j}$ is set to 1. Instead, if $v_i \notin e_j$, then $H_{i,j} = 0$.

We denote with $\boldsymbol{W}$ and $\boldsymbol{b}$ a learnable weight matrix and bias of a neural network, respectively. Generally, $\boldsymbol{x}_v$ and $\boldsymbol{z}_e$ are used to denote features for a node $v$ and a hyperedge $e$ respectively. Stacking all node features together we obtain the node feature matrix $\boldsymbol{X}$, while $\boldsymbol{Z}$ is instead the hyperedge feature matrix. $\sigma(\cdot)$ indicates a nonlinear activation function (such as ReLU, ELU or LeakyReLU) that depends on the model used. Finally, we use $\parallel$ to denote concatenation.

## E.1  ALLSET-LIKE MODELS

This Section addresses the models that are covered in the AllSet unified framework introduced in 4.1, and that can potentially be expressed as particular instances of equations 4 and 6. For a detailed proof of the claim for most of the following models, refer to Theorem 3.4 in Chien et al. (2022).

**CEGCN / CEGAT**

As introduced in the previous Sections, the CE of a hypergraph $\mathcal{G} = (\mathcal{V}, \mathcal{E})$ is a weighted graph obtained from $\mathcal{G}$ with the same set of nodes. In terms of incidence matrix, it can be described as $\boldsymbol{H}^{(CE)} = \boldsymbol{H}\boldsymbol{H}^T$ (Chien et al., 2022). A one-step update of the node feature matrix $\boldsymbol{X} \in \mathbb{R}^{n \times f}$ can be expressed both in a compact way as $\boldsymbol{H}^{(CE)}\boldsymbol{X}$ or directly as a node-level update rule, as

$$\boldsymbol{x}_v^{(t+1)} = \sum_{e \in \mathcal{E}_v} \sum_{u:u \in e} \boldsymbol{x}_v^{(t)}. \tag{15}$$

Some types of hypergraph convolutional layers in the literature adopt a CE-based propagation, for example generalizing popular graph-targeting models such as Graph Convolutional Networks (Kipf & Welling, 2017) and Graph Attention Networks (Veličković et al., 2017).

**HGNN**

Before describing how HGNN (Feng et al., 2019) works, it is necessary to define some notation. Let $\boldsymbol{H}$ be the hypergraph's incidence matrix. Suppose that each hyperedge $e \in \mathcal{E}$ is assigned a fixed positive weight $z_e$, and let $\boldsymbol{Z} \in \mathbb{R}^{m \times m}$ now denote the matrix stacking all these weights in the diagonal entries. Additionally, the vertex degree is defined as

$$d_v = \sum_{e \in \mathcal{E}_v} z_e \tag{16}$$

while the hyperedge degree, instead, is

$$b_e = \sum_{v:v \in e} 1. \tag{17}$$

The degree values can be used to define two diagonal matrices, $\boldsymbol{D} \in \mathbb{R}^{n \times n}$ and $\boldsymbol{B} \in \mathbb{R}^{m \times m}$.

The core of the hypergraph convolution introduced in Feng et al. (2019) can be expressed as

$$\boldsymbol{x}_v^{(t+1)} = \sigma \left( \sum_{e \in \mathcal{E}_v} \sum_{u:u \in e} z_e \boldsymbol{x}_v^{(t)} \boldsymbol{W}^{(t)} \right) \tag{18}$$

where $\sigma$ is a non-linear activation function like LeakyReLU and ELU, and $\boldsymbol{W}^{(t)} \in \mathbb{R}^{f^{(t)} \times f^{(t+1)}}$ is a weight matrix between the $(t)$-th and $(t+1)$-th layer, to be learnt during traning. Note that in this case the dimensionality of the node feature vectors $f^{(t)}$ can be layer-dependent.

The update step can be rewritten also in matrix form as

$$\boldsymbol{X}^{(t+1)} = \sigma(\boldsymbol{H}\boldsymbol{Z}\boldsymbol{H}^T\boldsymbol{X}^{(t)}\boldsymbol{W}^{(t)}), \tag{19}$$

where $\boldsymbol{X}^{(t+1)} \in \mathbb{R}^{n \times f^{(t+1)}}$ and $\boldsymbol{X}^{(t)} \in \mathbb{R}^{n \times f^{(t)}}$.

In practice, a normalized version of this update procedure is proposed. The matrix-based formulation allows to clearly express the symmetric normalization that is actually put in place through the vertex and hyperedge degree matrices $\boldsymbol{D}$ and $\boldsymbol{B}$ defined above:

$$\boldsymbol{X}^{(t+1)} = \sigma(\boldsymbol{D}^{-1/2}\boldsymbol{H}\boldsymbol{Z}\boldsymbol{B}^{-1}\boldsymbol{H}^T\boldsymbol{D}^{-1/2}\boldsymbol{X}^{(t)}\boldsymbol{W}^{(t)}). \tag{20}$$

**HCHA**

With respect to the previously described models, HCHA (Bai et al., 2021) uses a different kind of weights that depend on the node and hyperedge features. Specifically, starting from the same convolutional model proposed by Feng et al. (2019) and described in Equation 20, they explore the idea of introducing an attention learning model on $\boldsymbol{H}$.

Their starting point is the intuition that hypergraph convolution as implemented in Equation 20 implicitly puts in place some attentional mechanism, which derives from the fact that the afferent and efferent information flow to vertexes may be assigned different importance levels, which are statically encoded in the incidence matrix $\boldsymbol{H}$, hence depend only on the graph structure. In order to allow for such information on magnitude of importance to be determined dynamically and possibly vary from layer to layer, they introduce an attention learning module on the incidence matrix $\boldsymbol{H}$: instead of maintaining $\boldsymbol{H}$ as a binary matrix with predefined and fixed entries depending on the hypergraph connectivity, they suggest that its entries could be learnt during the training process. The entries of the matrix should express a probability distribution describing the degree of node-hyperedge connectivity, through non-binary and real values.

Nevertheless, the proposed hypergraph attention is only feasible when the hyperedge and vertex sets share the same homogeneous domain, otherwise, their similarities would not be compatible. In case the comparison is feasible, the computation of attentional scores is inspired by (Veličković et al., 2017): for a given vertex $v$ and a hyperedge $e$, the score is computed as

$$H_{v,e} = \frac{\exp(\sigma(\mathrm{sim}(\boldsymbol{x}_v\boldsymbol{W}, \boldsymbol{z}_e\boldsymbol{W})))}{\sum_{g \in \mathcal{E}_v} \exp(\sigma(\mathrm{sim}(\boldsymbol{x}_v\boldsymbol{W}, \boldsymbol{z}_g\boldsymbol{W})))}, \tag{21}$$

where $\sigma$ is a non-linear activation function, and sim is a similarity function defined as

$$\mathrm{sim}(\boldsymbol{x}_v, \boldsymbol{z}_e) = \boldsymbol{a}^T[\boldsymbol{x}_v \parallel \boldsymbol{z}_e] \tag{22}$$

in which $\boldsymbol{a}$ is a weight vector, and the resulting similarity value is a scalar.

**HyperGCN**

The method proposed by Yadati et al. (2019) can be described as performing two steps sequentially: first, a graph structure is defined starting from the input hypergraph, through a particular procedure, and then the well known CGN model (Kipf & Welling, 2017) for standard graph structures is executed on it. Depending on the approach followed in the first step, three slight variations of the same model can be identified: 1-HyperGCN, HyperGCN (enhancing 1-HyperGCN with so-called *mediators*) and FastHyperGCN.

Before analyzing the differences among the three techniques, we introduce some notation and express how the GCN-update step is performed. Suppose that the input hypergraph $\mathcal{G} = (\mathcal{V}, \mathcal{E})$ is equipped with initial edge weights $\{z_e\}_{e \in \mathcal{E}}$ and node features $\{\boldsymbol{x}_v\}_{v \in \mathcal{V}}$ (if missing, suppose to initialize them

randomly or with constant values). Let $\bar{\boldsymbol{A}}^{(t)}$ denote the normalized adjacency matrix associated to the graph structure at time-step $(t)$. The node-level one-step update for a specific node $v$ can be formalized as:

$$\boldsymbol{x}_v^{(t+1)} = \sigma\left(\left(\boldsymbol{W}^{(t)}\right)^T \sum_{u \in \mathcal{N}_v} \bar{A}_{u,v}^{(t)} \cdot \boldsymbol{x}_u^{(t)}\right) \tag{23}$$

in which $\boldsymbol{x}_v^{(t+1)}$ is the $(t+1)$-th step hidden representation of node $v$ and $\mathcal{N}_v$ is the set of neighbors of $v$. For what concerns $\bar{A}_{u,v}^{(t)}$, it refers to the element at index $u, v$ of $\bar{\boldsymbol{A}}^{(t)}$, which can be defined in the following ways according to the method:

1. 1-HyperGCN: starting from the hypergraph $\mathcal{G} = (\mathcal{V}, \mathcal{E})$, a simple graph is defined by considering exactly one representative simple edge for each hyperedge $e \in \mathcal{E}$, and it is defined as $(v_e, u_e)$ such that $(v_e, u_e) = \mathrm{argmax}_{v,u \in e}\|\left(\boldsymbol{W}^{(t)}\right)^T (\boldsymbol{x}_v^{(t)} - \boldsymbol{x}_u^{(t)})\|_2$. This implies that each hyperedge $e$ is represented by just one pairwise edge $(v_e, u_e)$, and this may also change from one step to the other, which leads to the graph adjacency matrix $\bar{\boldsymbol{A}}^{(t)}$ being layer-dependent, too.

2. HyperGCN: the model extends the graph construction procedure of 1-HyperGCN by also considering *mediator* nodes, that for each hyperedge $e$ consist in $K_e := \{k \in e : k \neq v_e, k \neq u_e\}$. Once the representative edge $(v_e, u_e)$ is determined and added to the newly defined graph, two edges for each mediator are also introduced, connecting the mediator to both $v_e$ and $u_e$. Because there are $2|e| - 3$ edges for each hyperedge $e$, each weight is chosen to be $\frac{1}{2|e|-3}$ in order for the weights in each hyperedge to sum to 1. The generalized Laplacian obtained this way satisfies all the properties of the HyperGCN's Laplacian (Yadati et al., 2019).

3. FastHyperGCN: in order to save training time, in this case the adjacency matrix $\bar{\boldsymbol{A}}^{(t)}$ is computed only once before training, by using only the initial node features of the input hypergraph.

**UniGCNII**

This model aims to extend to hypergraph structures the GCNII model proposed by Chen et al. (2020) for simple graph structures, that is a deep graph convolutional network that puts in place an initial residual connection and identity mapping as a way to reduce the oversmoothing problem (Huang & Yang, 2021).

Let $d_v$ denote the degree of vertex $v$, while $d_e = \frac{1}{|e|}\sum_{i \in e} d_i$ for each hyperedge $e \in \mathcal{E}$. A single node-level update step performed by UniGCNII can be expressed as:

$$\hat{\boldsymbol{x}}_v^{(t)} = \frac{1}{\sqrt{d_v}} \sum_{e \in \mathcal{E}_v} \frac{\boldsymbol{z}_e^{(t)}}{\sqrt{d_e}}, \tag{24}$$

$$\boldsymbol{x}_v^{(t+1)} = ((1-\beta)\boldsymbol{I} + \beta\boldsymbol{W}^{(t)})((1-\alpha)\hat{\boldsymbol{x}}_v^{(t)} + \alpha\boldsymbol{x}_v^{(0)}) \tag{25}$$

in which $\alpha$ and $\beta$ are hyperparameters, $\boldsymbol{I}$ is identity matrix and $\boldsymbol{x}_v^{(0)}$ is the initial feature of vertex $i$.

**HNHN**

For the HNHN model by Dong et al. (2020), hypernode and hyperedge features are supposed to share the same dimensionality $d$, hence in this case $\boldsymbol{X} \in \mathbb{R}^{n \times d}$ and $\boldsymbol{Z} \in \mathbb{R}^{m \times d}$. The update rule in this case can be easily expressed using the incidence matrix as

$$\boldsymbol{Z}^{(t+1)} = \sigma(\boldsymbol{H}^T \boldsymbol{X}^{(t)} \boldsymbol{W}_{\boldsymbol{Z}} + \boldsymbol{b}_{\boldsymbol{Z}}) \tag{26}$$

$$\boldsymbol{X}^{(t+1)} = \sigma(\boldsymbol{H} \boldsymbol{Z}^{(t+1)} \boldsymbol{W}_{\boldsymbol{X}} + \boldsymbol{b}_{\boldsymbol{X}}) \tag{27}$$

in which $\sigma$ is a nonlinear activation function, $\boldsymbol{W_X}, \boldsymbol{W_Z} \in \mathbb{R}^{d \times d}$ are weight matrices, and $\boldsymbol{b_X}, \boldsymbol{b_Z} \in \mathbb{R}^d$ are bias terms.

**AllSet**

The general formulation for the propagation setting of AllSet (Chien et al., 2022) is introduced in Subsection 4.1 and, starting from that, we now analyze the different instances of the model obtained by imposing specific design choices in the general framework.

In the practical implementation of the model, the update functions $f_{\mathcal{V} \to \mathcal{E}}$ and $f_{\mathcal{E} \to \mathcal{V}}$, that are required to be permutation invariant with respect to their first input, are parametrized and *learnt* for each dataset and task. Furthermore, the information of their second argument is not utilized in practice, hence their input can be more generally denoted as a set $\mathbb{S}$.

The two architectures AllDeepSets and AllSetTransformer are obtained in the following way, depending on whether the update functions are defined either as MLPs or Transformers:

1. AllDeepSets (Chien et al., 2022): $f_{\mathcal{V} \to \mathcal{E}}(\mathbb{S}) = f_{\mathcal{E} \to \mathcal{V}}(\mathbb{S}) = \mathrm{MLP}(\sum_{s \in \mathbb{S}} \mathrm{MLP}(s))$;

2. AllSetTransformer (Chien et al., 2022), in which the update functions are defined iteratively through multiple steps as they were first designed by Vaswani et al. (2017).

   The first set of operations corresponds to the self-attention module. Suppose that $h$ attention heads are considered: first of all, $h$ pairs of matrices $\boldsymbol{K}_i$ (keys) and $\boldsymbol{V}_i$ (values) with $i \in \{1, ..., h\}$ are computed from the input set through different MLPs. Additionally, $h$ weights $\theta_i, i \in \{1, ..., h\}$ are also learned and together with the keys and values they allow for the computation of each head-specific attention value $\boldsymbol{O}_i$ using an activation function $\omega$ (Vaswani et al., 2017). The $h$ attention heads are processed in parallel and they are then concatenated, leading to a unique vector being the result of the multi-head attention module $\mathrm{MH}_{h,\omega}$. After that, a sum operation and a Layer Normalization (LN) (Ba et al., 2016) are applied:

   $$\boldsymbol{K}_i = \mathrm{MLP}_i^K(\mathbb{S}), \boldsymbol{V}_i = \mathrm{MLP}_i^V(\mathbb{S}), \quad \text{where} \quad i \in \{1, ..., h\}, \tag{28}$$

   $$\theta \triangleq \|_{i=1}^h \theta_i, \tag{29}$$

   $$\boldsymbol{O}_i = \omega(\theta_i(\boldsymbol{K}_i)^T)\boldsymbol{V}_i, \quad \text{where} \quad i \in \{1, ..., h\}, \tag{30}$$

   $$\mathrm{MH}_{h,\omega}(\theta, \mathbb{S}, \mathbb{S}) = \|_{i=1}^h \boldsymbol{O}^{(i)}, \tag{31}$$

   $$\boldsymbol{Y} = \mathrm{LN}(\theta + \mathrm{MH}_{h,\omega}(\theta, \mathbb{S}, \mathbb{S})) \tag{32}$$

   A feed-forward module follows the self-attention computations, in which a MLP is applied to the feature matrix and then sum and LN are performed again, corresponding to the last operations to be performed:

   $$f_{\mathcal{V} \to \mathcal{E}}(\mathbb{S}) = f_{\mathcal{E} \to \mathcal{V}}(\mathbb{S}) = \mathrm{LN}(\boldsymbol{Y} + \mathrm{MLP}(\boldsymbol{Y})) \tag{33}$$

### E.2   OTHER MODELS

This Section describes the models that are considered for the experiments but that don't fall directly under the AllSet unified framework defined in Section 4.1.

**HAN**

The Heterogeneous Graph Attention Network model (Wang et al., 2019) is specifically designed for processing and performing inference on heterogeneous graphs. Heterogeneous graphs have various types of nodes and/or edges, and standard GNN models that treat all of them equally are not able to properly handle such complex information.

In order to apply this model on hypergraphs, (Chien et al., 2022) define a preprocessing step to derive a heterogeneous graph from a hypergraph. Specifically, a bipartite graph is defined such that there is a bijection between its nodes and the set of nodes and hyperedges in the original hypergraph. The nodes obtained in this way belong to one of two distinct types, that are the sets $\mathbb{V}$ and $\mathbb{E}$ (if they correspond to either a node or a hyperedge in the original hypergraph, respectively). Edges only connect nodes of two different types, and one edge exists between a node $u_v \in \mathbb{V}$ and a node $u_e \in \mathbb{E}$ if and only if $v \in e$ in the input hypergraph. We consider two types of so-called meta-paths (in this

case, paths of length 2) in the heterogeneous graph, that are $\mathbb{V} \to \mathbb{E} \to \mathbb{V}$ and $\mathbb{E} \to \mathbb{V} \to \mathbb{E}$. We denote the sets of such meta-paths as $\Phi_{\mathcal{V}}$ and $\Phi_{\mathcal{E}}$ respectively. Furthermore, let $\mathcal{N}_{u_v}^{\Phi_{\mathcal{V}}}$ denote the neighbors of node $u_v \in \mathbb{V}$ through paths $\gamma_v \in \Phi_{\mathcal{V}}$, and viceversa let $\mathcal{N}_{u_e}^{\Phi_{\mathcal{E}}}$ denote the neighbors of node $u_e \in \mathbb{E}$ through paths $\gamma_e \in \Phi_{\mathcal{V}}$.

At each step, the model updates separately and sequentially the node features of nodes in $\mathbb{V}$ and $\mathbb{E}$. Consider for example the case of nodes in $\mathbb{V}$ (for nodes in $\mathbb{E}$ the process is the same, except that $\Phi_{\mathcal{E}}$ is considered instead of $\Phi_{\mathcal{V}}$). The node-level update is performed as follows, for a certain $u \in \mathbb{V}$:

$$\hat{\boldsymbol{x}}_u^{(t)} = \boldsymbol{W}_{\Phi_{\mathcal{V}}}^{(t)} \boldsymbol{x}_u^{(t)}, \tag{34}$$

$$\boldsymbol{x}_u^{(t+1)} = \sigma \left( \sum_{w \in \mathcal{N}_u^{\Phi_{\mathcal{V}}}} \alpha_{u,w}^{\Phi_{\mathcal{V}}} \hat{\boldsymbol{x}}_u^{(t)} \right) \tag{35}$$

In the equations above, $\boldsymbol{W}_{\Phi_{\mathcal{V}}}^{(t)}$ is a meta-path dependent weight matrix while $\alpha_{u,w}^{\Phi_{\mathcal{V}}}$ is an attentional score computed between neighboring nodes in the same way as proposed in Veličković et al. (2017), through similar equations as 21 and 22. More generally, $h$ attention heads may be considered, that give rise to different attentional scores for each head and consequently multiple results for the node feature update, that are then concatenated to obtain a unique feature vector $\boldsymbol{x}_u^{(t+1)}$.

## MLP

We also add the MLP model as a baseline; this model doesn't use connectivity at all and only relies on the initial node features to predict their class. The node feature matrix $\boldsymbol{X}$ is obtained as

$$\boldsymbol{X}^{(t)} = \text{MLP}(\boldsymbol{X}^{(t-1)}) \tag{36}$$

## MMLP CB

MMLP stands for Multiple-Multilayer-Perceptrons, and it is an additional baseline model we implemented in order to test whether improving the results is achievable by deploying different tailored MLPs for different connectivity values.

In particular, 4 different MLPs are learned: 3 of them target the overall top-3 connectivity levels (most frequent sizes of hyperedges in the hypergraph), while the 4th one deals with all other hyperedges of different sizes. Specifically, we consider the sizes of all the hyperedge, order them according to the descending frequency of appearance in the graph, and select the top-3 ones. Hence, we use three separate encoders for the corresponding hyperedges (cardinalities in top-3). For the rest of the hyperedges, we used a 4th separate encoder.

## MLP CB

This model, in contrast to its counterparts, employs a sampling procedure as outlined in Section 4.2. In this procedure, we straightforwardly apply a Multilayer Perceptron to the initial features of nodes. During the training phase, we incorporate dropout by applying an MLP with distinct weights dropped out for each hyperedge, resulting in slightly different representations for nodes for each hyperedge they belong to. Furthermore, we execute the mini-batching procedure in accordance with the guidelines presented in Section 4.2. It is important to note that these two choices affect the training approach significantly so that the results of this model are very different from MLP's performances: see, for example, Table 1.

During the validation phase, dropout is not utilized, ensuring that the representations used for each hyperedge remain exactly the same. Consequently, there is no need for the readout operation in this context. Let $d_v$ denote the degree of vertex $v$; the node-level update is then described by:

$$\boldsymbol{x}_{v,e}^{(t+1)} = \boldsymbol{x}_{v,e}^{(t)} + \text{MLP}(\text{LN}(\boldsymbol{x}_{v,e}^{(t)})) \tag{37}$$

$$\boldsymbol{x}_v^{(T)} = \frac{1}{d_v} \sum_{e \in \mathcal{E}_v} \boldsymbol{x}_{v,e}^{(T)} \tag{38}$$

## Transformer

Along with MLP, we consider another simple baseline that is the basic Transformer model (Vaswani et al., 2017).

Also in this case, let $\mathbb{S}$ denote the set of input vectors, and define as $\boldsymbol{S} = \mathrm{MLP}(\mathbb{S})$ the matrix of input embeddings, obtained from the input set through a MLP. The operations performed on $\boldsymbol{S}$ generalize the ones described for the Transformer module adopted in AllSetTransformer, and they can be split in two main modules, that are the self-attention module and the feed-forward module:

1. Suppose that $h$ attention heads are considered in the self attention module. First of all, $h$ triples of matrices $\boldsymbol{K}_i$ (keys), $\boldsymbol{V}_i$ (values) and $\boldsymbol{Q}_i$ (queries) with $i \in \{1, ..., h\}$ are obtained from $\boldsymbol{S}$ through linear matrix multiplications with weight matrices $\boldsymbol{W}_i^K, \boldsymbol{W}_i^V$ and $\boldsymbol{W}_i^Q$ that are learned during training. The result for each attention module is computed through the key, query and key matrices using an activation function $\omega$ (Vaswani et al., 2017) and a normalization factor $d_k$, that corresponds to the dimension of the key and query vectors associated to each input element. The $h$ outputs of the different attention heads are then concatenated, leading to a unique result matrix. After that, a sum operation and a Layer Normalization (LN) (Ba et al., 2016) are applied:

$$\boldsymbol{K}_i = \boldsymbol{S}\boldsymbol{W}_i^K, \boldsymbol{V}_i = \boldsymbol{S}\boldsymbol{W}_i^V, \boldsymbol{Q}_i = \boldsymbol{S}\boldsymbol{W}_i^Q, \quad \text{where} \quad i \in \{1, ..., h\}, \tag{39}$$

$$\boldsymbol{O}_i = \omega \left( \frac{\boldsymbol{Q}_i(\boldsymbol{K}_i)^T}{\sqrt{d_k}} \right) \boldsymbol{V}_i, \quad \text{where} \quad i \in \{1, ..., h\}, \tag{40}$$

$$\mathrm{MH}_{h,\omega}(\boldsymbol{S}, \boldsymbol{S}) = \|_{i=1}^h \boldsymbol{O}^{(i)}, \tag{41}$$

$$\boldsymbol{Y} = \mathrm{LN}(\boldsymbol{S} + \mathrm{MH}_{h,\omega}(\boldsymbol{S}, \boldsymbol{S})) \tag{42}$$

2. As described for AllSetTransformer (Chien et al., 2022), in the feed-forward module a MLP is applied to the feature matrix, followed by a sum operation and Layer Normalization. After that, the output of the overall Transformer architecture is obtained:

$$\boldsymbol{Y}_{out} = \mathrm{LN}(\boldsymbol{Y} + \mathrm{MLP}(\boldsymbol{Y})). \tag{43}$$

## F  PROOF OF PROPOSITION 1

UniGCNII inherits the same hyperedge update rule of other hypergraph models (e.g. HGNN (Feng et al., 2019), HyperGCN (Yadati et al., 2019)), so it directly follows from Theorem 3.4 of (Chien et al., 2022) that it can be expressed through 4. By looking at the definition of the node update rule of UniGCNII (Eq. 24 and 25), we can re-express it as

$$\boldsymbol{x}_v^{(t+1)} = ((1-\beta)\boldsymbol{I} + \beta \boldsymbol{W}^{(t)}) \left( (1-\alpha) \frac{1}{\sqrt{d_v}} \sum_{e \in \mathcal{E}_v} \frac{\boldsymbol{z}_e^{(t+1)}}{\sqrt{d_e}} + \alpha \boldsymbol{x}_v^{(0)} \right) = f_{\mathcal{E} \to \mathcal{V}}(\{\boldsymbol{z}_e^{(t+1)}\}_{e \in \mathcal{E}_v}; \boldsymbol{x}_v^{(0)}).$$
$$\tag{44}$$

Note that this is a particular instance of the extended AllSet node update rule 6 where only a residual connection to the initial node features is considered. Lastly, it is straightforward to see that $f_{\mathcal{E} \to \mathcal{V}}$ is permutation invariant w.r.t the set $\{\boldsymbol{z}_e^{(t+1)}\}_{e \in \mathcal{E}_v}$, as it processes the set through a weighted mean. $\square$

## G  PROOF OF PROPOSITION 2

We prove this proposition by showing that we can obtain AllSet update rules 4-5 and 6 from our proposed MultiSet framework 7-8-9. This can easily follow by not distinguishing node representations among hyperedges, so $\mathbb{X}_v^{(t)} = \{\boldsymbol{x}_{v,e}^{(t)}\}_{e \in \mathcal{E}_v} = x_v^{(t)}$. With this particular choice, we directly get 4 from 7, and 6 can be obtained from 8 by further disregarding the hyperedge subscript –as there is only a single node representation to update. Analogously, we can get 5 from 8 if we additionally do not consider node residual connections, so $\{\mathbb{X}_v^{(k)}\}_{k=0}^t$ simply becomes $x_v^{(t)}$. Finally, the readout 9 can be defined as the identity function applied to the node representations at the last message passing step $T$. $\square$

## H    PROOF OF PROPOSITION 3

It is straightforward that functions $f_{\mathcal{V} \to \mathcal{E}}$, $f_{\mathcal{E} \to \mathcal{V}}$ and $f_{\mathcal{V} \to \mathcal{V}}$ defined in MultiSetMixer (Equations 10-12) are permutation invariant w.r.t their first argument: hyperedge update rule 10 and readout 12 process it through a mean operation, whereas the node update rule only receives a single-element set. The rest of the proof follows from the proof of Proposition 4.1 of Chien et al. (2022). □

## I    EXPERIMENTS

### I.1    HYPERPARAMETERS OPTIMIZATION

In order to implement the benchmark models, we followed the procedure described in Chien et al. (2022); in particular, the maximum epochs were set to 200 for all the models. The models were trained with categorical cross-entropy loss, and the best architecture was selected at the end of training depending on validation accuracy. For the AllDeepSets (Chien et al., 2022), AllSetTransformer (Chien et al., 2022), UniGCNII (Huang & Yang, 2021), CEGAT, CEGCN, HCHA (Bai et al., 2021), HGNN (Feng et al., 2019), HNHN (Dong et al., 2020), HyperGCN (Yadati et al., 2019), HAN(Wang et al., 2019), and HAN (mini-batching) (Wang et al., 2019) and MLP, we performed the same hyperparameter optimization proposed in Chien et al. (2022). For both the proposed model and the introduced baseline, we conducted a thorough hyperparameter search across the following values:

- Learning rate within the range of $0.001, 0.01$;
- Weight decay values from the set $0.0, 1e - 5, 1$;
- MLP hidden layer sizes of $64, 128, 256, 512$;
- Mini-batch sizes set to $256, 512$, with full-batch utilization when memory resources allow;
- The number of sampled neighbors per hyperedge ranged from $2, 3, 5, 10$.

It's important to note that the limitation of the number of sampled neighbors per hyperedge to this small range was intentional. This limitation showcases that even for datasets with large hyperedges, effective processing can be achieved by considering only a subset of neighbors.

The models' hyperparameters were optimized for a 50% split and subsequently applied to all the other splits.

### I.2    FURTHER INFORMATION ABOUT THE DATASETS

For our experiments we utilized various benchmark datasets from existing literature on hypergraph neural networks. For what concerns co-authorship networks (Cora-CA and DPBL-CA) and co-citation networks (Cora, Citeseer, and Pubmed), we relied on the datasets provided in Yadati et al. (2019). Additionally, we employed the Princeton ModelNet40 (Wu et al., 2015) and the National Taiwan University (Chen et al., 2003) dataset introduced for 3D object classification. For these two datasets, we complied with what Feng et al. (2019) and Yang et al. (2020) proposed for the construction of the hypergraphs, using both MVCNN (Su et al., 2015) and GVCNN (Feng et al., 2019) features. Additionally, we tested our model on three datasets with categorical attributes, namely 20Newsgroups, Mushroom, and ZOO, obtained from the UCI Categorical Machine Learning Repository (Dua et al., 2017). In order to construct hypergraphs for these datasets, we followed the approach described in Yadati et al. (2019), where a hyperedge is defined for all data points sharing the same categorical feature value.

Table 5: Statistics of hypergraph datasets: $|e|$ denotes the size of hyperedges while $d_v$ denotes the node degree.

| | Cora | Citeseer | Pubmed | CORACA | DBLP-CA | ZOO | 20Newsgroups | Mushroom | NTU2012 | ModelNet40 |
|---|---|---|---|---|---|---|---|---|---|---|
| $|\mathcal{E}|$ | 1579 | 1079 | 7963 | 1072 | 22363 | 43 | 100 | 298 | 2012 | 12311 |
| # classes | 7 | 6 | 3 | 7 | 6 | 7 | 4 | 2 | 67 | 40 |
| min $|e|$ | 2 | 2 | 2 | 2 | 2 | 1 | 29 | 1 | 5 | 5 |
| med $|e|$ | 3 | 2 | 3 | 3 | 3 | 40 | 537 | 72 | 5 | 5 |
| max $d_v$ | 145 | 88 | 99 | 23 | 18 | 17 | 44 | 5 | 19 | 30 |
| min $d_v$ | 0 | 0 | 0 | 0 | 1 | 17 | 1 | 5 | 1 | 1 |
| avg $d_v$ | 1.77 | 1.04 | 1.76 | 1.69 | 2.41 | 17 | 4.03 | 5 | 5 | 5 |
| med $d_v$ | 1 | 0 | 0 | 2 | 2 | 17 | 3 | 5 | 5 | 4 |
| CE Homophily | 89.74 | 89.32 | 95.24 | 80.26 | 86.88 | 82.88 | 75.25 | 85.33 | 46.07 | 24.07 |
| $\frac{1}{|\mathcal{V}|}\sum_{v\in\mathcal{V}} h_v^0$ | 84.10 | 78.25 | 82.05 | 80.81 | 88.86 | 91.13 | 81.26 | 88.05 | 53.24 | 42.16 |
| $\frac{1}{|\mathcal{V}|}\sum_{v\in\mathcal{V}} h_v^1$ | 78.08 | 74.18 | 75.73 | 76.51 | 86.01 | 85.79 | 74.78 | 84.41 | 41.95 | 29.42 |

We downloaded co-citation and co-authoring networks from Yadati et al. (2019). Below are the details on how the hypergraph was constructed. **Cora-CA**, **DBLP**: all documents co-authored by an author are in one hyperedge, following what was done in Yadati et al. (2019). Co-citation data **Citeseer**, **Pubmed**, **Cora**: all documents cited by a document are connected by a hyperedge. Each hypernode (document abstract) is represented by bag-of-words features (feature matrix $\boldsymbol{X}$).

**Citation and co-authorship datasets** In the co-citation and co-authorship networks datasets, the node features are the bag-of-words representations of the corresponding documents. In co-citation datasets (Cora, Citeseer, PubMed) all documents cited by a document are connected by a hyperedge. In co-authored datasets (CORA-CA, DBLP-CA), all documents co-authored by an author belong to the same hyperedge.

**Computer vision/graphics** The hyperedges are constructed using the $k$-nearest neighbor algorithm in which $k = 5$.

**Categorical datasets** There are instances with categorical attributes within the datasets. To construct the hypergraph, each attribute value is treated as a hyperedge, meaning that all instances (hypernodes) with the same attribute value are contained in a hyperedge. The node features of 20Newsgroups are the TF-IDF representations of news messages. The node features of mushrooms (in Mushroom dataset) represent categorical descriptions of 23 species. The node features of a zoo (in ZOO dataset) are a combination of categorical and numeric measurements describing various animals.

Table 6: Node Connectivity Statistics. For brevity we use the following notation in this table: under the columns labeled $|\mathcal{E}_v| = k$, we report the amount of nodes that belong to $k$ hyperedges. This value can be expressed in a more formal way as $|v \in \mathcal{V} : |\mathcal{E}_v| = k|$. Moreover, $|\mathcal{E}_v| = 0$, denotes the number of isolated nodes. In addition, the columns labeled "% $|\mathcal{E}_v| = k$" indicate the percentage of nodes belonging to $k$ hyperedges relatively to the total number of nodes. Finally, $\sum_{e \in \mathcal{E}} |e|$ corresponds to the number of hyperedge-dependent node representations.

| | $|\mathcal{V}|$ | $|\mathcal{E}_v|=0$ | $|\mathcal{E}_v|=1$ | $|\{v:|\mathcal{E}_v|=2\}|$ | $|\mathcal{E}_v|=3$ | $|\mathcal{E}_v|>3$ | % $|\mathcal{E}_v|=0$ | % $|\mathcal{E}_v|=1$ | % $|\mathcal{E}_v|=2$ | % $|\mathcal{E}_v|=31$ | % $|\mathcal{E}_v|>3$ | $\sum_{e\in\mathcal{E}}|e|$ |
|---|---|---|---|---|---|---|---|---|---|---|---|---|
| Cora | 2708 | 1274 | 575 | 327 | 156 | 376 | 47.05 | 21.23 | 12.08 | 5.76 | 13.88 | 6060 |
| Citeseer | 3312 | 1854 | 798 | 307 | 144 | 209 | 55.98 | 24.09 | 9.27 | 4.35 | 6.31 | 5307 |
| Pubmed | 19717 | 15877 | 339 | 313 | 292 | 2896 | 80.52 | 1.72 | 1.59 | 1.48 | 14.69 | 50506 |
| CORA-CA | 2708 | 320 | 995 | 951 | 287 | 155 | 11.82 | 36.74 | 35.12 | 10.60 | 5.72 | 4905 |
| DBLP-CA | 41302 | 0 | 8998 | 16724 | 9249 | 6331 | 0.00 | 21.79 | 40.49 | 22.39 | 15.33 | 99561 |
| Mushroom | 8124 | 0 | 0 | 0 | 0 | 8124 | 0.00 | 0.00 | 0.00 | 0.00 | 100.00 | 40620 |
| NTU2012 | 2012 | 0 | 173 | 256 | 296 | 1287 | 0.00 | 8.60 | 12.72 | 14.71 | 63.97 | 10060 |
| ModelNet40 | 12311 | 0 | 1491 | 1755 | 1594 | 7471 | 0.00 | 12.11 | 14.26 | 12.95 | 60.69 | 61555 |
| 20newsW100 | 16242 | 0 | 3053 | 3149 | 2720 | 7320 | 0.00 | 18.80 | 19.39 | 16.75 | 45.07 | 65451 |
| ZOO | 101 | 0 | 0 | 0 | 0 | 101 | 0.00 | 0.00 | 0.00 | 0.00 | 100.00 | 1717 |

## J EXPERIMENT RESULTS

### J.1 BENCHMARKING MODELS ACROSS MULTIPLE TRAINING PROPORTIONS SPLITS

Table 7: Hypergraph model performance benchmarks. Test accuracy in % averaged over 15 splits. Training split: 50%.

| Model | Cora | Citeseer | Pubmed | CORA-CA | DBLP-CA | Mushroom | NTU2012 | ModelNet40 | 20Newsgroups | ZOO | avg. ranking |
|---|---|---|---|---|---|---|---|---|---|---|---|
| AllDeepSets | 77.11 ± 1.00 | 70.67 ± 1.42 | 89.04 ± 0.45 | 82.23 ± 1.46 | 91.34 ± 0.27 | 99.96 ± 0.05 | 86.49 ± 1.86 | 96.70 ± 0.25 | 81.19 ± 0.49 | 89.10 ± 7.00 | 6.00 |
| AllSetTransformer | 79.54 ± 1.02 | 72.52 ± 0.88 | 88.74 ± 0.51 | 84.43 ± 1.14 | 91.61 ± 0.19 | 99.95 ± 0.05 | 88.22 ± 1.42 | 98.00 ± 0.12 | 81.59 ± 0.59 | 91.03 ± 7.31 | 2.85 |
| UniGCNII | 78.46 ± 1.14 | 73.05 ± 1.48 | 88.07 ± 0.47 | 83.92 ± 1.02 | 91.56 ± 0.18 | 99.89 ± 0.07 | 88.24 ± 1.56 | 97.84 ± 0.16 | 81.16 ± 0.49 | 89.61 ± 8.09 | 4.15 |
| CEGAT | 76.53 ± 1.58 | 71.58 ± 1.11 | 87.11 ± 0.49 | 77.50 ± 1.51 | 88.74 ± 0.31 | 96.81 ± 1.41 | 82.27 ± 1.60 | 92.79 ± 0.44 | nan | 44.62 ± 9.18 | 11.00 |
| CEGCN | 77.03 ± 1.31 | 70.87 ± 1.19 | 87.01 ± 0.62 | 77.55 ± 1.65 | 88.12 ± 0.25 | 94.91 ± 0.44 | 80.90 ± 1.74 | 90.04 ± 0.47 | nan | 49.23 ± 6.81 | 11.67 |
| HCHA | 79.53 ± 1.33 | 72.57 ± 1.06 | 86.97 ± 0.55 | 83.53 ± 1.12 | 91.21 ± 0.28 | 98.94 ± 0.54 | 86.60 ± 1.96 | 94.50 ± 0.33 | 80.75 ± 0.53 | 89.23 ± 6.81 | 6.75 |
| HGNN | 79.53 ± 1.33 | 72.24 ± 1.08 | 86.97 ± 0.55 | 83.45 ± 1.22 | 91.26 ± 0.26 | 98.94 ± 0.54 | 86.71 ± 1.48 | 94.50 ± 0.33 | 80.75 ± 0.52 | 89.23 ± 6.81 | 6.85 |
| HNHN | 77.68 ± 1.08 | 73.47 ± 1.36 | 87.88 ± 0.47 | 78.53 ± 1.15 | 86.73 ± 0.40 | 99.97 ± 0.04 | 88.28 ± 1.50 | 97.84 ± 0.15 | 81.53 ± 0.55 | 89.23 ± 7.85 | 5.05 |
| HyperGCN | 74.78 ± 1.11 | 66.06 ± 1.58 | 82.32 ± 0.62 | 77.48 ± 1.14 | 86.07 ± 3.32 | 69.51 ± 4.98 | 47.65 ± 5.01 | 46.10 ± 7.95 | 80.84 ± 0.49 | 51.54 ± 9.88 | 13.80 |
| HAN | 80.73 ± 1.37 | 73.69 ± 0.95 | 86.34 ± 0.61 | 84.19 ± 0.81 | 91.10 ± 0.20 | 91.33 ± 0.91 | 83.78 ± 1.75 | 93.85 ± 0.33 | 79.67 ± 0.55 | 80.26 ± 6.42 | 8.10 |
| HAN minibatch | 80.24 ± 2.17 | 73.55 ± 1.13 | 85.41 ± 2.32 | 82.04 ± 2.56 | 90.52 ± 0.50 | 93.87 ± 1.04 | 80.62 ± 2.00 | 92.06 ± 0.63 | 79.76 ± 0.56 | 70.39 ± 11.29 | 9.90 |
| MultiRepMixer | 79.38 ± 1.08 | 72.79 ± 1.12 | 85.71 ± 0.49 | 82.62 ± 1.20 | 90.97 ± 0.35 | 95.85 ± 3.21 | 88.73 ± 1.29 | 98.15 ± 0.19 | 87.83 ± 2.68 | 78.67 ± 9.08 | 6.40 |
| MLP CB | 74.06 ± 1.26 | 71.93 ± 1.53 | 85.83 ± 0.51 | 74.39 ± 1.40 | 84.91 ± 0.44 | 96.83 ± 2.18 | 85.43 ± 1.51 | 96.41 ± 0.32 | 86.13 ± 2.82 | 81.61 ± 10.98 | 9.80 |
| MMLP CB | 71.05 ± 2.03 | 69.26 ± 1.91 | 85.20 ± 0.54 | 71.16 ± 2.17 | 84.73 ± 0.42 | 95.71 ± 2.42 | nan | nan | 85.04 ± 4.04 | 83.89 ± 9.52 | 12.75 |
| MLP Classic | 73.27 ± 1.09 | 72.07 ± 1.65 | 87.13 ± 0.49 | 73.27 ± 1.09 | 84.77 ± 0.41 | 99.91 ± 0.08 | 79.70 ± 1.56 | 95.31 ± 0.28 | 80.93 ± 0.59 | 85.13 ± 6.90 | 10.40 |
| Transformer Classic | 74.15 ± 1.17 | 71.82 ± 1.51 | 87.37 ± 0.49 | 73.61 ± 1.55 | 85.26 ± 0.38 | 99.95 ± 0.08 | 82.88 ± 1.93 | 96.29 ± 0.29 | 81.17 ± 0.54 | 88.72 ± 10.25 | 9.05 |

Table 8: Hypergraph model performance benchmarks. Test accuracy in % averaged over 15 splits. Training split: 40%.

| Model | Cora | Citeseer | Pubmed | CORA-CA | DBLP-CA | Mushroom | NTU2012 | ModelNet40 | 20Newsgroups | ZOO | avg. ranking |
|---|---|---|---|---|---|---|---|---|---|---|---|
| AllDeepSets | 76.09 ± 1.22 | 70.32 ± 1.39 | 88.58 ± 0.46 | 81.32 ± 1.27 | 90.96 ± 0.24 | 99.94 ± 0.08 | 85.60 ± 1.41 | 96.71 ± 0.21 | 81.11 ± 0.43 | 89.57 ± 5.91 | 6.30 |
| AllSetTransformer | 78.81 ± 0.99 | 71.65 ± 1.05 | 88.17 ± 0.45 | 83.26 ± 1.12 | 91.26 ± 0.24 | 99.94 ± 0.09 | 87.04 ± 1.07 | 97.92 ± 0.14 | 81.30 ± 0.41 | 91.72 ± 6.38 | 3.15 |
| UniGCNII | 77.78 ± 1.15 | 72.30 ± 1.45 | 87.86 ± 0.37 | 83.39 ± 0.95 | 91.32 ± 0.19 | 99.88 ± 0.09 | 87.30 ± 1.34 | 97.86 ± 0.16 | 81.14 ± 0.45 | 89.68 ± 6.42 | 3.55 |
| CEGAT | 75.68 ± 1.09 | 70.59 ± 0.89 | 86.39 ± 0.47 | 76.91 ± 1.22 | 88.18 ± 0.31 | 96.72 ± 1.50 | 80.97 ± 1.30 | 92.46 ± 0.29 | nan | 45.27 ± 9.41 | 10.67 |
| CEGCN | 76.19 ± 1.06 | 70.08 ± 1.26 | 86.22 ± 0.50 | 76.17 ± 1.44 | 87.61 ± 0.29 | 95.00 ± 0.38 | 79.41 ± 1.26 | 89.79 ± 0.39 | nan | 51.40 ± 7.24 | 11.83 |
| HCHA | 78.87 ± 1.04 | 71.73 ± 0.91 | 86.28 ± 0.43 | 83.05 ± 0.99 | 91.04 ± 0.23 | 99.00 ± 0.48 | 85.53 ± 1.43 | 94.53 ± 0.28 | 80.77 ± 0.31 | 90.54 ± 5.29 | 6.30 |
| HGNN | 78.87 ± 1.04 | 71.44 ± 1.00 | 86.28 ± 0.43 | 82.95 ± 1.06 | 91.06 ± 0.24 | 99.00 ± 0.48 | 85.71 ± 1.37 | 94.53 ± 0.28 | 80.77 ± 0.31 | 90.54 ± 5.29 | 6.30 |
| HNHN | 76.47 ± 0.90 | 72.25 ± 1.10 | 87.17 ± 0.45 | 77.27 ± 1.11 | 86.61 ± 0.31 | 99.96 ± 0.08 | 87.14 ± 1.23 | 97.82 ± 0.17 | 81.28 ± 0.49 | 89.46 ± 6.50 | 5.30 |
| HyperGCN | 73.56 ± 0.91 | 64.65 ± 1.28 | 82.09 ± 0.67 | 76.44 ± 1.06 | 86.22 ± 2.93 | 69.49 ± 5.02 | 46.78 ± 4.61 | 45.34 ± 7.34 | 80.82 ± 0.59 | 52.80 ± 8.90 | 13.50 |
| HAN | 79.89 ± 0.78 | 73.16 ± 1.04 | 86.11 ± 0.56 | 83.84 ± 0.91 | 90.96 ± 0.19 | 91.39 ± 0.93 | 82.79 ± 1.16 | 93.83 ± 0.27 | 79.52 ± 0.47 | 80.11 ± 6.46 | 7.95 |
| HAN minibatch | 80.07 ± 1.68 | 72.30 ± 1.02 | 86.08 ± 0.84 | 82.33 ± 1.91 | nan | 93.60 ± 0.91 | 79.41 ± 1.62 | nan | 79.43 ± 0.91 | 64.84 ± 12.59 | 10.81 |
| MultiRepMixer | 78.60 ± 0.89 | 71.75 ± 1.20 | 85.00 ± 0.54 | 81.67 ± 0.84 | 89.74 ± 0.27 | 97.76 ± 1.70 | 87.07 ± 1.06 | 98.17 ± 0.14 | 86.85 ± 3.30 | 79.01 ± 8.08 | 6.60 |
| MLP CB | 72.60 ± 1.44 | 71.08 ± 1.68 | 85.14 ± 0.47 | 72.63 ± 1.31 | 84.65 ± 0.46 | 97.26 ± 2.00 | 83.72 ± 1.40 | 96.25 ± 0.25 | 87.28 ± 3.50 | 81.42 ± 11.55 | 9.50 |
| MMLP CB | 69.85 ± 1.51 | 67.46 ± 1.62 | 84.91 ± 0.45 | 70.17 ± 2.01 | 83.68 ± 0.32 | 95.23 ± 2.54 | nan | nan | 86.81 ± 3.32 | 83.57 ± 7.45 | 12.62 |
| MLP Classic | 70.61 ± 7.44 | 70.96 ± 1.65 | 86.60 ± 0.40 | 70.70 ± 7.33 | 84.42 ± 0.28 | 99.91 ± 0.09 | 77.83 ± 1.63 | 95.24 ± 0.23 | 80.95 ± 0.54 | 85.38 ± 8.02 | 10.25 |
| Transformer Classic | 72.65 ± 1.15 | 70.70 ± 1.50 | 86.79 ± 0.34 | 71.96 ± 1.03 | 84.97 ± 0.27 | 99.92 ± 0.09 | 80.69 ± 1.55 | 96.18 ± 0.24 | 80.95 ± 0.46 | 89.68 ± 7.31 | 8.90 |

Table 9: Hypergraph model performance benchmarks. Test accuracy in % averaged over 15 splits. Training split: 30%.

| Model | Cora | Citeseer | Pubmed | CORA-CA | DBLP-CA | Mushroom | NTU2012 | ModelNet40 | 20Newsgroups | ZOO | avg. ranking |
|---|---|---|---|---|---|---|---|---|---|---|---|
| AllDeepSets | 74.78 ± 1.02 | 69.10 ± 1.34 | 88.01 ± 0.39 | 79.69 ± 1.44 | 90.57 ± 0.20 | 99.95 ± 0.06 | 84.04 ± 1.39 | 96.49 ± 0.26 | 80.99 ± 0.41 | 87.04 ± 6.74 | 6.45 |
| AllSetTransformer | 77.67 ± 0.92 | 71.06 ± 1.00 | 87.62 ± 0.34 | 82.14 ± 0.96 | 91.10 ± 0.18 | 99.94 ± 0.09 | 85.73 ± 1.36 | 97.73 ± 0.21 | 81.05 ± 0.49 | 90.19 ± 6.18 | 3.30 |
| UniGCNII | 76.29 ± 1.05 | 71.38 ± 1.32 | 87.48 ± 0.35 | 81.93 ± 1.09 | 90.97 ± 0.20 | 99.88 ± 0.08 | 85.59 ± 1.60 | 97.89 ± 0.16 | 81.10 ± 0.44 | 88.33 ± 6.29 | 3.65 |
| CEGAT | 74.25 ± 1.24 | 69.75 ± 0.90 | 85.41 ± 0.44 | 75.24 ± 1.05 | 87.54 ± 0.23 | 96.86 ± 1.27 | 79.12 ± 1.60 | 91.89 ± 0.30 | nan | 42.87 ± 8.96 | 10.94 |
| CEGCN | 74.84 ± 1.35 | 69.17 ± 0.93 | 85.17 ± 0.41 | 74.83 ± 1.72 | 87.10 ± 0.25 | 95.08 ± 0.40 | 78.13 ± 1.35 | 89.34 ± 0.40 | nan | 48.70 ± 5.96 | 11.44 |
| HCHA | 77.81 ± 1.07 | 71.10 ± 1.11 | 84.97 ± 0.41 | 81.81 ± 1.08 | 90.85 ± 0.18 | 99.00 ± 0.47 | 84.34 ± 1.41 | 94.38 ± 0.28 | 80.78 ± 0.43 | 90.65 ± 5.58 | 6.30 |
| HGNN | 77.81 ± 1.07 | 70.88 ± 1.05 | 84.97 ± 0.41 | 81.78 ± 1.13 | 90.85 ± 0.16 | 99.00 ± 0.47 | 84.40 ± 1.38 | 94.38 ± 0.28 | 80.78 ± 0.43 | 90.65 ± 5.58 | 6.60 |
| HNHN | 74.85 ± 1.14 | 71.34 ± 1.03 | 86.34 ± 0.39 | 75.46 ± 1.02 | 86.33 ± 0.25 | 99.95 ± 0.07 | 84.93 ± 1.49 | 97.75 ± 0.20 | 81.28 ± 0.49 | 86.37 ± 7.96 | 5.50 |
| HyperGCN | 71.54 ± 1.26 | 63.82 ± 1.34 | 81.87 ± 0.56 | 74.44 ± 1.25 | 85.63 ± 2.89 | 69.49 ± 5.02 | 46.70 ± 4.01 | 45.28 ± 8.18 | 80.64 ± 0.48 | 55.46 ± 6.78 | 13.70 |
| HAN | 78.60 ± 1.28 | 72.44 ± 1.05 | 85.89 ± 0.44 | 82.69 ± 0.77 | 90.85 ± 0.19 | 91.47 ± 0.79 | 81.54 ± 1.44 | 93.79 ± 0.20 | 79.51 ± 0.58 | 79.81 ± 6.61 | 7.10 |
| HAN minibatch | 78.84 ± 1.19 | 72.26 ± 0.93 | 85.70 ± 0.81 | 79.81 ± 1.53 | nan | 93.59 ± 0.84 | 77.97 ± 1.63 | nan | 79.46 ± 1.10 | 45.74 ± 13.83 | 9.25 |
| MultiRepMixer | 77.43 ± 0.80 | 71.02 ± 1.19 | 84.07 ± 0.61 | 80.64 ± 0.94 | 89.55 ± 0.26 | 97.65 ± 1.11 | 85.49 ± 1.60 | 98.04 ± 0.18 | 85.97 ± 3.14 | 75.79 ± 10.03 | 6.80 |
| MLP CB | 71.14 ± 1.09 | 70.21 ± 1.14 | 84.24 ± 0.63 | 71.14 ± 1.61 | 84.17 ± 0.24 | 97.46 ± 1.59 | 81.18 ± 1.79 | 96.10 ± 0.22 | 85.94 ± 5.83 | 79.58 ± 8.22 | 10.10 |
| MMLP CB | 67.43 ± 1.12 | 66.40 ± 1.20 | 84.32 ± 0.52 | 68.53 ± 1.25 | 83.00 ± 0.30 | 95.24 ± 2.67 | nan | nan | 86.75 ± 3.67 | 82.88 ± 7.48 | 11.88 |
| MLP Classic | 66.14 ± 11.37 | 69.92 ± 1.32 | 85.86 ± 0.29 | 66.14 ± 11.37 | 83.96 ± 0.25 | 99.89 ± 0.11 | 75.08 ± 1.69 | 95.05 ± 0.31 | 80.82 ± 0.45 | 82.87 ± 7.37 | 10.80 |
| Transformer Classic | 70.41 ± 1.13 | 69.75 ± 1.33 | 86.05 ± 0.28 | 69.93 ± 0.92 | 84.57 ± 0.25 | 99.93 ± 0.10 | 78.20 ± 1.73 | 95.92 ± 0.20 | 80.87 ± 0.37 | 86.85 ± 9.97 | 9.25 |

Table 10: Hypergraph model performance benchmarks. Test accuracy in % averaged over 15 splits. Training split: 20%.

| Model | Cora | Citeseer | Pubmed | CORA-CA | DBLP-CA | Mushroom | NTU2012 | ModelNet40 | 20Newsgroups | ZOO | avg. ranking |
|---|---|---|---|---|---|---|---|---|---|---|---|
| AllDeepSets | 72.56 ± 1.60 | 67.49 ± 1.57 | 87.24 ± 0.36 | 77.24 ± 1.52 | 89.91 ± 0.26 | 99.92 ± 0.07 | 80.71 ± 1.32 | 96.30 ± 0.24 | 80.59 ± 0.33 | 84.72 ± 7.95 | 6.25 |
| AllSetTransformer | 75.69 ± 1.09 | 69.39 ± 1.30 | 86.63 ± 0.40 | 80.54 ± 0.94 | 90.72 ± 0.17 | 99.92 ± 0.07 | 82.58 ± 1.31 | 97.48 ± 0.24 | 80.82 ± 0.28 | 85.61 ± 7.29 | 3.35 |
| UniGCNII | 74.11 ± 1.28 | 70.51 ± 1.48 | 86.97 ± 0.41 | 79.41 ± 1.23 | 90.47 ± 0.19 | 99.90 ± 0.06 | 82.54 ± 1.60 | 97.83 ± 0.15 | 80.88 ± 0.32 | 88.21 ± 5.55 | 3.50 |
| CEGAT | 71.86 ± 1.42 | 68.11 ± 1.34 | 84.03 ± 0.51 | 73.18 ± 1.32 | 86.98 ± 0.27 | 96.19 ± 1.38 | 76.14 ± 1.20 | 91.34 ± 0.31 | nan | 41.22 ± 6.16 | 10.67 |
| CEGCN | 73.25 ± 1.35 | 67.23 ± 1.39 | 83.47 ± 0.52 | 72.50 ± 1.44 | 86.29 ± 0.21 | 94.98 ± 0.31 | 75.55 ± 1.37 | 88.60 ± 0.41 | nan | 49.51 ± 6.31 | 11.56 |
| HCHA | 76.04 ± 1.30 | 69.90 ± 1.25 | 83.65 ± 0.37 | 80.03 ± 0.87 | 90.53 ± 0.17 | 99.03 ± 0.43 | 81.48 ± 1.29 | 94.31 ± 0.20 | 80.60 ± 0.29 | 89.35 ± 5.89 | 5.40 |
| HGNN | 76.04 ± 1.30 | 69.59 ± 1.22 | 83.65 ± 0.37 | 80.02 ± 0.88 | 90.51 ± 0.19 | 99.03 ± 0.43 | 81.60 ± 1.24 | 94.31 ± 0.20 | 80.60 ± 0.29 | 89.35 ± 5.89 | 5.60 |
| HNHN | 72.47 ± 1.06 | 69.44 ± 1.21 | 85.10 ± 0.47 | 73.16 ± 0.99 | 85.82 ± 0.21 | 99.88 ± 0.09 | 81.56 ± 1.54 | 97.61 ± 0.24 | 80.48 ± 0.32 | 82.60 ± 7.71 | 7.20 |
| HyperGCN | 68.59 ± 1.79 | 62.08 ± 1.28 | 81.57 ± 0.43 | 71.42 ± 1.27 | 85.45 ± 2.31 | 69.36 ± 5.10 | 44.01 ± 3.47 | 45.34 ± 7.34 | 80.82 ± 0.59 | 53.25 ± 8.74 | 13.60 |
| HAN | 76.73 ± 1.18 | 71.21 ± 1.13 | 85.72 ± 0.44 | 80.83 ± 0.89 | 90.56 ± 0.15 | 91.50 ± 0.98 | 79.46 ± 1.30 | 93.77 ± 0.23 | 79.33 ± 0.45 | 78.54 ± 6.50 | 6.75 |
| HAN minibatch | 76.89 ± 1.51 | nan | 85.59 ± 0.72 | 78.55 ± 1.43 | nan | 93.22 ± 1.34 | 73.79 ± 1.54 | nan | 79.50 ± 0.42 | 47.72 ± 14.96 | 9.57 |
| MultiRepMixer | 75.64 ± 1.01 | 69.77 ± 1.30 | 82.95 ± 0.70 | 78.47 ± 1.32 | 89.24 ± 0.26 | 96.60 ± 1.78 | 82.36 ± 1.27 | 97.90 ± 0.22 | 84.98 ± 2.90 | 76.12 ± 7.88 | 6.60 |
| MLP CB | 68.07 ± 1.50 | 68.64 ± 1.34 | 83.06 ± 0.58 | 68.37 ± 1.23 | 83.43 ± 0.25 | 96.48 ± 1.89 | 77.01 ± 1.69 | 95.85 ± 0.27 | 85.66 ± 5.70 | 75.61 ± 9.73 | 10.00 |
| MMLP CB | 63.62 ± 1.52 | 64.30 ± 1.14 | 83.57 ± 0.49 | 65.64 ± 1.58 | 82.01 ± 0.31 | 93.81 ± 3.29 | nan | nan | 85.76 ± 2.79 | 78.77 ± 7.30 | 11.62 |
| MLP Classic | 51.73 ± 17.51 | 68.20 ± 1.21 | 85.09 ± 0.34 | 51.73 ± 17.51 | 83.22 ± 0.21 | 99.84 ± 0.12 | 69.35 ± 9.49 | 94.69 ± 0.34 | 80.58 ± 0.31 | 78.54 ± 8.98 | 10.95 |
| Transformer Classic | 67.34 ± 1.26 | 68.06 ± 1.39 | 85.31 ± 0.40 | 66.61 ± 1.50 | 83.93 ± 0.27 | 99.86 ± 0.13 | 74.17 ± 1.65 | 95.65 ± 0.29 | 80.51 ± 0.38 | 81.45 ± 6.81 | 9.80 |

Table 11: Hypergraph model performance benchmarks. Test accuracy in % averaged over 15 splits. Training split: 10%.

| Model | Cora | Citeseer | Pubmed | CORA-CA | DBLP-CA | Mushroom | NTU2012 | ModelNet40 | 20Newsgroups | ZOO | avg. ranking |
|---|---|---|---|---|---|---|---|---|---|---|---|
| AllDeepSets | 68.51 ± 1.64 | 64.50 ± 1.43 | **85.55 ± 0.38** | 73.67 ± 1.79 | 88.82 ± 0.25 | **99.88 ± 0.08** | 73.44 ± 1.91 | 95.96 ± 0.21 | 79.61 ± 0.36 | 76.81 ± 7.05 | 6.15 |
| AllSetTransformer | 71.82 ± 1.18 | 65.96 ± 1.48 | 84.71 ± 0.55 | 76.16 ± 1.36 | **90.09 ± 0.18** | 99.86 ± 0.09 | **75.78 ± 1.96** | 96.93 ± 0.21 | 80.18 ± 0.31 | 75.22 ± 10.78 | 4.05 |
| UniGCNII | 69.36 ± 1.63 | 66.41 ± 1.59 | 85.51 ± 0.50 | 75.84 ± 1.13 | 89.70 ± 0.25 | 99.86 ± 0.09 | 74.86 ± 2.20 | **97.58 ± 0.18** | 80.44 ± 0.26 | **79.86 ± 7.97** | **3.65** |
| CEGAT | 68.08 ± 1.65 | 64.15 ± 1.60 | 81.83 ± 0.43 | 69.04 ± 1.60 | 85.92 ± 0.23 | 96.01 ± 1.31 | 69.26 ± 2.27 | 90.17 ± 0.37 | nan | 39.20 ± 6.19 | 10.89 |
| CEGCN | 70.22 ± 1.62 | 62.68 ± 1.49 | 82.13 ± 0.44 | 67.45 ± 1.54 | 85.41 ± 0.26 | 94.85 ± 0.36 | 68.31 ± 2.08 | 87.28 ± 0.39 | nan | 49.20 ± 5.69 | 11.00 |
| HCHA | **72.76 ± 1.82** | 66.15 ± 1.27 | 82.41 ± 0.36 | 76.97 ± 0.95 | 90.00 ± 0.19 | 98.93 ± 0.41 | 74.44 ± 2.31 | 94.04 ± 0.21 | 80.23 ± 0.32 | 79.78 ± 7.89 | 4.95 |
| HGNN | **72.76 ± 1.82** | 65.69 ± 1.57 | 82.41 ± 0.36 | 76.96 ± 1.10 | 90.00 ± 0.18 | 98.93 ± 0.41 | 74.53 ± 2.44 | 94.04 ± 0.21 | 80.23 ± 0.32 | 79.78 ± 7.89 | 5.15 |
| HNHN | 67.43 ± 1.62 | 65.02 ± 1.40 | 82.33 ± 0.76 | 68.10 ± 1.67 | 84.74 ± 0.31 | 99.69 ± 0.18 | 73.82 ± 2.11 | 97.34 ± 0.25 | 80.00 ± 0.26 | 73.12 ± 6.57 | 7.80 |
| HyperGCN | 63.21 ± 1.95 | 57.81 ± 1.91 | 80.83 ± 0.46 | 65.58 ± 2.02 | 84.37 ± 1.73 | 67.56 ± 8.16 | 40.30 ± 3.67 | 45.92 ± 7.60 | 79.57 ± 0.38 | 51.96 ± 6.32 | 13.60 |
| HAN | 72.08 ± 1.47 | 67.86 ± 1.46 | 85.10 ± 0.43 | **77.48 ± 1.22** | **90.02 ± 0.17** | 91.67 ± 0.86 | 72.91 ± 1.88 | 93.52 ± 0.32 | 78.77 ± 0.49 | 70.94 ± 14.54 | 6.50 |
| HAN minibatch | 69.61 ± 6.86 | **68.25 ± 1.15** | 84.93 ± 0.65 | 76.27 ± 1.54 | nan | nan | 63.36 ± 2.66 | nan | nan | 43.62 ± 9.44 | 7.50 |
| MultiRepMixer | 71.54 ± 1.37 | 66.54 ± 1.44 | 81.40 ± 0.85 | 74.94 ± 1.25 | 88.50 ± 0.31 | 93.93 ± 2.80 | 75.05 ± 1.94 | 97.50 ± 0.24 | 83.82 ± 3.17 | 66.30 ± 7.66 | 6.80 |
| MLP CB | 62.42 ± 1.37 | 64.85 ± 1.30 | 81.43 ± 0.86 | 62.82 ± 1.80 | 82.02 ± 0.36 | 94.17 ± 2.92 | 68.80 ± 1.51 | 95.16 ± 0.26 | **85.21 ± 3.81** | 67.46 ± 9.14 | 10.10 |
| MMLP CB | 57.17 ± 1.86 | 59.40 ± 1.86 | 82.06 ± 0.59 | 59.03 ± 1.52 | 80.22 ± 0.35 | 91.88 ± 2.50 | nan | nan | 85.03 ± 2.38 | 69.95 ± 8.10 | 12.00 |
| MLP Classic | 38.64 ± 12.37 | 64.21 ± 1.53 | 83.56 ± 0.49 | 37.85 ± 11.79 | 81.88 ± 0.21 | 99.72 ± 0.15 | 63.39 ± 2.24 | 93.71 ± 0.38 | 79.63 ± 0.42 | 72.17 ± 9.21 | 10.80 |
| Transformer Classic | 61.45 ± 1.66 | 63.75 ± 1.39 | 83.86 ± 0.50 | 60.65 ± 1.87 | 82.40 ± 0.47 | 99.74 ± 0.20 | 65.14 ± 1.61 | 94.66 ± 0.42 | 79.61 ± 0.39 | 68.82 ± 8.32 | 10.25 |

Table 12: Average Rankings for All Splits and Mean Average Ranking.

| Model | AllSetTransformer | UniGCNII | HCHA | HNHN | HGNN | AllDeepSets | MultiSetMixer | HAN | Transformer Classic | MLP HP | MLP classic | CEGAT | CEGCN | MMLP HP | MLP | HyperGCN |
|---|---|---|---|---|---|---|---|---|---|---|---|---|---|---|---|---|
| 0.1 | 3.75 | 3.35 | 4.85 | 7.45 | 5.05 | 5.85 | 6.700 | 6.80 | 9.320 | 10.100 | 10.090 | 10.670 | 11.000 | 11.880 | 12.05 | 13.70 |
| 0.2 | 3.35 | 3.50 | 5.50 | 7.20 | 5.70 | 6.25 | 6.700 | 7.25 | 9.270 | 10.300 | 10.500 | 11.000 | 11.890 | 12.000 | 12.70 | 14.10 |
| 0.3 | 3.30 | 3.65 | 6.40 | 5.50 | 6.70 | 6.55 | 6.900 | 7.60 | 8.680 | 10.400 | 10.270 | 11.280 | 11.780 | 12.120 | 13.00 | 14.20 |
| 0.4 | 3.15 | 3.55 | 6.50 | 5.30 | 6.50 | 6.40 | 6.700 | 8.45 | 8.360 | 9.800 | 9.770 | 11.000 | 12.170 | 12.880 | 12.80 | 14.00 |
| 0.5 | 2.85 | 4.15 | 6.95 | 5.05 | 7.05 | 6.10 | 6.270 | 8.60 | 8.500 | 9.640 | 9.910 | 11.330 | 12.000 | 11.890 | 12.90 | 14.30 |
| avg | 3.28 | 3.64 | 6.04 | 6.10 | 6.20 | 6.23 | 6.654 | 7.74 | 8.826 | 10.048 | 10.108 | 11.056 | 11.768 | 12.154 | 12.69 | 14.06 |

## J.2 CONNECTIVITY MODIFICATION

Table 13: Adjust connectivity. Test accuracy in % averaged over 15 splits.

| Model | Type | Cora | Citeseer | Pubmed | CORA-CA | DBLP-CA | Mushroom | NTU2012 | ModelNet40 | 20Newsgroups | ZOO | avg. ranking |
|---|---|---|---|---|---|---|---|---|---|---|---|---|
| AllDeepSets | Label Based | **82.24 ± 1.12** | **75.65 ± 1.57** | **90.49 ± 0.40** | **91.12 ± 0.92** | **96.59 ± 0.17** | 99.96 ± 0.04 | **93.13 ± 1.29** | **99.52 ± 0.11** | **99.79 ± 0.13** | **91.54 ± 7.24** | **1.05** |
| | k-means | 75.20 ± 1.11 | 70.87 ± 1.54 | 88.96 ± 0.48 | 79.59 ± 1.42 | 89.75 ± 0.25 | **99.94 ± 0.09** | 84.23 ± 1.50 | 97.17 ± 0.13 | 81.18 ± 0.54 | 86.92 ± 7.73 | 2.80 |
| | Original | 77.11 ± 1.00 | 70.67 ± 1.42 | 89.04 ± 0.45 | 82.23 ± 1.46 | 91.34 ± 0.27 | **99.96 ± 0.05** | 86.49 ± 1.86 | 96.70 ± 0.25 | 81.19 ± 0.49 | 89.10 ± 7.00 | 2.15 |
| AllSetTransformer | Label Based | **83.43 ± 1.36** | **76.45 ± 1.43** | **90.19 ± 0.42** | **91.71 ± 0.89** | **96.75 ± 0.16** | 99.96 ± 0.05 | **94.81 ± 1.04** | **99.68 ± 0.09** | **99.93 ± 0.03** | **94.10 ± 6.91** | **1.05** |
| | k-means | 77.14 ± 1.46 | 72.83 ± 1.07 | 88.60 ± 0.41 | 81.92 ± 1.35 | 89.79 ± 0.30 | **99.96 ± 0.06** | 87.95 ± 1.28 | 97.29 ± 0.20 | 81.58 ± 0.55 | 88.72 ± 7.69 | 2.75 |
| | Original | 79.54 ± 1.02 | 72.52 ± 0.88 | 88.74 ± 0.51 | 84.43 ± 1.14 | 91.61 ± 0.19 | **99.95 ± 0.05** | 88.22 ± 1.42 | 98.00 ± 0.12 | 81.59 ± 0.59 | 91.03 ± 7.31 | 2.20 |
| UniGCNII | Label Based | **82.12 ± 1.11** | **75.23 ± 1.64** | **89.18 ± 0.50** | **89.80 ± 0.95** | **94.78 ± 0.13** | 99.93 ± 0.07 | **92.87 ± 1.32** | **99.31 ± 0.10** | **99.70 ± 0.10** | **94.74 ± 6.35** | **1.00** |
| | k-means | 76.49 ± 1.01 | 72.73 ± 1.50 | 88.02 ± 0.48 | 81.13 ± 1.41 | 90.13 ± 0.26 | 99.88 ± 0.07 | 88.05 ± 1.78 | 97.10 ± 0.21 | 81.06 ± 0.48 | 89.23 ± 7.52 | 3.00 |
| | Original | 78.46 ± 1.14 | 73.05 ± 1.48 | 88.07 ± 0.47 | 83.92 ± 1.02 | 91.56 ± 0.18 | 99.89 ± 0.07 | 88.24 ± 1.56 | 97.84 ± 0.16 | 81.16 ± 0.49 | 89.61 ± 8.09 | 2.00 |
| CEGAT | Label Based | **83.05 ± 1.08** | **77.82 ± 1.59** | **90.25 ± 0.39** | **91.42 ± 0.88** | **96.25 ± 0.13** | 99.91 ± 0.07 | **94.23 ± 0.77** | **99.26 ± 0.14** | OOM | **93.85 ± 7.39** | **1.00** |
| | k-means | 75.45 ± 1.54 | 72.57 ± 1.12 | 87.32 ± 0.47 | 77.11 ± 1.51 | 87.27 ± 0.29 | 97.66 ± 0.72 | 85.48 ± 1.66 | 96.39 ± 0.26 | OOM | 68.08 ± 8.28 | 2.33 |
| | Original | 76.53 ± 1.58 | 71.58 ± 1.11 | 87.11 ± 0.49 | 77.50 ± 1.51 | 88.74 ± 0.31 | 96.81 ± 1.41 | 82.27 ± 1.60 | 92.79 ± 0.44 | OOM | 44.62 ± 9.18 | 2.67 |
| CEGCN | Label Based | **83.70 ± 1.02** | **77.50 ± 1.53** | **90.08 ± 0.42** | **91.28 ± 0.97** | **96.68 ± 0.14** | 99.95 ± 0.05 | **94.03 ± 1.24** | **99.30 ± 0.14** | OOM | **95.00 ± 7.08** | **1.00** |
| | k-means | 75.89 ± 1.53 | 72.07 ± 1.18 | 87.13 ± 0.51 | 76.43 ± 1.41 | 86.76 ± 0.24 | 94.84 ± 0.47 | 85.34 ± 1.71 | 95.77 ± 0.31 | OOM | 73.72 ± 7.89 | 2.44 |
| | Original | 77.03 ± 1.31 | 70.87 ± 1.19 | 87.01 ± 0.62 | 77.55 ± 1.65 | 88.12 ± 0.25 | 94.91 ± 0.44 | 80.90 ± 1.74 | 90.04 ± 0.47 | OOM | 49.23 ± 6.81 | 2.56 |
| HCHA | Label Based | **84.06 ± 1.08** | **77.12 ± 1.37** | **88.81 ± 0.43** | **92.77 ± 0.73** | **96.70 ± 0.12** | 99.96 ± 0.06 | **95.21 ± 1.27** | **99.67 ± 0.09** | **99.93 ± 0.04** | **94.61 ± 6.97** | **1.00** |
| | k-means | 77.51 ± 1.41 | 72.62 ± 1.33 | 86.89 ± 0.48 | 81.19 ± 1.31 | 89.42 ± 0.29 | 99.56 ± 0.27 | 87.62 ± 1.33 | 96.98 ± 0.15 | 80.58 ± 0.57 | 84.10 ± 9.83 | 2.60 |
| | Original | 79.53 ± 1.33 | 72.57 ± 1.06 | 86.97 ± 0.55 | 83.53 ± 1.12 | 91.21 ± 0.28 | 98.94 ± 0.54 | 86.60 ± 1.96 | 94.50 ± 0.33 | 80.75 ± 0.53 | 89.23 ± 6.81 | 2.40 |
| HGNN | Label Based | **84.06 ± 1.08** | **77.11 ± 1.47** | **88.81 ± 0.43** | **92.86 ± 0.65** | **96.70 ± 0.11** | 99.96 ± 0.06 | **95.34 ± 1.07** | **99.67 ± 0.09** | **99.93 ± 0.04** | **94.61 ± 6.97** | **1.00** |
| | k-means | 77.51 ± 1.41 | 72.41 ± 1.55 | 86.89 ± 0.48 | 81.19 ± 1.38 | 89.42 ± 0.27 | 99.56 ± 0.27 | 87.52 ± 1.51 | 96.98 ± 0.15 | 80.58 ± 0.57 | 84.10 ± 9.83 | 2.60 |
| | Original | 79.53 ± 1.33 | 72.24 ± 1.08 | 86.97 ± 0.55 | 83.45 ± 1.22 | 91.26 ± 0.26 | 98.94 ± 0.54 | 86.71 ± 1.48 | 94.50 ± 0.33 | 86.71 ± 1.48 | 89.23 ± 6.81 | 2.40 |
| HyperGCN | Label Based | 72.88 ± 1.23 | **66.10 ± 1.79** | 82.18 ± 0.62 | 76.20 ± 1.50 | 84.86 ± 0.39 | 69.68 ± 4.90 | 43.37 ± 4.65 | **47.19 ± 6.42** | **82.14 ± 0.43** | 53.97 ± 8.24 | 1.50 |
| | k-means | 45.76 ± 1.97 | 49.96 ± 1.68 | 77.97 ± 0.75 | 47.63 ± 1.36 | 40.88 ± 4.04 | 69.53 ± 4.91 | 32.21 ± 2.57 | 41.96 ± 2.40 | 80.85 ± 0.46 | 53.46 ± 8.65 | 2.70 |
| | Original | **74.78 ± 1.11** | 66.06 ± 1.58 | **82.32 ± 0.62** | **77.48 ± 1.14** | **86.07 ± 3.32** | 69.51 ± 4.98 | **47.65 ± 5.01** | 46.10 ± 7.95 | 80.84 ± 0.49 | 51.54 ± 9.88 | 1.80 |
| MultiSetMixer | Label Based | **82.59 ± 0.94** | **76.14 ± 1.03** | **88.35 ± 0.59** | **90.86 ± 0.67** | **96.38 ± 0.22** | 99.97 ± 0.04 | **93.72 ± 1.00** | **99.56 ± 0.12** | **99.85 ± 0.06** | **91.79 ± 6.90** | **1.05** |
| | kmeans based | 76.78 ± 1.15 | 73.10 ± 1.18 | 85.84 ± 0.59 | 80.06 ± 1.45 | 88.54 ± 0.27 | **99.97 ± 0.05** | 87.75 ± 1.09 | 96.94 ± 0.26 | 81.14 ± 0.47 | 85.41 ± 6.77 | 2.55 |
| | Original | 79.38 ± 1.08 | 72.79 ± 1.12 | 85.71 ± 0.49 | 82.62 ± 1.20 | 89.87 ± 0.29 | 95.85 ± 3.21 | 88.73 ± 1.29 | 98.15 ± 0.19 | 87.83 ± 2.68 | 78.67 ± 9.08 | 2.40 |
| MLP CB | Label Based | **74.54 ± 1.51** | **72.41 ± 1.47** | **86.02 ± 0.50** | **74.71 ± 1.16** | **84.88 ± 0.38** | 99.97 ± 0.06 | **85.94 ± 1.59** | 96.38 ± 0.32 | 81.09 ± 0.52 | **87.56 ± 7.33** | **1.50** |
| | k-means | 74.53 ± 1.34 | 72.23 ± 1.55 | 85.99 ± 0.39 | 74.46 ± 1.32 | 84.78 ± 0.35 | **99.97 ± 0.04** | **86.16 ± 1.54** | 96.31 ± 0.27 | 81.09 ± 0.55 | 87.44 ± 7.75 | 2.10 |
| | Original | 74.06 ± 1.26 | 71.93 ± 1.53 | 85.83 ± 0.51 | 74.39 ± 1.40 | **84.91 ± 0.44** | 96.83 ± 2.18 | 85.43 ± 1.51 | **96.41 ± 0.32** | 86.13 ± 2.82 | 81.61 ± 10.98 | 2.40 |
| MMLP CB | Label Based | **74.39 ± 1.52** | **72.09 ± 1.68** | 84.94 ± 0.61 | **74.61 ± 1.42** | 83.68 ± 0.37 | 99.95 ± 0.07 | NA | NA | 81.31 ± 0.51 | **87.18 ± 8.10** | **1.62** |
| | k-means | 74.18 ± 1.54 | 71.21 ± 1.41 | **85.51 ± 0.47** | 74.49 ± 1.44 | 82.44 ± 0.41 | **99.97 ± 0.05** | NA | NA | 81.23 ± 0.53 | 85.90 ± 8.65 | 2.00 |
| | Original | 71.05 ± 2.03 | 69.26 ± 1.91 | 85.20 ± 0.54 | 71.16 ± 2.17 | **84.08 ± 0.42** | 95.71 ± 2.42 | NA | NA | 85.04 ± 4.04 | 83.89 ± 9.52 | 2.38 |

Table 14: Drop connectivity. Test accuracy in % averaged over 15 splits.

| Model | Type | Cora | Citeseer | Pubmed | CORA-CA | DBLP-CA | Mushroom | NTU2012 | ModelNet40 | 20Newsgroups | ZOO | avg. ranking |
|---|---|---|---|---|---|---|---|---|---|---|---|---|
| AllDeepSets | Original | 77.11 ± 1.00 | 70.67 ± 1.42 | 89.04 ± 0.45 | 82.23 ± 1.46 | 91.34 ± 0.27 | 99.96 ± 0.05 | 86.49 ± 1.86 | 96.70 ± 0.25 | 81.19 ± 0.49 | 89.10 ± 7.00 | **2.85** |
| | Random 25% | 76.65 ± 1.03 | 70.83 ± 1.70 | 88.87 ± 0.47 | 80.39 ± 1.51 | 90.36 ± 0.28 | 99.91 ± 0.09 | 85.49 ± 1.74 | 96.85 ± 0.26 | 81.15 ± 0.52 | 88.72 ± 5.97 | 4.05 |
| | Random 50% | 75.66 ± 1.18 | 70.70 ± 1.77 | 88.86 ± 0.41 | 77.97 ± 1.18 | 89.36 ± 0.25 | 99.93 ± 0.05 | 84.42 ± 1.75 | 96.88 ± 0.27 | 81.21 ± 0.39 | 84.49 ± 6.66 | 4.90 |
| | Random 75% | 74.96 ± 1.08 | 70.46 ± 1.66 | 88.76 ± 0.45 | 76.09 ± 1.27 | 87.68 ± 0.30 | 99.53 ± 0.13 | 81.70 ± 1.57 | 96.84 ± 0.27 | 81.31 ± 0.50 | 81.15 ± 6.88 | 7.50 |
| | Retention 25% | 76.87 ± 0.98 | **70.96 ± 1.82** | 88.94 ± 0.48 | 81.63 ± 1.26 | 90.93 ± 0.18 | 99.83 ± 0.09 | 85.13 ± 2.05 | 96.85 ± 0.26 | 81.09 ± 0.46 | 86.67 ± 7.26 | 3.65 |
| | Retention 50% | 75.80 ± 1.06 | 70.44 ± 1.63 | 88.84 ± 0.40 | 80.50 ± 1.38 | 90.55 ± 0.21 | 99.91 ± 0.09 | 82.25 ± 2.21 | 96.42 ± 0.23 | 81.04 ± 0.45 | 69.61 ± 9.28 | 6.75 |
| | Retention 75% | 75.52 ± 1.49 | 70.36 ± 1.71 | 88.78 ± 0.47 | 79.50 ± 1.09 | 89.48 ± 0.21 | **99.97 ± 0.04** | 78.85 ± 1.93 | 96.44 ± 0.27 | 81.19 ± 0.45 | 74.49 ± 9.97 | 6.85 |
| | Trimming 25% | 74.12 ± 1.30 | 70.95 ± 1.92 | 88.77 ± 0.45 | 74.87 ± 1.32 | 86.39 ± 0.31 | 99.85 ± 0.15 | 77.47 ± 1.67 | 96.18 ± 0.28 | **81.61 ± 0.47** | **89.23 ± 8.11** | 7.10 |
| | Trimming 50% | 75.24 ± 0.99 | 70.42 ± 1.62 | 88.87 ± 0.46 | 75.89 ± 1.53 | 87.14 ± 0.31 | 99.93 ± 0.06 | 82.76 ± 1.61 | 96.75 ± 0.23 | 81.47 ± 0.48 | 86.28 ± 8.73 | 6.15 |
| | Trimming 75% | 76.03 ± 1.39 | 70.86 ± 1.48 | 88.83 ± 0.48 | 77.50 ± 1.52 | 88.64 ± 0.27 | 99.93 ± 0.09 | 84.74 ± 1.81 | 96.82 ± 0.20 | 81.20 ± 0.48 | 86.03 ± 8.48 | 5.20 |
| AllSetTransformer | Original | 79.54 ± 1.02 | 72.52 ± 0.88 | 88.74 ± 0.51 | 84.43 ± 1.14 | 91.61 ± 0.19 | 99.95 ± 0.05 | 88.22 ± 1.42 | **98.00 ± 0.12** | 81.59 ± 0.59 | 91.03 ± 7.31 | **1.95** |
| | Random 25% | 79.11 ± 0.99 | **72.75 ± 1.14** | 88.67 ± 0.47 | 82.36 ± 1.38 | 90.61 ± 0.29 | 99.94 ± 0.09 | 87.50 ± 1.36 | 97.98 ± 0.17 | **81.70 ± 0.52** | 89.87 ± 7.66 | 3.10 |
| | Random 50% | 77.77 ± 1.34 | 72.21 ± 1.25 | 88.50 ± 0.45 | 79.73 ± 1.58 | 89.46 ± 0.27 | 99.96 ± 0.04 | 87.34 ± 1.55 | 97.83 ± 0.17 | 81.55 ± 0.66 | 89.49 ± 6.30 | 5.85 |
| | Random 75% | 76.92 ± 1.20 | 72.40 ± 1.22 | 88.54 ± 0.47 | 77.88 ± 1.74 | 87.73 ± 0.32 | 99.76 ± 0.15 | 86.31 ± 1.34 | 97.52 ± 0.20 | 81.46 ± 0.62 | 87.69 ± 6.09 | 8.10 |
| | Retention 25% | 79.19 ± 1.11 | 72.49 ± 0.86 | 88.73 ± 0.40 | 83.58 ± 1.30 | 91.18 ± 0.17 | 99.93 ± 0.09 | 87.21 ± 1.58 | 97.82 ± 0.17 | 81.63 ± 0.48 | 86.92 ± 7.18 | 4.10 |
| | Retention 50% | 78.16 ± 0.98 | 72.55 ± 1.13 | 88.70 ± 0.37 | 82.90 ± 1.15 | 90.80 ± 0.22 | 99.89 ± 0.18 | 86.67 ± 1.64 | 97.36 ± 0.21 | 81.61 ± 0.49 | 88.08 ± 7.51 | 5.20 |
| | Retention 75% | 77.38 ± 1.35 | 72.43 ± 0.98 | 88.71 ± 0.39 | 81.07 ± 1.20 | 89.83 ± 0.25 | **99.97 ± 0.04** | 85.58 ± 1.70 | 97.27 ± 0.22 | 81.58 ± 0.48 | 88.97 ± 6.91 | 5.80 |
| | Trimming 25% | 75.83 ± 1.31 | 72.39 ± 1.50 | 88.40 ± 0.45 | 76.51 ± 1.35 | 86.38 ± 0.32 | 99.84 ± 0.13 | 86.88 ± 1.66 | 97.10 ± 0.24 | 81.55 ± 0.55 | **93.08 ± 7.79** | 8.05 |
| | Trimming 50% | 77.37 ± 1.17 | 72.32 ± 1.30 | 88.49 ± 0.40 | 77.41 ± 1.73 | 87.03 ± 0.27 | 99.91 ± 0.12 | 86.86 ± 1.53 | 97.86 ± 0.21 | 81.45 ± 0.50 | 89.74 ± 8.53 | 7.50 |
| | Trimming 75% | 78.15 ± 1.11 | 72.67 ± 1.00 | 88.48 ± 0.39 | 78.91 ± 1.54 | 88.55 ± 0.26 | 99.92 ± 0.09 | 87.68 ± 1.56 | 97.90 ± 0.23 | 81.41 ± 0.61 | 91.03 ± 7.17 | 5.35 |
| CEGAT | Original | 76.53 ± 1.58 | 71.58 ± 1.11 | 87.11 ± 0.49 | 77.50 ± 1.51 | 88.74 ± 0.31 | 96.81 ± 1.41 | 82.27 ± 1.60 | 92.79 ± 0.44 | nan | 44.62 ± 9.18 | 5.17 |
| | Random 25% | 75.88 ± 1.53 | 71.81 ± 1.05 | 87.03 ± 0.47 | **78.00 ± 1.68** | 87.58 ± 0.35 | 94.56 ± 2.09 | 82.03 ± 1.47 | 93.14 ± 0.34 | nan | 46.03 ± 9.01 | 5.94 |
| | Random 50% | 75.34 ± 1.52 | 71.86 ± 1.22 | 86.91 ± 0.48 | 76.92 ± 1.00 | 86.81 ± 0.32 | 95.14 ± 2.00 | 81.73 ± 1.44 | 93.62 ± 0.35 | nan | 47.69 ± 8.78 | 6.89 |
| | Random 75% | 75.26 ± 1.45 | 72.17 ± 1.59 | 87.02 ± 0.47 | 76.37 ± 1.26 | 85.79 ± 0.35 | 96.90 ± 1.40 | 82.66 ± 1.39 | 94.54 ± 0.43 | nan | 59.87 ± 8.55 | 5.22 |
| | Retention 25% | 75.96 ± 1.16 | 71.39 ± 1.33 | 87.13 ± 0.50 | 77.35 ± 1.52 | 88.48 ± 0.30 | 96.65 ± 1.49 | 80.39 ± 1.53 | 93.20 ± 0.45 | nan | 45.26 ± 9.40 | 6.39 |
| | Retention 50% | 75.36 ± 1.30 | 71.56 ± 1.27 | 87.16 ± 0.53 | 77.35 ± 1.52 | 88.14 ± 0.31 | 96.73 ± 1.59 | 80.56 ± 2.16 | 93.86 ± 0.41 | nan | 45.38 ± 9.97 | 6.00 |
| | Retention 75% | 75.02 ± 1.64 | 72.06 ± 1.41 | 87.22 ± 0.48 | 77.20 ± 1.47 | 87.54 ± 0.28 | 97.49 ± 0.89 | 81.82 ± 1.23 | **94.94 ± 0.29** | nan | 45.38 ± 9.22 | 5.06 |
| | Trimming 25% | 75.40 ± 1.45 | **72.67 ± 1.76** | **87.68 ± 0.52** | 76.14 ± 1.10 | 85.32 ± 0.42 | 99.72 ± 0.10 | **84.94 ± 1.57** | 94.42 ± 0.33 | nan | **89.23 ± 7.38** | **3.67** |
| | Trimming 50% | 75.90 ± 1.48 | 72.15 ± 1.62 | 87.41 ± 0.51 | 76.09 ± 1.65 | 85.69 ± 0.40 | **99.80 ± 0.12** | 82.04 ± 1.32 | 93.22 ± 0.38 | nan | 67.05 ± 7.87 | 4.44 |
| | Trimming 75% | 76.19 ± 1.68 | 71.82 ± 1.32 | 87.03 ± 0.56 | 76.04 ± 0.95 | 86.18 ± 0.38 | 99.31 ± 0.24 | 81.74 ± 1.48 | 92.79 ± 0.33 | nan | 53.33 ± 6.60 | 6.22 |
| CEGCN | Original | 77.03 ± 1.31 | 70.87 ± 1.19 | 87.01 ± 0.62 | 77.55 ± 1.65 | 88.12 ± 0.25 | 94.91 ± 0.44 | 80.90 ± 1.74 | 90.04 ± 0.47 | nan | 49.23 ± 6.81 | 4.61 |
| | Random 25% | 76.08 ± 1.55 | 71.35 ± 1.44 | 86.89 ± 0.59 | 76.51 ± 1.53 | 87.01 ± 0.39 | 93.11 ± 0.46 | 80.68 ± 1.86 | 90.36 ± 0.46 | nan | 49.74 ± 6.22 | 6.22 |
| | Random 50% | 75.55 ± 1.43 | 71.42 ± 1.60 | 86.70 ± 0.48 | 75.27 ± 1.22 | 86.24 ± 0.35 | 93.28 ± 0.61 | 80.63 ± 1.78 | 90.69 ± 0.54 | nan | 56.92 ± 7.24 | 6.33 |
| | Random 75% | 75.34 ± 1.62 | 71.73 ± 1.90 | 86.97 ± 0.51 | 74.53 ± 1.56 | 85.36 ± 0.26 | 93.01 ± 0.45 | 80.56 ± 1.76 | **91.91 ± 0.54** | nan | 63.20 ± 5.59 | 6.33 |
| | Retention 25% | 76.12 ± 1.58 | 70.87 ± 1.42 | 86.94 ± 0.56 | 76.98 ± 1.53 | 87.90 ± 0.29 | 94.94 ± 0.48 | 79.20 ± 1.42 | 90.59 ± 0.59 | nan | 49.87 ± 7.59 | 5.50 |
| | Retention 50% | 75.43 ± 1.28 | 70.83 ± 1.52 | 86.95 ± 0.54 | 76.87 ± 1.49 | 87.58 ± 0.28 | 94.97 ± 0.40 | 78.53 ± 1.90 | 90.09 ± 0.56 | nan | 45.77 ± 6.88 | 6.89 |
| | Retention 75% | 75.53 ± 1.25 | 71.72 ± 1.42 | 87.11 ± 0.53 | 76.36 ± 1.42 | 87.03 ± 0.28 | 94.74 ± 0.39 | 79.82 ± 1.41 | **92.29 ± 0.46** | nan | 40.38 ± 5.42 | 5.44 |
| | Trimming 25% | 75.58 ± 1.56 | **72.26 ± 1.52** | **87.36 ± 0.51** | 74.84 ± 1.31 | 84.97 ± 0.31 | **99.60 ± 0.11** | **83.10 ± 1.69** | 91.85 ± 0.42 | nan | **87.69 ± 7.31** | **3.44** |
| | Trimming 50% | 76.57 ± 1.47 | 71.81 ± 1.44 | 87.07 ± 0.55 | 74.66 ± 1.68 | 85.24 ± 0.33 | 99.54 ± 0.18 | 80.72 ± 1.64 | 90.64 ± 0.54 | nan | 71.28 ± 6.60 | 4.00 |
| | Trimming 75% | 76.53 ± 1.50 | 71.45 ± 1.45 | 86.75 ± 0.54 | 74.56 ± 1.32 | 85.56 ± 0.33 | 99.14 ± 0.23 | 80.38 ± 1.91 | 90.06 ± 0.37 | nan | 58.46 ± 7.17 | 6.22 |
| HCHA | Original | 79.53 ± 1.33 | 72.57 ± 1.06 | 86.97 ± 0.55 | 83.53 ± 1.12 | 91.21 ± 0.28 | 98.94 ± 0.54 | 86.60 ± 1.96 | 94.50 ± 0.33 | **80.75 ± 0.53** | 89.23 ± 6.81 | **2.40** |
| | Random 25% | 78.74 ± 1.30 | 72.33 ± 1.28 | 86.84 ± 0.56 | 81.98 ± 1.34 | 90.09 ± 0.35 | 98.55 ± 0.55 | 85.94 ± 1.76 | 94.78 ± 0.28 | 80.16 ± 0.46 | 89.10 ± 6.71 | 4.30 |
| | Random 50% | 77.65 ± 1.46 | 72.11 ± 1.42 | 86.67 ± 0.48 | 79.23 ± 1.41 | 88.88 ± 0.35 | 98.54 ± 0.38 | 85.32 ± 1.75 | 95.17 ± 0.28 | 79.68 ± 0.50 | 87.56 ± 6.97 | 6.60 |
| | Random 75% | 76.56 ± 1.60 | 72.23 ± 1.33 | 86.72 ± 0.56 | 77.11 ± 1.28 | 87.07 ± 0.34 | 98.59 ± 0.76 | 84.88 ± 1.44 | **95.57 ± 0.34** | 79.49 ± 0.43 | 82.18 ± 6.58 | 7.30 |
| | Retention 25% | 79.09 ± 1.25 | 72.29 ± 1.17 | 86.95 ± 0.52 | 83.06 ± 1.09 | 90.63 ± 0.25 | 98.82 ± 0.50 | 85.56 ± 1.66 | 94.73 ± 0.34 | 80.27 ± 0.44 | 86.03 ± 5.20 | 3.85 |
| | Retention 50% | 77.77 ± 1.38 | 72.20 ± 1.20 | 86.82 ± 0.48 | 82.16 ± 1.27 | 90.17 ± 0.29 | 98.22 ± 0.28 | 84.80 ± 1.79 | 94.54 ± 0.22 | 79.96 ± 0.44 | 75.77 ± 6.86 | 6.70 |
| | Retention 75% | 77.05 ± 1.53 | 72.37 ± 1.20 | 86.79 ± 0.47 | 80.79 ± 0.95 | 88.97 ± 0.24 | 97.62 ± 0.30 | 84.35 ± 1.45 | 95.21 ± 0.30 | 79.40 ± 0.52 | 84.36 ± 6.77 | 6.70 |
| | Trimming 25% | 75.68 ± 1.21 | 72.15 ± 1.72 | 86.83 ± 0.47 | 75.94 ± 1.35 | 85.85 ± 0.40 | **99.87 ± 0.11** | 85.60 ± 1.92 | 95.02 ± 0.26 | 80.66 ± 0.56 | **91.92 ± 7.10** | 5.50 |
| | Trimming 50% | 77.76 ± 1.28 | 72.10 ± 1.49 | 86.96 ± 0.48 | 77.26 ± 1.17 | 86.57 ± 0.36 | 99.84 ± 0.11 | 84.58 ± 1.37 | 94.73 ± 0.28 | 80.18 ± 0.60 | 83.08 ± 8.72 | 6.45 |
| | Trimming 75% | 78.38 ± 1.30 | 72.22 ± 1.12 | 86.81 ± 0.53 | 78.82 ± 0.95 | 88.09 ± 0.27 | 99.73 ± 0.18 | 86.05 ± 1.60 | 94.61 ± 0.27 | 80.02 ± 0.57 | 89.36 ± 8.49 | 5.20 |
| HGNN | Original | 79.53 ± 1.33 | 72.24 ± 1.08 | 86.97 ± 0.55 | 83.45 ± 1.22 | 91.26 ± 0.26 | 98.94 ± 0.54 | 86.71 ± 1.48 | 94.50 ± 0.33 | **80.75 ± 0.52** | 89.23 ± 6.81 | **2.50** |
| | Random 25% | 78.74 ± 1.30 | 72.15 ± 1.36 | 86.84 ± 0.56 | 81.94 ± 1.31 | 90.11 ± 0.34 | 98.55 ± 0.55 | 85.82 ± 1.65 | 94.78 ± 0.28 | 80.16 ± 0.43 | 89.10 ± 6.71 | 4.70 |
| | Random 50% | 77.65 ± 1.46 | 72.20 ± 1.62 | 86.67 ± 0.48 | 79.20 ± 1.48 | 88.84 ± 0.42 | 98.54 ± 0.42 | 85.59 ± 1.49 | 95.17 ± 0.28 | 79.68 ± 0.50 | 87.56 ± 6.97 | 5.95 |
| | Random 75% | 76.56 ± 1.60 | 72.16 ± 1.56 | 86.72 ± 0.56 | 77.03 ± 1.37 | 86.95 ± 0.34 | 98.59 ± 0.76 | 85.12 ± 1.27 | **95.57 ± 0.34** | 79.50 ± 0.42 | 82.18 ± 6.58 | 7.30 |
| | Retention 25% | 79.09 ± 1.25 | 72.13 ± 1.17 | 86.95 ± 0.52 | 83.11 ± 1.09 | 90.66 ± 0.23 | 98.82 ± 0.50 | 85.33 ± 1.52 | 94.73 ± 0.34 | 80.23 ± 0.44 | 86.03 ± 5.20 | 4.25 |
| | Retention 50% | 77.77 ± 1.38 | 72.20 ± 1.32 | 86.82 ± 0.48 | 82.20 ± 1.29 | 90.21 ± 0.27 | 98.22 ± 0.28 | 84.51 ± 1.77 | 94.54 ± 0.22 | 79.93 ± 0.46 | 75.77 ± 6.86 | 6.45 |
| | Retention 75% | 77.05 ± 1.53 | **72.35 ± 1.40** | 86.79 ± 0.47 | 80.88 ± 0.93 | 89.02 ± 0.23 | 97.62 ± 0.30 | 84.19 ± 1.49 | 95.21 ± 0.30 | 79.36 ± 0.54 | 84.36 ± 6.77 | 6.60 |
| | Trimming 25% | 75.68 ± 1.21 | 71.91 ± 1.61 | 86.83 ± 0.47 | 75.73 ± 1.44 | 85.78 ± 0.41 | **99.87 ± 0.11** | 85.89 ± 1.67 | 95.02 ± 0.26 | 80.66 ± 0.56 | **91.92 ± 7.10** | 5.55 |
| | Trimming 50% | 77.76 ± 1.28 | 71.91 ± 1.50 | 86.96 ± 0.48 | 77.29 ± 1.18 | 86.48 ± 0.34 | 99.84 ± 0.11 | 84.67 ± 1.43 | 94.73 ± 0.28 | 80.18 ± 0.60 | 83.08 ± 8.72 | 6.30 |
| | Trimming 75% | 78.38 ± 1.30 | 72.07 ± 1.25 | 86.81 ± 0.53 | 78.78 ± 1.04 | 87.99 ± 0.34 | 99.73 ± 0.18 | 86.00 ± 1.55 | 94.61 ± 0.27 | 80.02 ± 0.57 | 89.36 ± 8.49 | 5.40 |
| HyperGCN | Original | 74.78 ± 1.11 | 66.06 ± 1.58 | 82.32 ± 0.62 | 77.48 ± 1.14 | 86.07 ± 3.32 | 46.51 ± 4.98 | 47.65 ± 5.01 | 46.10 ± 7.95 | **80.84 ± 0.49** | 51.54 ± 9.88 | **3.40** |
| | Random 25% | 35.60 ± 1.76 | 34.71 ± 1.62 | 68.80 ± 0.62 | 55.31 ± 1.83 | 81.18 ± 0.39 | **69.61 ± 4.77** | 57.42 ± 3.22 | 47.78 ± 7.33 | 77.50 ± 0.54 | 51.41 ± 9.82 | 6.05 |
| | Random 50% | 33.75 ± 2.58 | 39.94 ± 1.72 | 69.37 ± 0.59 | 40.11 ± 1.97 | 67.36 ± 2.94 | 67.59 ± 6.63 | 49.36 ± 3.42 | 48.12 ± 5.98 | 71.74 ± 0.58 | 51.67 ± 9.40 | 6.80 |
| | Random 75% | 42.42 ± 2.51 | 49.31 ± 1.85 | 70.99 ± 0.65 | 37.25 ± 1.94 | 50.33 ± 0.74 | 66.01 ± 8.15 | 45.31 ± 3.01 | 49.08 ± 2.52 | 62.76 ± 0.73 | 51.92 ± 9.02 | 6.60 |
| | Retention 25% | 37.56 ± 1.65 | 35.87 ± 1.80 | 68.73 ± 0.53 | 63.64 ± 1.22 | 84.26 ± 0.32 | **69.61 ± 4.81** | 61.33 ± 2.63 | 72.36 ± 3.39 | 79.24 ± 0.48 | 51.54 ± 8.84 | 4.55 |
| | Retention 50% | 34.87 ± 2.14 | 37.98 ± 1.70 | 69.04 ± 0.53 | 56.45 ± 1.70 | 77.98 ± 0.36 | 69.58 ± 4.75 | 76.59 ± 2.60 | 81.69 ± 1.75 | 75.60 ± 0.57 | 51.54 ± 9.45 | 4.70 |
| | Retention 75% | 36.71 ± 1.95 | 44.39 ± 1.69 | 69.98 ± 0.52 | 45.09 ± 2.09 | 63.74 ± 1.25 | 69.20 ± 5.16 | 77.44 ± 3.62 | **84.44 ± 2.23** | 67.99 ± 0.51 | **52.18 ± 8.61** | 4.60 |
| | Trimming 25% | 50.59 ± 1.72 | 55.15 ± 1.57 | 74.16 ± 0.66 | 52.78 ± 1.99 | 68.13 ± 0.79 | 52.37 ± 1.41 | **79.05 ± 2.74** | 81.31 ± 4.34 | 73.13 ± 0.92 | 46.67 ± 21.96 | 4.60 |
| | Trimming 50% | 36.20 ± 2.74 | 44.47 ± 1.38 | 71.35 ± 0.58 | 39.84 ± 2.35 | 55.53 ± 0.45 | 54.57 ± 7.40 | 59.52 ± 1.81 | 65.29 ± 3.52 | 68.07 ± 1.26 | 51.03 ± 10.20 | 6.70 |
| | Trimming 75% | 34.73 ± 1.52 | 36.60 ± 1.89 | 69.59 ± 0.54 | 37.81 ± 1.86 | 61.89 ± 1.82 | 61.73 ± 3.19 | 65.10 ± 2.77 | 73.05 ± 1.78 | 73.42 ± 0.63 | 50.90 ± 11.14 | 7.00 |
| UniGCNII | Original | 78.46 ± 1.14 | **73.05 ± 1.48** | 88.07 ± 0.47 | 83.92 ± 1.02 | 91.56 ± 0.18 | 99.89 ± 0.07 | 88.24 ± 1.56 | **97.84 ± 0.16** | 81.16 ± 0.49 | 89.61 ± 8.09 | **2.25** |
| | Random 25% | 78.31 ± 1.29 | 72.91 ± 1.24 | 88.09 ± 0.47 | 82.42 ± 1.05 | 90.86 ± 0.22 | 99.85 ± 0.10 | 87.82 ± 1.46 | 97.87 ± 0.19 | 81.08 ± 0.52 | 89.10 ± 7.76 | 3.55 |
| | Random 50% | 77.36 ± 1.34 | 72.54 ± 1.40 | 87.94 ± 0.52 | 80.17 ± 1.16 | 89.96 ± 0.24 | 99.85 ± 0.12 | 87.42 ± 1.44 | 97.77 ± 0.15 | 81.06 ± 0.55 | 87.31 ± 8.21 | 6.10 |
| | Random 75% | 76.70 ± 1.35 | 72.23 ± 1.81 | 87.91 ± 0.50 | 77.75 ± 1.31 | 88.54 ± 0.25 | 99.87 ± 0.09 | 87.56 ± 1.56 | 97.49 ± 0.18 | 81.06 ± 0.54 | 87.05 ± 6.50 | 7.30 |
| | Retention 25% | **78.80 ± 0.92** | 72.67 ± 1.24 | **88.10 ± 0.51** | 83.68 ± 0.96 | 91.26 ± 0.20 | 99.87 ± 0.06 | 87.56 ± 1.43 | 97.76 ± 0.16 | 81.01 ± 0.49 | 87.18 ± 7.58 | 4.05 |
| | Retention 50% | 77.18 ± 1.32 | 72.51 ± 1.54 | 88.02 ± 0.47 | 82.81 ± 1.32 | 90.99 ± 0.17 | 99.83 ± 0.08 | 87.16 ± 1.33 | 97.29 ± 0.17 | 80.82 ± 0.49 | 86.15 ± 8.49 | 7.15 |
| | Retention 75% | 76.63 ± 1.23 | 72.64 ± 1.15 | 88.07 ± 0.52 | 81.33 ± 1.27 | 90.17 ± 0.20 | 99.83 ± 0.14 | 86.71 ± 1.33 | 97.04 ± 0.16 | 80.87 ± 0.45 | 87.44 ± 7.49 | 7.20 |
| | Trimming 25% | 75.34 ± 1.26 | 72.68 ± 1.57 | 87.81 ± 0.47 | 76.18 ± 1.19 | 87.42 ± 0.30 | 99.87 ± 0.10 | 87.43 ± 1.53 | 97.00 ± 0.17 | **81.50 ± 0.47** | **92.95 ± 8.15** | 6.70 |
| | Trimming 50% | 76.75 ± 1.10 | 72.30 ± 1.64 | 87.87 ± 0.48 | 77.19 ± 1.42 | 88.00 ± 0.27 | **99.90 ± 0.10** | 87.47 ± 1.56 | 97.71 ± 0.18 | 81.22 ± 0.54 | 90.26 ± 7.40 | 6.15 |
| | Trimming 75% | 77.27 ± 1.08 | 72.69 ± 1.31 | 87.93 ± 0.52 | 78.68 ± 0.96 | 89.26 ± 0.28 | 99.84 ± 0.10 | 87.93 ± 1.69 | 97.86 ± 0.16 | 81.22 ± 0.45 | 90.64 ± 6.90 | 4.55 |
| MultiSetMixer | Original | 79.38 ± 1.08 | 72.79 ± 1.12 | 85.71 ± 0.49 | 82.62 ± 1.20 | 89.87 ± 0.29 | 95.85 ± 3.21 | 88.73 ± 1.29 | **98.15 ± 0.19** | **97.83 ± 2.61** | 78.67 ± 9.08 | **2.75** |
| | Random 25% | 78.63 ± 1.30 | 72.37 ± 1.50 | 85.71 ± 0.55 | 81.18 ± 1.16 | 89.11 ± 0.31 | 93.80 ± 4.69 | 87.92 ± 1.50 | 98.01 ± 0.19 | 76.65 ± 1.76 | 77.60 ± 9.00 | 4.75 |
| | Random 50% | 77.66 ± 1.18 | 72.24 ± 1.42 | 85.92 ± 0.45 | 78.51 ± 1.58 | 88.13 ± 0.34 | 94.36 ± 3.79 | 86.22 ± 2.21 | 97.92 ± 0.13 | 74.36 ± 1.23 | 75.53 ± 14.10 | 6.40 |
| | Random 75% | 76.59 ± 1.27 | 72.12 ± 1.43 | 86.10 ± 0.53 | 76.91 ± 1.43 | 86.42 ± 0.42 | 98.74 ± 0.90 | 85.31 ± 1.64 | 97.48 ± 0.21 | 76.53 ± 0.75 | 58.75 ± 17.97 | 7.25 |
| | Retention 25% | 78.99 ± 1.00 | 72.12 ± 1.28 | 85.73 ± 0.44 | 82.01 ± 1.56 | 89.61 ± 0.33 | | 88.17 ± 2.51 | 97.95 ± 0.19 | **88.17 ± 2.51** | 80.15 ± 8.87 | 3.85 |
| | Retention 50% | 77.88 ± 1.28 | 72.32 ± 1.36 | 85.89 ± 0.52 | 80.85 ± 1.14 | 89.24 ± 0.31 | 97.72 ± 1.42 | 84.56 ± 1.97 | 97.39 ± 0.24 | 85.04 ± 2.06 | 76.31 ± 12.45 | 5.20 |
| | Retention 75% | 77.44 ± 1.32 | 72.18 ± 1.32 | 85.93 ± 0.54 | 78.67 ± 1.22 | 87.86 ± 0.35 | 94.75 ± 3.86 | 83.94 ± 1.79 | 97.00 ± 0.26 | 84.65 ± 1.52 | 67.06 ± 18.55 | 7.10 |
| | Trimming 25% | 75.54 ± 1.17 | 72.57 ± 1.45 | **87.26 ± 0.38** | 76.30 ± 1.11 | 85.57 ± 0.34 | **99.97 ± 0.03** | 83.19 ± 1.55 | 96.87 ± 0.29 | 78.80 ± 0.52 | **88.51 ± 9.76** | 6.00 |
| | Trimming 50% | 76.91 ± 1.18 | 72.30 ± 1.63 | 86.79 ± 0.53 | 77.54 ± 1.44 | 86.16 ± 0.33 | 99.91 ± 0.13 | 84.20 ± 1.75 | 97.70 ± 0.23 | 72.70 ± 0.88 | 69.83 ± 14.25 | 6.60 |
| | Trimming 75% | 78.06 ± 1.16 | 72.53 ± 1.30 | 86.45 ± 0.57 | 79.03 ± 1.16 | 87.83 ± 0.29 | 98.49 ± 0.61 | 86.59 ± 1.58 | 97.86 ± 0.24 | 61.17 ± 1.32 | 76.08 ± 10.14 | 5.10 |
| MLP CB | Original | 74.06 ± 1.26 | 71.93 ± 1.53 | 85.83 ± 0.51 | 74.39 ± 1.40 | 84.91 ± 0.44 | 96.83 ± 2.18 | 85.43 ± 1.68 | 96.38 ± 0.29 | 81.61 ± 0.49 | 81.61 ± 10.98 | 4.25 |
| | Random 25% | 73.94 ± 1.58 | 72.21 ± 1.40 | 85.93 ± 0.49 | 74.57 ± 1.92 | 84.65 ± 0.37 | 96.34 ± 2.35 | 85.27 ± 1.68 | 96.38 ± 0.29 | 76.60 ± 1.74 | 77.14 ± 10.93 | 5.00 |
| | Random 50% | 74.06 ± 1.46 | 72.00 ± 1.70 | 85.86 ± 0.52 | 73.93 ± 1.47 | 84.54 ± 0.44 | 95.33 ± 0.34 | 84.53 ± 1.73 | 96.33 ± 0.34 | 74.17 ± 1.77 | 74.95 ± 14.05 | 6.95 |
| | Random 75% | **74.32 ± 1.44** | 72.20 ± 1.72 | 86.23 ± 0.47 | 74.26 ± 1.48 | 84.27 ± 0.39 | 99.25 ± 0.36 | 84.09 ± 1.97 | 96.33 ± 0.32 | 76.84 ± 0.86 | 65.07 ± 18.93 | 5.20 |
| | Retention 25% | 74.14 ± 1.45 | 72.01 ± 1.54 | 85.79 ± 0.43 | 74.21 ± 1.68 | 84.89 ± 0.40 | 98.11 ± 1.79 | 84.69 ± 1.84 | 96.15 ± 0.29 | **87.77 ± 2.39** | 81.33 ± 11.46 | 6.70 |
| | Retention 50% | 74.10 ± 1.56 | 71.90 ± 1.66 | 85.82 ± 0.51 | 73.16 ± 1.57 | 84.73 ± 0.53 | 98.46 ± 1.46 | 83.02 ± 2.01 | 95.81 ± 0.35 | 85.56 ± 2.37 | 76.89 ± 13.21 | 6.70 |
| | Retention 75% | 74.20 ± 1.47 | 71.73 ± 1.64 | 86.00 ± 0.39 | 72.59 ± 1.77 | 84.38 ± 0.45 | 83.94 ± 3.65 | 83.65 ± 1.47 | 95.83 ± 0.36 | 84.55 ± 1.47 | 72.78 ± 16.06 | 7.60 |
| | Trimming 25% | 74.25 ± 1.37 | 72.49 ± 1.44 | **86.97 ± 0.43** | 74.30 ± 1.31 | 84.61 ± 0.32 | **99.97 ± 0.05** | 82.23 ± 1.94 | 96.12 ± 0.38 | 79.00 ± 0.52 | **84.79 ± 7.81** | **4.15** |
| | Trimming 50% | 74.25 ± 1.40 | 72.44 ± 1.45 | 86.56 ± 0.54 | **74.75 ± 1.27** | 84.63 ± 0.32 | 99.94 ± 0.08 | 82.54 ± 1.76 | 96.19 ± 0.32 | 72.64 ± 0.84 | 64.76 ± 13.35 | 4.90 |
| | Trimming 75% | 74.09 ± 1.40 | 72.32 ± 1.44 | 86.12 ± 0.59 | 74.70 ± 1.11 | 84.63 ± 0.38 | 98.91 ± 0.59 | 83.71 ± 1.97 | 96.33 ± 0.28 | 60.80 ± 1.20 | 74.24 ± 9.22 | 5.25 |

## J.3 REWIRING CONNECTIVITY

In this experiment, we preserve the original connectivity and investigate the influence of homophilic hyperedges on performance. To do so, we adjust the given connectivity in two different ways. The first strategy aims to unveil the full potential of homophily for each dataset by dividing the given hyperedges into fully homophilic ones based on the *node labels*. In contrast, the second strategy explores the possibility of splitting hyperedges based on their *initial node features*. More in detail,

the hyperedge division results from applying multiple times $k$-means for each hyperedge $e$, varying at each iteration the number of centroids $m$ from 2 to $\min(C, |e|)$; the elbow method is then used to determine the optimal hyperedge partitioning.

AS we can see the 'Label Based' strategy enhances performance for all datasets and models, as seen in Table 15. Notably, the graph-based method CEGCN achieves similar results to HGNNs in this strategy. Additionally, on average, only CEGCN performs better with the 'k-means' strategy and mitigates distribution shifts for MultiSetMixer. These findings collectively suggest the crucial role of connectivity preprocessing, especially for graph-based models.

Table 15: Adjust connectivity. Test accuracy in % averaged over 15 splits.

| Model | Type | Cora | Citeseer | Pubmed | CORA-CA | DBLP-CA | Mushroom | NTU2012 | ModelNet40 | 20Newsgroups | ZOO | avg. ranking |
|---|---|---|---|---|---|---|---|---|---|---|---|---|
| AllDeepSets | Label Based | **82.24 ± 1.12** | **75.65 ± 1.57** | **90.49 ± 0.40** | **91.12 ± 0.92** | **96.59 ± 0.17** | 99.96 ± 0.04 | **93.13 ± 1.29** | **99.52 ± 0.11** | **99.79 ± 0.13** | **91.54 ± 7.24** | **1.05** |
| | k-means | 75.20 ± 1.11 | 70.87 ± 1.54 | 88.96 ± 0.48 | 79.59 ± 1.42 | 89.75 ± 0.25 | 99.94 ± 0.09 | 84.23 ± 1.50 | 97.17 ± 0.13 | 81.18 ± 0.54 | 86.92 ± 7.73 | 2.80 |
| | Original | 77.11 ± 1.00 | 70.67 ± 1.42 | 89.04 ± 0.45 | 82.23 ± 1.46 | 91.34 ± 0.27 | 99.96 ± 0.05 | 86.49 ± 1.86 | 96.70 ± 0.25 | 81.19 ± 0.49 | 89.10 ± 7.00 | 2.15 |
| AllSetTransformer | Label Based | **83.43 ± 1.36** | **76.45 ± 1.43** | **90.19 ± 0.42** | **91.71 ± 0.89** | **96.75 ± 0.16** | 99.96 ± 0.05 | **94.81 ± 1.04** | **99.68 ± 0.09** | **99.93 ± 0.03** | **94.10 ± 6.91** | **1.05** |
| | k-means | 77.14 ± 1.46 | 72.83 ± 1.07 | 88.60 ± 0.41 | 81.92 ± 1.35 | 89.79 ± 0.30 | 99.96 ± 0.06 | 87.95 ± 1.28 | 97.29 ± 0.20 | 81.58 ± 0.55 | 88.72 ± 7.69 | 2.75 |
| | Original | 79.54 ± 1.02 | 72.52 ± 0.88 | 88.74 ± 0.51 | 84.43 ± 1.14 | 91.61 ± 0.19 | 99.95 ± 0.05 | 88.22 ± 1.42 | 98.00 ± 0.12 | 81.59 ± 0.59 | 91.03 ± 7.31 | 2.20 |
| CEGCN | Label Based | **83.70 ± 1.02** | **77.50 ± 1.53** | **90.08 ± 0.42** | **91.28 ± 0.97** | **96.68 ± 0.14** | **99.95 ± 0.05** | **94.03 ± 1.24** | **99.30 ± 0.14** | OOM | **95.00 ± 7.08** | **1.00** |
| | k-means | 75.89 ± 1.53 | 72.07 ± 1.18 | 87.13 ± 0.51 | 76.43 ± 1.41 | 86.76 ± 0.24 | 94.84 ± 0.47 | 85.34 ± 1.71 | 95.77 ± 0.31 | OOM | 73.72 ± 7.89 | 2.44 |
| | Original | 77.03 ± 1.31 | 70.87 ± 1.19 | 87.01 ± 0.62 | 77.55 ± 1.65 | 88.12 ± 0.25 | 94.91 ± 0.44 | 80.90 ± 1.74 | 90.04 ± 0.47 | OOM | 49.23 ± 6.81 | 2.56 |
| MultiSetMixer | Label Based | **82.59 ± 0.94** | **76.14 ± 1.03** | **88.35 ± 0.59** | **90.86 ± 0.67** | **96.38 ± 0.22** | 99.97 ± 0.04 | **93.72 ± 1.00** | **99.56 ± 0.12** | **99.85 ± 0.06** | **91.79 ± 6.90** | **1.05** |
| | kmeans based | 76.78 ± 1.15 | 73.10 ± 1.18 | 85.84 ± 0.59 | 80.06 ± 1.45 | 88.54 ± 0.27 | **99.97 ± 0.05** | 87.75 ± 1.09 | 96.94 ± 0.26 | 81.14 ± 0.47 | 85.41 ± 6.77 | 2.55 |
| | Original | 79.38 ± 1.08 | 72.79 ± 1.12 | 85.71 ± 0.49 | 82.62 ± 1.20 | 89.87 ± 0.29 | 95.85 ± 3.21 | 88.73 ± 1.29 | 98.15 ± 0.19 | 87.83 ± 2.68 | 78.67 ± 9.08 | 2.40 |

## J.4 ADDITIONAL EXPERIMENTS WITH HETEROPHILIC DATASETS

Here we include additional experiments on the set of heterophilic datasets used in Wang et al. (2023): Senate, Congress, House and Walmart. In contrast to our selected dataset benchmark, they do not include node features; this implies that node attributes need to be artificially generated before hypergraph models can be applied. Table 16 shows the results of our model and AllSet-like architectures.

Looking at the results, we observe that our MultiSetMixer model outperforms AllSetTransformer and AllDeepSets on the Senate and House datasets, while being on par within a standard deviation in the other two. Overall, we observe that our MultiSetMixer architecture shows good performance over low-homophilic datasets.

Table 16: Additional Hypergraph model performance benchmarks on heterophilic datasets. Test Accuracy. Performances of AllDeepSets and AllSetTransformer are taken from Wang et al. (2023)

| | MultiSet | AllDeepSets | AllSetTransformer |
|---|---|---|---|
| **Senate** | **61.34 ± 3.45** | 48.17 ± 5.67 | 51.83 ± 5.22 |
| **Congress** | 92.13 ± 1.30 | 91.80 ± 1.53 | **92.16 ± 1.05** |
| **House** | **70.77 ± 2.03** | 67.82 ± 2.40 | 69.33 ± 2.20 |
| **Walmart** | 64.23 ± 0.41 | 64.55 ± 0.33 | **65.46 ± 0.25** |

## K SAMPLING ANALYSIS

As it has been discussed in Section 4.3, the proposed mini-batching procedure consists of two steps. At *step 1*, it samples $B$ hyperedges from $\mathcal{E}$. The hyperedge sampling over $\mathcal{E}$ can be either uniform or weighted (e.g. by taking into account hyperedge cardinalities). Then in *step 2* $L$ nodes are in turn sampled from each sampled hyperedge $e$, padding the hyperedge with $L - |e|$ special padding tokens if $|e| < L$ –consisting of $\mathbf{0}$ vectors that can be easily discarded in some computations. Overall, the shape of the obtained mini-batch $X$ has fixed size $B \times L$.

*Step 0* (hyperedge mini-batching) is particularly beneficial for large datasets; however, it can be skipped when the network fits fully into memory. Empirically, we found *step 1* (node mini-batching within a hyperedge) to be useful for two reasons: (i) pooling operations over a large set may over-squash the signal, and (ii) node batching leads to the training distribution shift, hence it can be useful to keep it even when the full hyperedge can be stored in memory.

When both *step 1* and *step 2* are employed, considering the hidden dimension size, the batch size required to be stored in memory during the forward pass is $B \times L \times d$, where $d$ represents the hidden dimension. If only *step 2* is employed, the batch size is $|\mathcal{E}| \times L \times d$, where $|\mathcal{E}|$ is the number of hyperedges within the hypernetwork. Finally, when no mini-batching steps are used, the batch size is $|\mathcal{E}| \times \max_{e \in \mathcal{E}} |e| \times d$, where $\max_{e \in \mathcal{E}} |e|$ is the size of the longest hyperedge.

**Theoretical Analysis** In this Section, we provide an analysis regarding the uniform sampling of the hyperedges in Step 1. We propose sampling $X$ mini-batches of a certain size $B$ at each iteration. At *step 1*, we sample $B$ hyperedges from $\mathcal{E}$; in *step 2*, for each hyperedge we sample a fixed number of nodes, that are randomly chosen among the ones belonging to that specific hyperedge. If the hyperedge does not contain enough samples, we use padding so that the size of the set of sampled nodes is increased to the desired value. By choosing to sample the nodes uniformly at random from the hyperedge, there is no guarantee that we will eventually sample all the nodes of each hyperedge. Indeed, sampling uniformly at random $c$ items from a set of size $n$, the probability of not sampling our desired one is $1 - \frac{c}{n}$. The probability of having to wait for $T$ independent trials before finding node $x$ among the sampled nodes is described by the geometric distribution.

Namely, let $x \in e$ and $|e| = n$, and assume the size of the considered mini-batch is $c$:

$$\mathbb{P}\left( \text{ Sample node } x \text{ from hyperedge } e \text{ for the first time at epoch } T\right) = \left(1 - \frac{c}{n}\right)^{T-1} \frac{c}{n}. \tag{45}$$

It follows that

$$\mathbb{E}\left[ \text{ \# of epochs to wait before sampling node } x\right] = \frac{n}{c}.$$

Assume now that a node $x$ belongs to $k$ hypergraphs $e_1, \ldots, e_k$ of respective sizes $n_1, \ldots, n_k$. The events $\{$Node $x$ is sampled from hyperedge $e_i$ $\}$ and $\{$Node $x$ is sampled from hyperedge $e_j$ $\}$ are independent if $i \neq j$. It follows that the random variable $\{$ \# of epochs to wait until we sample node $x$ from all the hyperedges $e_1, \ldots, e_k\}$ is the maximum of $k$ independent non-identically distributed geometric distributions. Denote by $T_i$ the random variable that corresponds to the number of epochs we have to wait before sampling sample $x$ from edge $e_i$. The exact distribution for the random variable $T$, that is, the number of epochs we have to wait until we sample node $x$ from all hyperedges $e_1, \ldots e_k$ at least once, is

$$\mathbb{P}\left(T \leq h\right) = \mathbb{P}\left(\max_{i=1,\ldots,k} T_i \leq h\right) = \prod_{i=1}^{k} \mathbb{P}\left(T_i \leq h\right) = \prod_{i=1}^{k} \left(1 - p_i\right)^h$$

It follows that

$$\mathbb{E}\left( \text{ \# of epochs to wait until we sample node } x \text{ from all the hyperedges } e_1, \ldots, e_k\right) = \tag{46}$$

$$\mathbb{E}\left(\max_{i=1,\ldots,k} T_i\right) = \sum_{i=0}^{k} \left(1 - \prod_{i=1}^{h} \left(1 - \left(1 - p_i\right)^k\right)\right) \tag{47}$$

This can't be expressed in closed form: we can use the Moment Generating Function to bound the expected value of the maximum. Alternatively, we can also try to use the inequality due to Aven (1985), so that

$$\mathbb{E}\left(\max_{i=1,\ldots,k} T_i\right) \tag{48}$$

$$\leq \max_{i=1,\ldots k} \mathbb{E}\left(T_i\right) + \sqrt{\frac{k-1}{k} \sum_{i=1}^{k} \mathbb{V}\left(T_i\right)} = \max_{i=1,\ldots k} \frac{n}{c} + \sqrt{\frac{k-1}{k} k \left[\frac{n}{c}\left(1 - \frac{n}{c}\right)\right]} \tag{49}$$

$$= \frac{n}{c} + \sqrt{k-1 \left[\frac{n}{c}\left(1 - \frac{n}{c}\right)\right]} \tag{50}$$

**Probability that a specific node is not sampled in one epoch** Let $v$ be a node and let $d_v$ be its degree. In one epoch, we "see" all hyperedges but, of course, not necessarily all their nodes. It holds that

$$\mathbb{P}\left( \text{ node } v \text{ is sampled in epoch } T \right) = 1 - \mathbb{P}\left( \text{ node } v \text{ is not sampled in epoch } T \right) \tag{51}$$

We can write the event

$$\{ \text{ node } v \text{ is not sampled in epoch } T \} = \cap_{e \ s.t. v \in e} \{ \text{ node } v \text{ is not sampled in } e \}.$$

It follows that

$$\mathbb{P}\left(\text{ node } v \text{ is sampled in epoch } T \right) = \max\left\{ 1 - \prod_{i=1}^{d_v} \frac{c}{|e_i| - 1}, \mathbb{1}_{\min_{i=1,\ldots,d_v} |e_i| < c} \right\} \quad (52)$$

Indeed, if any on the edges $v$ belongs to has a size smaller than the batch size for nodes ($c$), the node is for sure seen in the first epoch.

## L NODE HOMOPHILY

In this Section, we report the node homophily plots for the datasets not illustrated in Figure 1. For each dataset, we choose to illustrate 3 different levels of node homophily, respectively $0, 1$ and $10-$ level homophily, using Equation 2 at $t = 0, 1$ and $10$ (left, middle, and right plots respectively). Horizontal lines depict class mean homophily, with numbers above indicating the number of visualized points per class.

### L.1 FIGURE - NODE HOMOPHILY

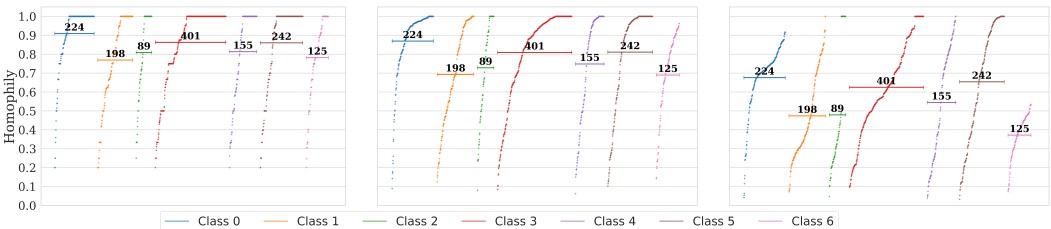

Figure 7: Node Homophily Distribution Scores for Cora.

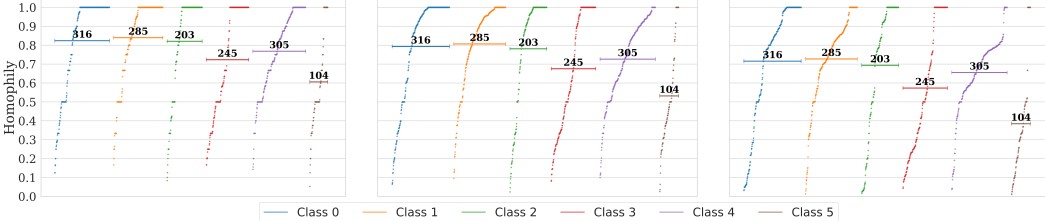

Figure 8: Node Homophily Distribution Scores for Citeseer.

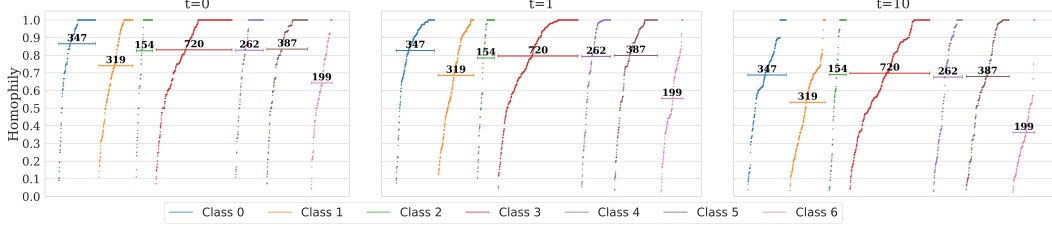

Figure 9: Node Homophily Distribution Scores for CORA-CA.

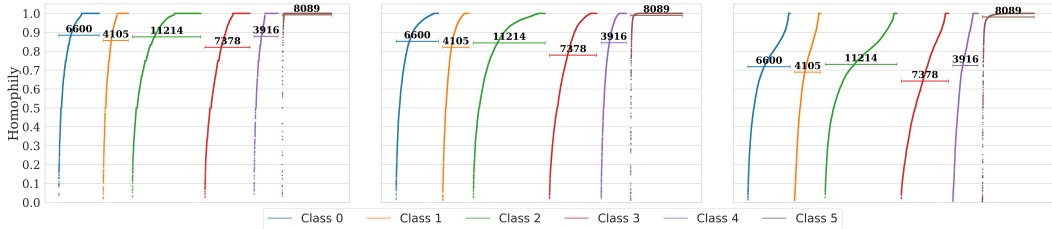

Figure 10: Node Homophily Distribution Scores for DBLP-CA.

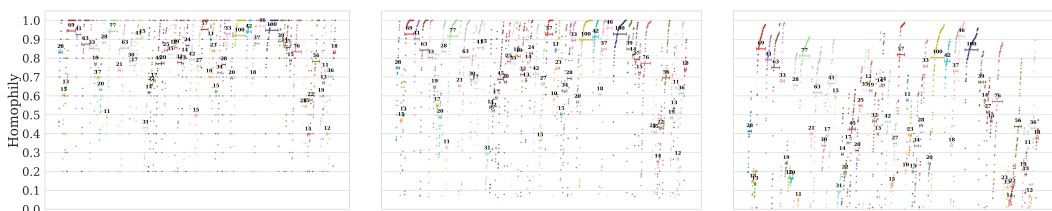

Figure 11: Node Homophily Distribution Scores for NTU2012.

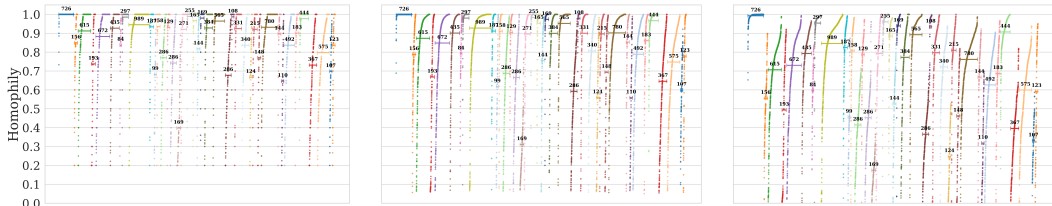

Figure 12: Node Homophily Distribution Scores for ModelNet40.

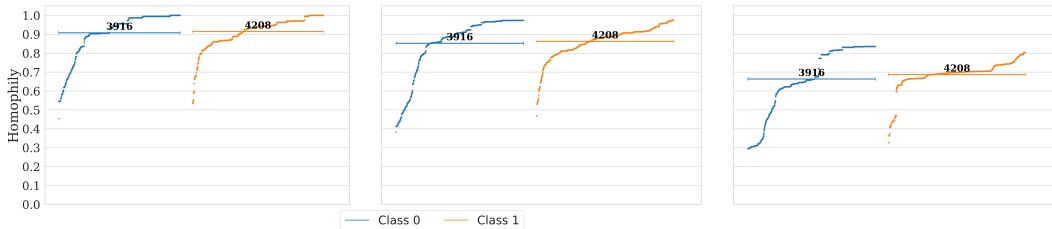

Figure 13: Node Homophily Distribution Scores for Mushroom.

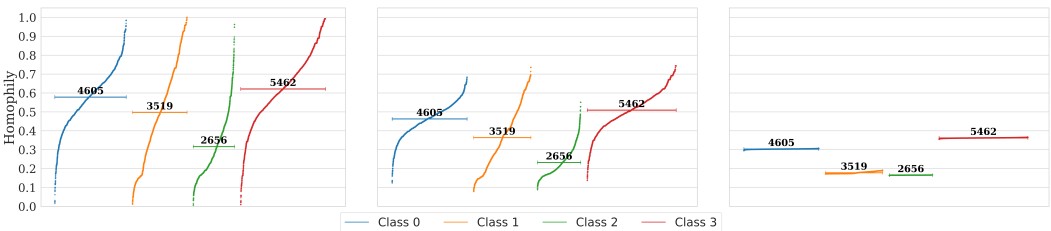

Figure 14: Node Homophily Distribution Scores for 20NewsW100.

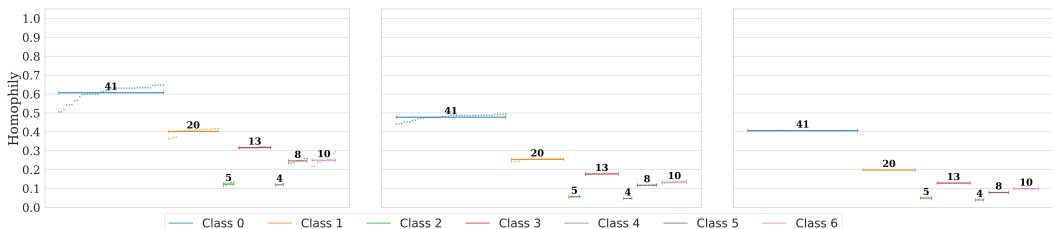

Figure 15: Node Homophily Distribution Scores for ZOO.

# M  COMPARISONS WITH OTHERS HOMOPHILY MEASURE IN LITERATURE

### K-UNIFORM HOMOPHILY MEASURE

**Hyperedge homophily**  Veldt et al. (2023) defines the group homophily measure for $k$-uniform hypergraphs as $G_k = (\mathcal{V}, \mathcal{E})$. The type $t$-affinity score for each $t \in \{1, \ldots, k\}$, indicates the likelihood of a node belonging to class $c$ participating in groups in which exactly $t$ group members also belong to class $c$ and defined as in equation 53. $d_t(v)$ is the number of hyperedges that $v$ belongs to with exactly $t$ members from class $c$. The authors also consider a standard baseline score $b_t(c)$, equation 54, that measures the probability that a class-$c$ node is in the group where $t$ members are from class $c$, given that the other $k - 1$ nodes were chosen uniformly at random.

$$\eta_t(c) = \frac{\sum_{v:y_v=c} d_t(v)}{\sum_{v:y_v=c} d_v}, \qquad (53) \qquad b_t(c) = \frac{\binom{n_c-1}{t-1}\binom{n-n_c}{k-t}}{\binom{n-1}{k-1}} \qquad (54)$$

$n_c$ is the number of nodes in class $c$ and $n$ is the total number of nodes in the hypergraph. The $k$-uniform hypergraph homophily measure can be expressed as a ratio of affinity and baseline scores, with a ratio value of 1 indicating that the group is formed uniformly at random, while any other number indicates that group interactions are either overexpressed or underexpressed for class $c$.

They suggest three possible ways for extending the concept of homophily to the hypergraph context. The first one is the *simple* homophily and it means that $\eta_t(c) > b_t(c)$ for $t = k$ and check whether a class has a higher-than-baseline affinity for group interactions that only involve members of their class. The second one is *order-j majority* homophily that is obtained when the top $j$ affinity scores for one class are higher than the baseline, i.e. $\eta_{k-j+1}(c) > b_{k-j+1}(c), \ldots, \eta_k(c) > b_k(c)$. The last one they consider is *order-j monotonic* homophily, which corresponds to the case when top $j$ ratio scores are increasing monotonic, i.e. $\eta_k(c)/b_k(c) > \eta_{k-1}(c)/b_{k-1}(c) > \cdots > \eta_{k-j+1}(c) > b_{k-j+1}(c)$.

Finally, considering that the value $\eta_t(c) - b_t(c)$ is the *bias* of class $c$ for type $t$, they introduce a type-$t$ *normalized bias* score that normalizes the maximum possible bias, hence the obtained metric is bounded in $[0, 1]$ and it is computed as:

$$
f_t(c) = \begin{cases} \frac{\eta_t(c) - b_t(c)}{1 - b_t(c)} & \text{if } \eta_t(c) \geq b_t(c) \\ \frac{\eta_t(c) - b_t(c)}{b_t(c)} & \text{if } \eta_t(c) < b_t(c) \end{cases} \tag{55}
$$

**Comparisons to our measure** Unlike Veldt et al. (2023) our measure of homophily does not assume a k-uniform hypergraph structure and can be defined for any hypergraph. Furthermore, the proposed measure enables the definition of a score for each node and hyperedge for any neighborhood resolution, i.e., the connectivity of the hypergraph can be explicitly investigated. It gives a definitaion of homophily that puts more emphasis on the connections following the two-step message passing mechanism starting from the hyperedges of the hypergraph.

### M.1 Figure - $k$-uniform homophily

Some of the hypergraphs have a lot of different size for hyperedges, here, we report the plots only for some $k$ for each dataset for brevity. All additional plots can be found in the supplementary material's zip files.

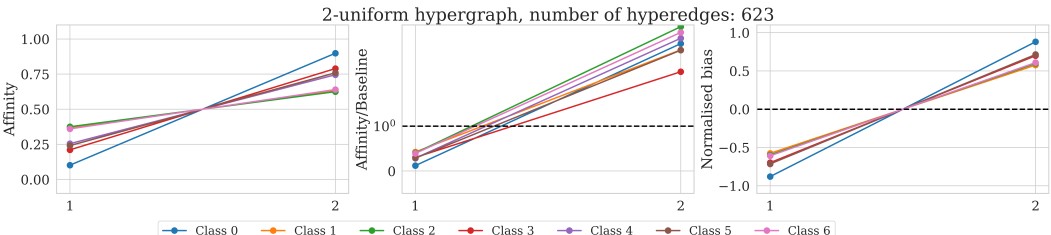

Figure 16: $k$-uniform homophily Cora.

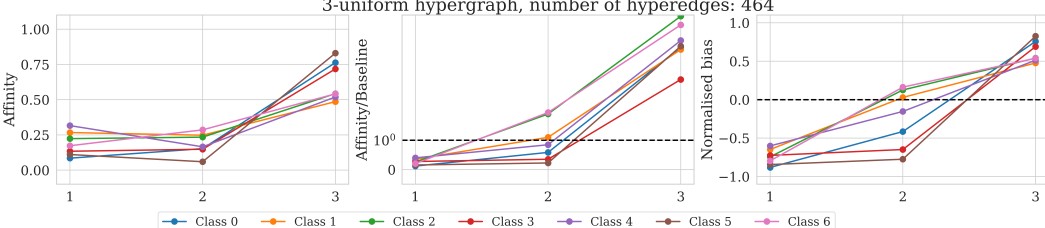

Figure 17: $k$-uniform homophily Cora.

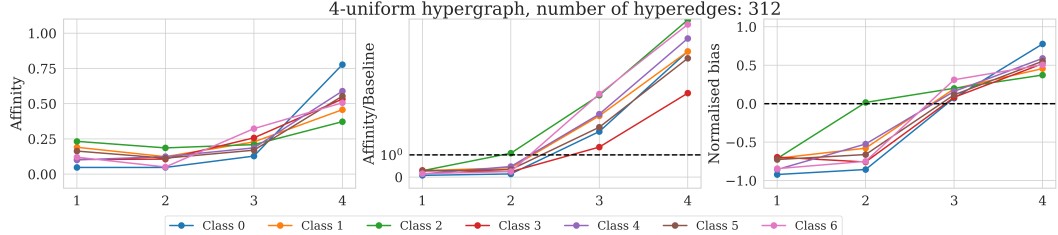

Figure 18: $k$-uniform homophily Cora.

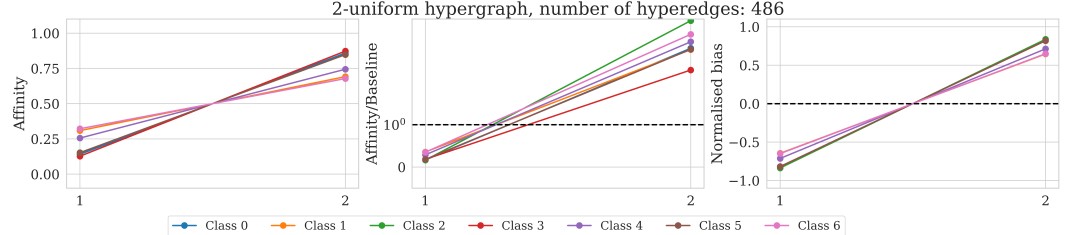

Figure 19: $k$-uniform homophily CORA-CA.

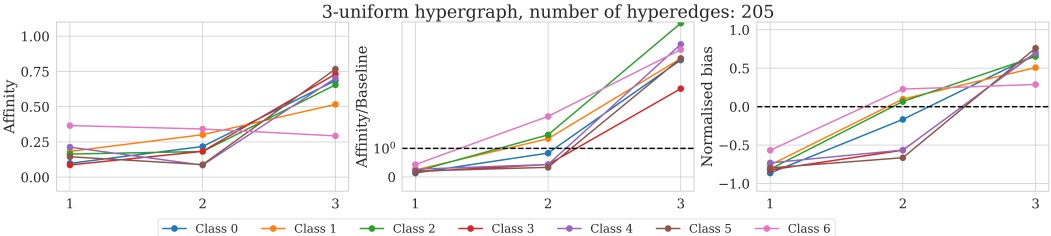

Figure 20: $k$-uniform homophily CORA-CA.

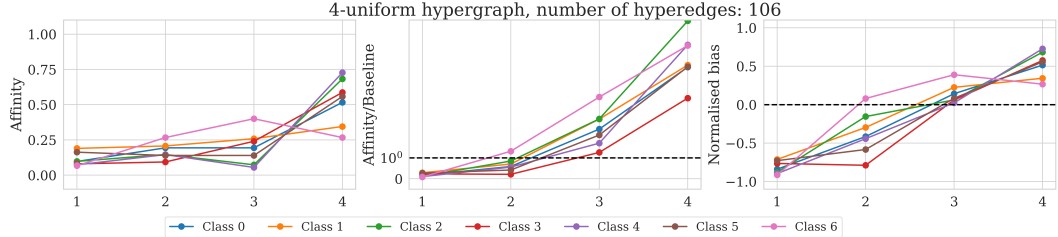

Figure 21: $k$-uniform homophily CORA-CA.

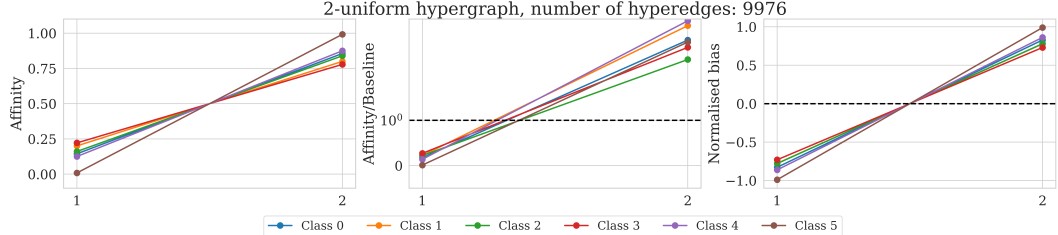

Figure 22: $k$-uniform homophily DBLP-CA.

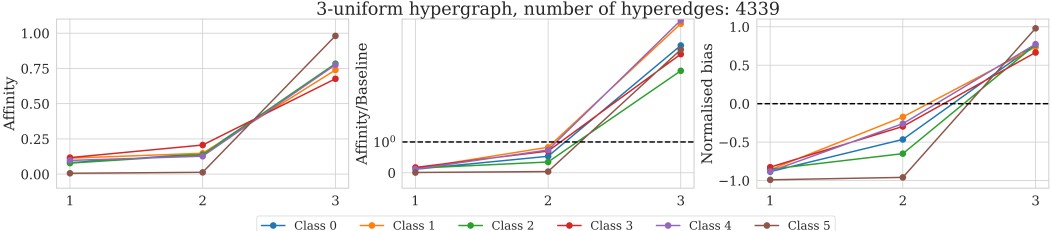

Figure 23: $k$-uniform homophily DBLP-CA.

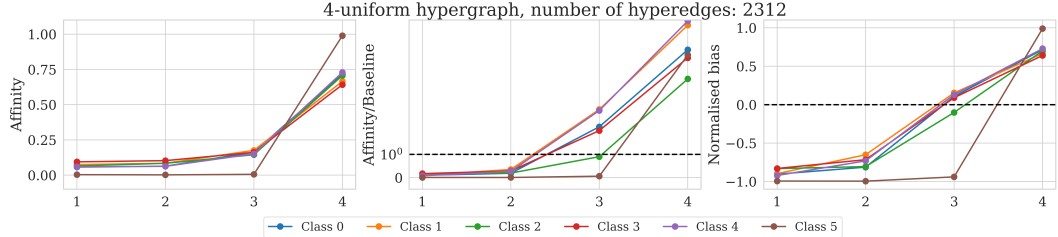

Figure 24: $k$-uniform homophily DBLP-CA.

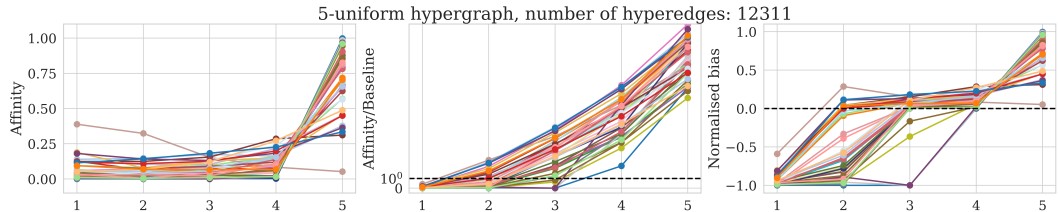

Figure 25: $k$-uniform homophily ModelNet40.

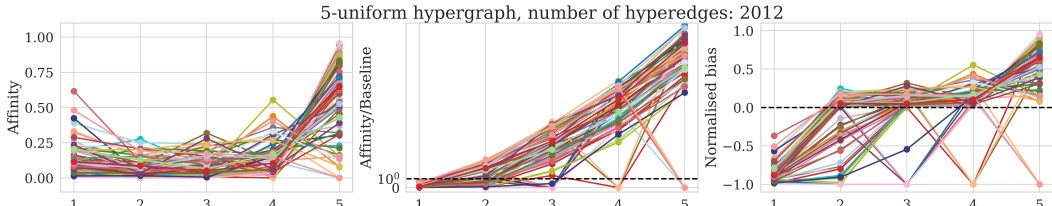

Figure 26: $k$-uniform homophily NTU2012.

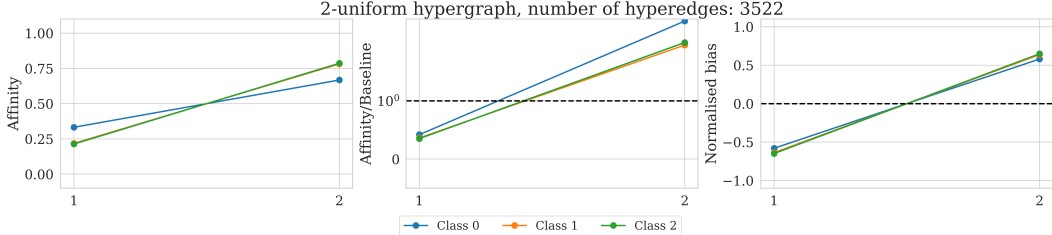

Figure 27: $k$-uniform homophily Pubmed.

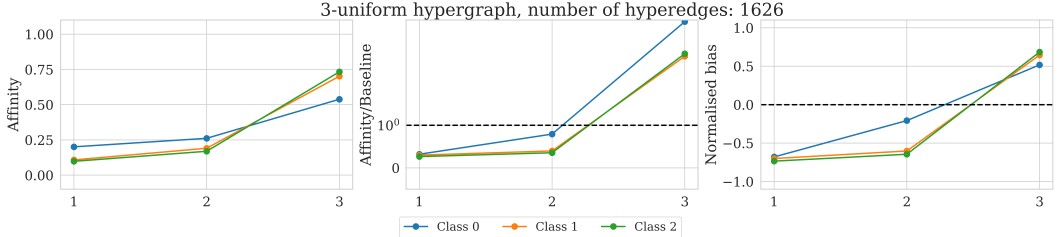

Figure 28: $k$-uniform homophily Pubmed.

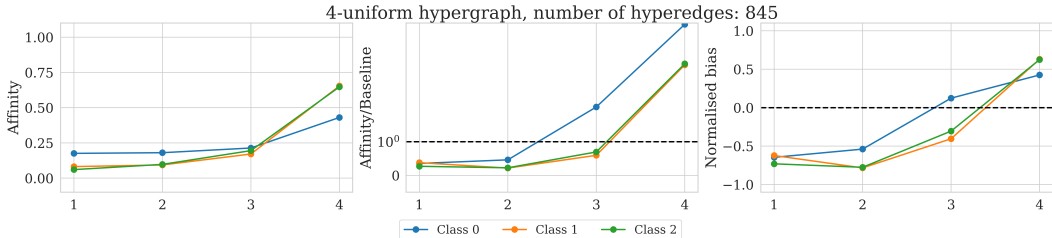

Figure 29: $k$-uniform homophily Pubmed.

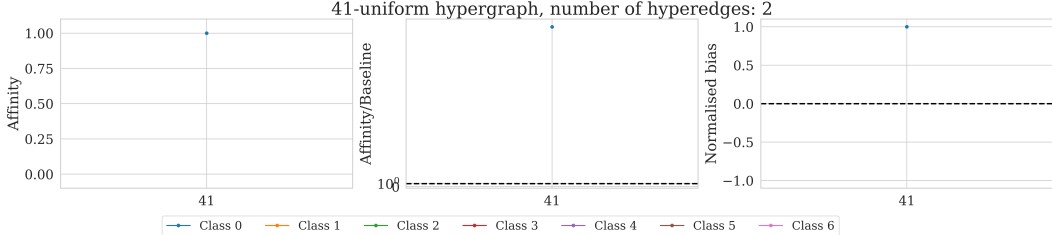

Figure 30: $k$-uniform homophily ZOO.

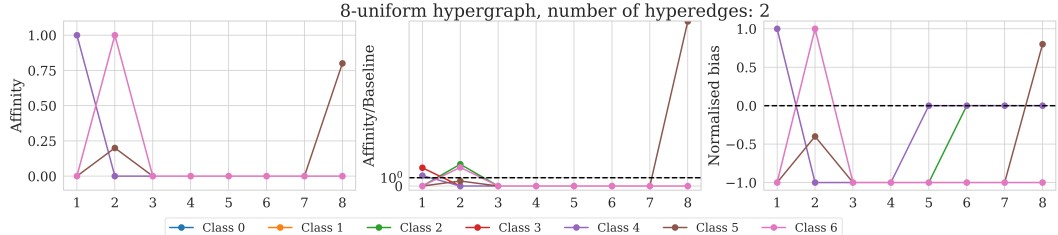

Figure 31: $k$-uniform homophily ZOO.

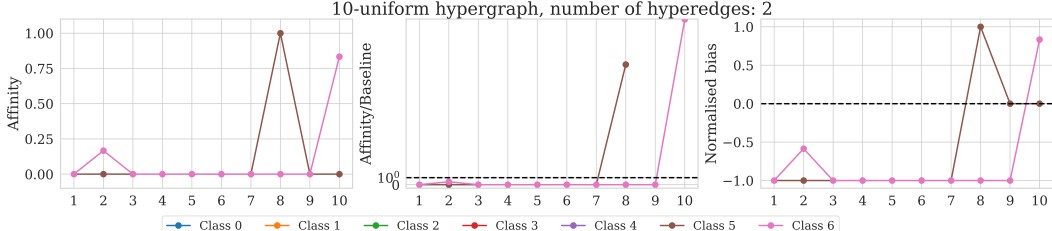

Figure 32: $k$-uniform homophily ZOO.

### SOCIAL EQUIVALENCE

The social equivalence Sun et al. (2023) is calculated through an expectation taken over pairs of users sampled from probability distributions. Specifically, $E(u, v \sim P(E_p))$ represents the expectation over positive user pairs sampled from the distribution of the hyperedges of the hypergraph $P(E_p)$. In contrast, $E(u_0, v_0 \sim P(V \times V \setminus E_p))$ represents the expectation over negative user pairs sampled from $P(V \times V \setminus E_p)$. The measure is a fraction between the numerator that involves the expected Jaccard index of environments for positive user pairs and the denominator, which comprises a similar calculation for negative user pairs. The Jaccard index is used to measure the similarity between two sets, and in this context, it assesses the similarity of hyperedges associated with positive user pairs. If the expected Jaccard index for positive user pairs is higher than that for negative pairs, the measure exceeds 1, indicating a significant level of observed social equivalence among users. **Comparisons to our measure** The concept of Social Equivalence, as introduced by Sun et al. (2023), differs significantly from our definition of homophily. In our approach, the measure is initially defined at both the node and hyperedge levels. Our primary objective is to address the question: 'How similar a node is to its neighbors?' Following a message passing scheme, our definition allows us to examine different time points, attempting to answer how similar a node is to the nodes it can reach with $t$ steps of message passing. This consideration also extends to edges, where we seek to understand the coherence or uniformity of an edge within itself. The guiding notion of similarity in our work is belonging to the same class. Specifically, in level-0 homophily, we evaluate whether a node is more connected with nodes of the same classes. We can then aggregate the node-level/hyperedge-level homophily to provide a definition for hypergraph homophily. In contrast, in the work of Sun et al.

(2023), the concept of social equivalence yields a single result for the entire hypergraph. The approach involves comparing the set of similar nodes that are connected with those that are not connected. The key question is whether the set of non-connected nodes is, on average, more similar or if the set of connected pairs exhibits greater similarity. It's important to note that this definition makes sense only for the entire hypergraph and captures a different notion of similarity.

### SOCIAL CONFORMITY

Social conformity, as described in Sun et al. (2023), involves leveraging learned representations of users and hyperedges within a model to understand and quantify the level of conformity among users in a social network.

### NODE HOMOPHILY COMPUTED ON THE CLIQUE EXPANSION OF THE HYPERGRAPH

Clique-expanded (CE) homophily, employed in Wang et al. (2023), is determined by calculating node homophily Pei et al. (2020) on the graph derived from the clique-expanded hypergraphs.

**Comparisons to our measure** The node homophily for a graph computes the fraction of neighbors with the same class for all nodes and then averages these values across the nodes. In contrast to our metric, CE homophily is not defined directly on the hypergraph but necessitates an expanded clique representation. While sharing some similarities with our node-wise measure $h_0$, it lacks the dynamic aspect inherent in the MP homophily measure, consequently failing to capture the dynamic information within connections. Our analysis in Appendix A underscores the significance of this dynamic element in understanding the correlation between homophily measures and the observed patterns in Hypergraph Neural Networks (HGNNs).

### FURTHER CONSIDERATION ON THE CONCEPTS OF SIMULATED SOCIAL ENVIRONMENT EVOLVING AND GROUP ENTROPY FROM SUN ET AL. (2023)

Further exploration of the concepts of simulated social environment evolving and group entropy is presented by Sun et al. (Sun et al., 2023). In their study, a dynamic analysis of specific hypergraph characteristics is conducted through message passing. They specifically focus on the evolving proportion of 'significant nodes' within the hyperedge relative to the original nodes across different epochs. These 'significant nodes' are identified as those with a probability of belonging to the hyperedge greater than 0.5, initially determined by multiplying the representation of the node with that of the hyperedge (averaged over its constituent nodes).

A noteworthy distinction from our methodology lies in their reliance on representations provided by a model, in contrast to our representation-independent approach. Despite the shared use of message passing in both approaches, we underscore these methodological differences.

It's important to highlight that the concept of group entropy introduced by Sun et al. is also noteworthy, representing an evolving model concept; however, its computation requires node representations provided by a model.

We posit that our measure and the metrics employed in Sun et al. (2023)'s paper can complement each other effectively.

## N    CLASS DISTRIBUTION SHIFT

We now report the results for the class distribution shift obtained by applying the mini-batch sampling procedure described in Section E. For each dataset, we choose to illustrate 3 different distributions: the one corresponding to the original labels ("Node"), the one obtained by applying both *Step 1* and *Step 2* described in the mini-batch paragraph of Section 4.2 and the one obtained by only applying *Step 2*.

### N.1    FIGURE - NODE HOMOPHILY

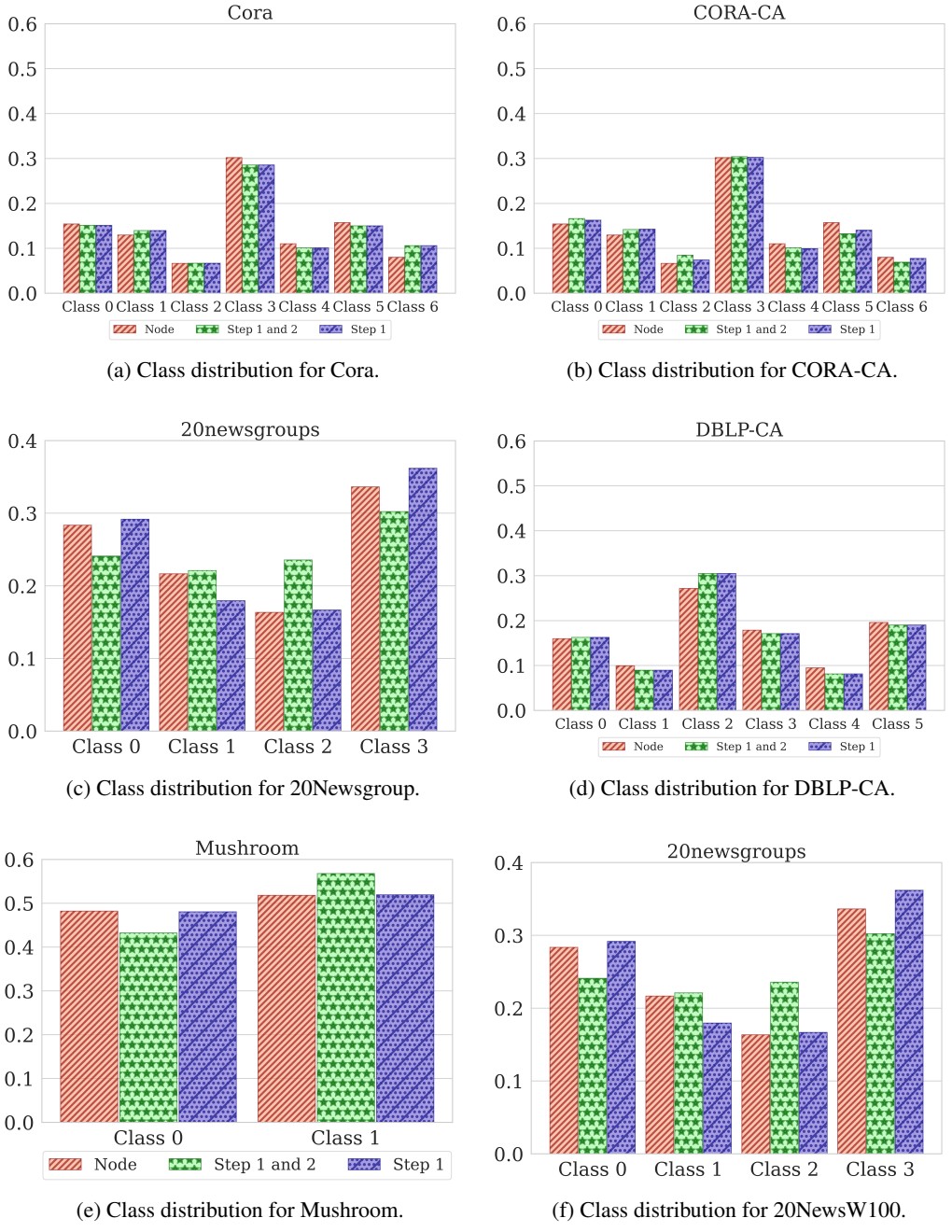

Figure 33: Class distribution shifts.

