# OpenReview forum: "Hypergraph Neural Networks through the Lens of Message Passing: A Common Perspective to Homophily and Architecture Design"
_ICLR.cc/2024/Conference — Submitted to ICLR 2024_

### Official Review · Reviewer_wbgf · 2023-10-18

**Soundness:** 3 good
**Presentation:** 3 good
**Contribution:** 2 fair
**Rating:** 6
**Confidence:** 4

**Summary:**

The paper offers an approach to understanding homophily by examining node class distribution in hypergraph datasets, accompanied by the introduction of MultiSetMixer, a dedicated hypergraph neural network crafted for efficient learning on hypergraphs.

Message passing is the central element for grasping homophily and enhancing current hypergraph neural networks with the proposed MultiSetMixer model.

Experiments
* demonstrate that the MultiSetMixer model performs favourably,
* uncover insights about connectivity patterns in benchmark datasets, emphasising the influence of large hyperedge cardinalities on performance, and
*  identify common failure modes related to distribution shifts and message-passing mechanisms.

**Strengths:**

### Clarity
1. Through clear explanations and qualitative analysis through Figure 1, the paper elucidates homophily in real-world hypergraphs.
2. The design of MultiSetMixer's architecture is clearly explained, and Propositions 1, 2, and 3 enhance the clarity of the presentation.
3. The paper exhibits a well-organised structure, complemented by a comprehensive supplementary material that thoroughly covers all aspects of the research content.


$~$
### Originality
4. The paper introduces an original homophily concept based on message passing, breaking away from an existing assumption of a k-uniform hypergraph structure.
5. MultiSetMixer extends the existing hypergraph neural network models (such as AllSet and UniGCN) by offering a broader scope and enabling hyperedge-dependent node representations.

**Weaknesses:**

### Quality
1. To enhance quality, the paper could provide a conceptual framework to describe how the proposed homophily measure integrates with MultiSetMixer to strengthen their mutual reliance on message passing.
2. To bolster the paper's quality, exploring datasets like Walmart and Congress, characterised by lower homophily patterns as shown in prior studies [Wang et al., 2023], could broaden the analysis beyond the datasets examined in this work.

$~$
### Significance
3. The mini-batching sampling method is not compelling when the entire input hypergraph can be stored in memory, particularly given the moderate sizes of all datasets as indicated in Table 6.
4. The importance of hyperedge-dependent node representations could be reinforced with more compelling experiments, such as exploring hyperedge-dependent node classes [Choe et al., 2023].

\
References
* [Wang et al., 2023]: Equivariant Hypergraph Diffusion Neural Operators, ICLR'23.
* [Choe et al., 2023]: Classification of Edge-dependent Labels of Nodes in Hypergraphs, KDD'23

**Questions:**

1. Were there specific examples or illustrations that clarify the conceptual framework linking the homophily measure and MultiSetMixer through message passing, ensuring a deeper understanding of their interrelation?
2. Related to the previous question, is it the case that MultiSetMixer can (provably) handle datasets with a high degree of heterophily better than existing models such as AllSet and UniGCN(II)?
3. What challenges or limitations were encountered in exploring datasets like Walmart and Congress, known for their lower homophily, and how could these challenges be addressed to enhance the scope of the analysis in this work?
4. Were there any insights into the rationale behind using mini-batching sampling, especially for datasets that can be entirely stored in memory?

---

> ### Author Response · Authors · 2023-11-22
> **Response to Reviewer wbgf (1/2)**
>
> Thank you for your detailed review and the recognition of the strengths of our work. We appreciate the constructive comments you provided, as it will undeniably help us improve it. We would like to address the concerns and questions you have raised:
>
>
>
> ### **Weaknesses**
>
> **W1.** Please refer to **Section 5.2** (Can homophily help us Understand our experemental results) and detailed discussion in **Supplementary Material A** (Interplay of Message Passing Homophily and Models Performances) where we address this concern. We illustrate the application of the proposed message-passing homophily measure. These sections emphasize its utility in capturing the dynamic nature of homophily and demonstrate its effectiveness in assessing hypergraph models' performance in comparison to the classical clique-expanded homophily measure [Wang et al., 2023] across diverse datasets.
>
> **W2.** Originally, we decided not to consider these datasets because, in contrast to our selected dataset benchmark, they do not include node features. This implies that node attributes need to be artificially generated before hypergraph models can be applied, and we were concerned about the implications of assessing the results under these circumstances.
>
> However, we agree that broadening the set of homophilic datasets could improve the overall quality of our paper, and can potentially reveal valuable insights. Therefore, following your suggestion, we have expanded our evaluation over Senate, Congress, House and Walmart datasets characterized in [Wang et al., 2023]. The results are added in Table 16 to the revised manuscript in Section J in the Supplementary Materials, and they can also be seen in the following Table:
>
> |            | MultiSet          | AllDeepSets      | AllSetTransformer |
> |------------|-------------------|------------------|-------------------|
> | Senate     | 61.34 ± 3.45      | 48.17 ± 5.67     | 51.83 ± 5.22      |
> | Congress   | 92.13 ± 1.30      | 91.80 ± 1.53     | 92.16 ± 1.05      |
> | House      | 70.77 ± 2.03      | 67.82 ± 2.40     | 69.33 ± 2.20      |
> | Walmart    | 64.23 ± 0.41      | 64.55 ± 0.33     | 65.46 ± 0.25      |
>
>
> Looking at the results, we observe that our MultiSetMixer model outperforms AllSetTransformer and AllDeepSets on the Senate and House datasets, while being on par within a standard deviation in the other two. Overall, we observe that our MultiSetMixer architecture shows good performance over low-homophilic datasets.
>
> **W3.** Indeed, this is an interesting aspect that needed further clarification in the paper. As suggested by the reviewer, the hyperedge mini-batching is particularly useful for very large datasets; whenever the network fits into memory, this step can be skipped. Regarding the node mini-batching within a hyperedge, we empirically found that, due to the involved distribution shifts, it can be useful to keep it even when the full hyperedge can be stored in memory. The revised manuscript includes these reflections in Supplementary Materials Section K.
>
> **W4.** Thank you for the suggestion, we agree that this could be an insightful line of research to further exploit hyperedge-based node representations. Unfortunately, this might fall out of the scope of this paper; we believe that the consideration of hyperedge-dependent node/edge classes represent a substantial shift in our evaluation (from the definition of datasets to its rigorous assessment) and perhaps could represent a contribution by itself in a future extension of this work. In this regard, we have added to our revised paper this interesting possibility as a future work. Please refer to **Supplementary Materials C** (Extended Conclusion and Discussion)
>
> ### **Questions**
>
> **Q1.** Thank you for the suggestion. In response, we have incorporated **Section A** (Interplay of Message Passing Homophily and Models' Performances) into the supplementary material. This section underscores the significance of formulating a dynamic homophily measure and illustrates its effectiveness in assessing the performance of hypergraph models.
>
> **Q2.** It is an interesting question. We provide an in-depth response in Supplementary Materials A (Interplay of Message Passing Homophily and Models' Performances) and Supplementary Materials B (Comparison and Analysis between MultiSet and AllSet Framework Performances). These sections comprehensively explore the performance of the proposed MultiSetMixer and the other message-passing models across diverse datasets characterized by varying levels of homophily.
>
> **Q3.** We kindly refer to the previous response of the Weaknesses 2, as we already described there the challenges and insights observed when tackling the new set of low-homophilic datasets (Wallmart, Congress, Senate and House). Thank you again for the suggestion.

---

> ### Author Response · Authors · 2023-11-22
> **Response to Reviewer wbgf (2/2)**
>
> **Q4.** As previously commented in the response to Weaknesses 3, we skip hyperedge mini-batching when the dataset fits into memory. However, the adoption of node mini-batching (within hyperedges) in the MultiSet framework is driven by a fundamental shift in data processing. Specifically, the generation of multiple node representations corresponding to the hyperedges they belong to benefits from this approach. Given that the number of potential connections grows exponentially with the number of nodes, the development and application of novel mini-batching schemas become a relevant tool to scale hypergraph models. Apart from adding this rationale in our revised manuscript, we additionally updated Table 6 of the paper to represent the number of hyperedge-dependent node representations and visualize our argument.
>
>
> |                                       | Cora | Citeseer | Pubmed | CORA-CA) | DBLP-CA | Mushroom | NTU2012 | ModelNet40 | 20Newsgroups | ZOO  |
> |---------------------------------------|------|----------|--------|---------|---------|----------|---------|------------|--------------|------|
> | Hyperedge Dependent Node Representations | 6060 | 5307     | 50506  | 4905    | 99561   | 40620    | 10060   | 61555      | 65451        | 1717 |
>
>
>
> ### **References**
>
> [Wang et al., 2023]: Equivariant Hypergraph Diffusion Neural Operators, ICLR'23.

---

### Official Review · Reviewer_gAXJ · 2023-10-31

**Soundness:** 2 fair
**Presentation:** 2 fair
**Contribution:** 2 fair
**Rating:** 5
**Confidence:** 4

**Summary:**

This paper points out that most current hypergraph learning methods and benchmarking datasets have been adapted from the graph domain, leading to the neglect of the hypergraph network foundation. The paper defines a new concept of homophily in hypergraphs using message passing and introduces a novel framework called MultiSet. Furthermore, it suggests a mini-batching method for hypergraph networks and conducts several experiments related to this approach. Finally, the paper conducts experiments involving changes of connectivity and provides a comprehensive analysis of various hypergraph models.

**Strengths:**

1. The paper points out the limitations of previous hypergraph research, emphasizing the need for new homophily and architecture in HGNN.
2. This paper analyzed hypergraph models through various experiments.

**Weaknesses:**

1. The author proposed homophily that can be applied to non-uniform hypergraphs and provided an analysis by demonstrating the homophily distribution changes with levels in Cora-CA and 20Newsgroups datasets. However, there is a lack of explanation regarding how the changes in homophily with levels are related to the performance of HGNN models or the characteristics of those datasets.
2. The novelty of the MultiSet framework proposed in the paper is limited. This is because previous research, such as EDHNN2 [1] and HNN [2], has already explored the use of different representations for each hyperedge.
3. The MultiSetMixer shows only marginal improvements in performance compared to existing baseline models and, in some cases, even exhibits lower performance. However, the paper lacks an analysis of the reasons behind these performance variations. It would be beneficial to have a comparison between the performance of MultiSet and AllSet along with explanations.
4. The main focus and key contributions of the paper are not entirely clear. While the motivation section highlights the shortcomings of existing hypergraph research and emphasizes the need for new homophily measures, architecture, and benchmark datasets, the actual experiments seem to place more emphasis on the mini-batching method than on the newly proposed homophily or the MultiSet architecture.


[1] Equivariant Hypergraph Diffusion Neural Operators, ICLR 23

[2] A Hypergraph Neural Network Framework for Learning Hyperedge-Dependent Node Embeddings, arxiv 22

**Questions:**

1. In the Supplementary Materials (Page 19, MMLP CB explanation), could you please elaborate on the meaning of "connectivity level"? I'm having some difficulty understanding the concept of "top-3 connectivity level.“
2. In step 2 of Mini-batching, it is mentioned that hyperedges are padded using a special padding token. What is this special padding token? Additionally, it is written that the padding condition is |e| > L, but shouldn't it be |e| < L?

---

> ### Author Response · Authors · 2023-11-22
> **Response to Reviewer gAXJ**
>
> We appreciate your detailed feedback and for raising relevant aspects in your review to improve the quality of the paper; we thoroughly addressed your concerns in our revised manuscript. Please, let us detail how exactly we confronted your comments as follows:
>
> ### **Weaknesses**
>
> **W1** Please refer to **Section 5.2** (Can Homophily Help us Understand our Experemental Results) and detailed discussion in **Supplementary Material A** (Interplay of Message Passing Homophily and Models Performances) where we address this concern. We illustrate the application of the proposed message-passing homophily measure. These sections emphasize its utility in capturing the dynamic nature of homophily and demonstrate its effectiveness in assessing hypergraph models' performance in comparison to the classical clique-expanded homophily measure [Wang et al., 2023] across diverse datasets.
>
> **W2** First of all, apologies for not including these two related works in our manuscript. We have updated Sections D in the supplementary materials (Extended Related Works on Hypergraph Neural Networks) to include a brief description of them as well as to explain their main differences with respect to our model. In particular, let us summarize them as follows:
>
> EDHNN [Wang et al., 2023] does not generate independent, hyperedge-based representations of nodes. This method incorporates the option of hyperedge-dependent messages from hyperedges to nodes, but at each iteration of the Message Passing it aggregates all these messages to generate a unique node hidden representation. In contrast to this, our MultiSet framework allows to keep across the MP procedure as many representations of a node as the number of hyperedges it belongs to, only requiring a pooling/aggregation at the readout phase.
>
> HNN [Aponte et al., 2022] does allow multiple hyperedge-based representations across the MP, but we did not initially include it in our related work because the theoretical formulation of this unpublished paper is not clear and rigorous, and the evaluation is neither reproducible nor comparable to other hypergraph models; not only its code is not publicly available, but they only use AUC as performance metric, instead of the gold-standard accuracy used in the graph/hypergraph literature. Whereas we realize now we should have added a comment on this in the original manuscript, we argue that our MultiSet framework represents a step forward by rigorously formulating a simple but general MP framework for hypergraph modeling that is flexible enough to deal with hyperedge-based node representations and residual connections, demonstrating as well that it generalizes previous hypergraph and graph models.
>
> Overall, we reckon that MultiSet can be a positive and novel contribution to our community by defining a strong and rigorous theoretical formulation upon which new hypergraph models can be designed and implemented.
>
> **W3** Thank you for this suggestion. We agree that an extensive analysis of the reasons behind the performance variations between MultiSet and AllSet frameworks can improve the quality of the paper. Consequently, we have added a new section –Appendix B– dedicated to highlighting the performance differences observed across all datasets analyzed between the main implementations of both frameworks, MultiSetMixer and AllSet models.
>
> **W4** Thank you for pointing out the need to improve the presentation of the key contributions. To address it, we have updated the main body of the paper. In particular, Sections 1 (Introduction), 3 (Defining and Measuring Homophily in Hypergraphs), 4 (Methods), 5 (Experemental Results), and 6 (Discussion). In addition to this, and  following the reviewer suggestion about giving more relevance to the contributions of the MultiSet framework and the introduced homophily measure, we have updated Sections 1, 5 and 6 to elaborate on these aspects and added discussions in Supplementary Materials A and B.
>
> ### **Questions**
> **Q1** When we mention "top-3 connectivity level," we are referring to the three most frequent sizes of hyperedges in the hypergraph. We've incorporated the explanation within the model description (Supplementary Materials E). Specifically, we consider the sizes of all the hyperedges, order them according to descending frequency of appearance in the hypergraphgraph, and select the top-3 ones. Hence, we use three separate encoders for the corresponding hyperedges (those cardinalities that are in top-3). For the rest of the hyperedges we used a 4th separate encoder.
>
> **Q2** We apologize for the oversight; as you pointed out, the padding is performed when |e| < L. We have corrected this in the revised manuscript and specified that the special padding tokens consist in 0-valued vectors.
>
> ### **References**
> [Wang et al., 2023]: Equivariant Hypergraph Diffusion Neural Operators, ICLR'23.
> [Aponte et al., 2022]: A Hypergraph Neural Network Framework for Learning Hyperedge-Dependent Node Embeddings, arxiv 22

---

> ### Comment · Reviewer_gAXJ · 2023-11-23
>
> Thank you for addressing my concerns with your response. I read your updated paper in detail. Part of the concerns were resolved (raised score accordingly), however, there are still some remaining issues.
>
> (weakness 1) In usual heterophily related papers, experiments showing the change of accuracies w.r.t homophily measure are provided. Such experiments with usual settings are preferable(Refer to Figure 2 in [3]) Also, as you have shown the change of homophily score w.r.t the number of layers, it would be more persuasive if you compare how the performance of traditional hypergraph NN methods and homophily score change as layer increases.
>
> Also, even though the point of multiset framework is about generalizing allset( or UNIGCN2) with some theoretical analysis, novelty seems to be marginal. Moreover, the necessity of generalizing allset needs to be provided. (Example, why allset does not fit to some problems or data. And how multiset can solve those problems)
>
> Reference
> [3] : [NIPS 22] Revisiting Heterophily For Graph Neural Networks (https://openreview.net/pdf?id=NjeEfP7e3KZ)

---

> ### Author Response · Authors · 2023-11-23
>
> Thank you very much for providing feedback of our revised manuscript, and for your time given the short notice.
>
> **1.** It is a good point that in usual heterophily-related papers experiments show the change of accuracies w.r.t homophily measure. However, we argue that there is currently a big gap between the understanding of homophily in the graph and hypergraph domains.  Whereas in graph literature the concept is well-defined and has already been shown to strongly correlate with downstream task performance, the notion of homophily is still vague in higher-order networks, without even a practical definition.
>
>  As detailed in our paper, there have been multiple attempts to define higher-order homophily (e.g. CE  and k-uniform versions), but they currently have strong limitations: as we show in the revised paper, CE homophily doesn't capture how models for hypergraphs behave, and the  k-uniform homophily measure is not applicable to non-uniform hypergraphs, which highly restricts its applicability.
>
> Therefore, by showing that our message-passiing homophily can help explain HGNN model performances, we argue that our work represents a meaningful contribution to our community. While further exploration of connections, as depicted in Figure 2 of [3], could be beneficial, we believe it goes beyond the scope of our paper. However, we think our introduced homophily concept can be really helpful in addressing these goals in future work.
>
> **2.** Regarding the justification for introducing the novel framework, MultiSet framework extends the existing hypergraph neural network models (such as AllSet and UniGCN) by offering a broader scope and enabling hyperedge-dependent node representations. As has been highlighted by Reviewer wbgf, this is indeed a strength of our proposal, as it opens new research opportunities in this yet not so-well explored field; for instance, it will allow to explore datasets where node classes are hyperedge-dependent  [Choe et al., 2023] (please refer to Supplementary Materials C (Extended Conclusion and Discussion)). While all this is not yet tackled in our current work, we strongly believe it will pave the path for the future research, helping to design novel architectures and benchmarking datasets.
>
> ### **References**
> [3] : [NIPS 22] Revisiting Heterophily For Graph Neural Networks (https://openreview.net/pdf?id=NjeEfP7e3KZ)
>
> [Choe et al., 2023]: Classification of Edge-dependent Labels of Nodes in Hypergraphs, KDD'23

---

### Official Review · Reviewer_U2hQ · 2023-11-01

**Soundness:** 3 good
**Presentation:** 3 good
**Contribution:** 3 good
**Rating:** 5
**Confidence:** 4

**Summary:**

This paper proposed a Unified Homophily Conceptualization and MultiSet Framework for Higher-Order Networks. Their method unifies the analytical frameworks of data and network architectures, providing a coherent lens for exploring intricate higher-order network structures and dynamics. They also introduce MultiSet, a pioneering message-passing framework that reimagines Hypergraph Neural Networks (HGNNs) by enabling hyperedge-dependent node representations. Additionally, they unveil a new architecture, the MultiSetMixer, which capitalizes on an innovative hyperedge sampling strategy.

**Strengths:**

S1. they clearly present their experimental details
S2. their method makes sense and is interesting.

**Weaknesses:**

W1. The concept of homophily for hypergraphs is previously discussed. (e.g. X Sun, et al. "Self-supervised Hypergraph Representation Learning for Sociological Analysis". TKDE.) It is recommended to include these works in your paper and discuss the differences.

W2. homophily is not a gold standard criterion in the graph learning area. Some network data are not isomorphic. The limitations and applicable situations should be discussed.


W3. In traditional dyadic graph neural networks, there are two branches. The 1st one is GCN-based methods, which take node features as input and aggregate information by topological structure. The 2nd one is Transformer-based methods, which take node features+position encoding as input and use self-attention to update node embeddings. The differences between hypergraphs and dyadic graphs are their topological structures (e.g. dyadic graphs contain pair-wise relations and hypergraphs have higher-order hyperedges). This difference makes sense when we design a hypergraph neural network by the 1st branch. I wonder what the significant contribution of hypergraphs will be if we design a hypergraph neural network by the 2nd branch.

W4. the experimental performance seems not that surprising. From my own experience, the message-passing pattern: "node to hyperedge and then hyperedge to node", is usually not better than directly aggregating nodes without hyperedge representations. For example, $X=\Theta X W$ where $\Theta$ is the hypergraph Laplacian matrix  Your MULTISET extension seems also to confirm my guess: you use multiset aggregating and it is very similar to aggregating nodes from a more fine-grained view (you split hyperedge representation more carefully compared with traditional simple hyperedge aggregating). Then I wondered what would happen if we completely removed the hyperedge representation in the network updating. It seems the residual connections play a crucial role in your "node to hyperedge and then hyperedge to node" message-passing flow, I wonder what would happen if you completely remove this component and compare your hypergraph structure with directly aggregating nodes without hyperedge representation.


W5. I admit that the new extension is helpful for message-passing-based methods. However, as more generalized data, hypergraphs should be more general than dyadic graphs. The hypergraph model should be more general than traditional dyadic graphs and it may be promising to achieve more general AI on graph domains. From this view, transformer-based methods are usually larger than message-passing-based methods. I wonder how to design a large hypergraph model with heavy parameters (like GPT in NLP) to achieve the "ChatGPT moment" in the graph area. It seems that the 2nd branch is more promising than the 1st branch because the 2nd branch can easily generate many large models while the 1st branch suffers from shallow layers. I suggest reconsidering the contribution of this work from a bigger picture: I wonder what the meaning of their extension, or how helpful their extension can contribute to achieving this AGI vision.


W6. The experimental work should be more extensive. For example, "Can the concept of homophily play a crucial role in HGNNs, similar to its significance in graph-based research? " Following this question proposed by the authors, the experimental section should clearly present the applicable situation and whether the homophily indeed contributes to the final performance and how much.

"Given that current HGNNs are predominantly extensions of GNN architectures adapted to the hypergraph domain, are these extended methodologies suitable, or should we explore new strategies tailored specifically for handling hypergraph-based data? " I didn't see the answer to your question. "Are the existing hypergraph benchmarking datasets truly meaningful and representative enough to draw robust and valid conclusions?" I didn't see your response to this question.

W7.  Following W4. I think the challenges, target problems, and the motivation of your new extended multiset structure, are not well formulated, and not clearly discussed.

**Questions:**

see W1-7

I would like to see the rebuttal to the questions mentioned in the above section “Paper Weakness”. I’m afraid that I might have not sufficient time to see a very long rebuttal. A concise and clear one would be good.

The potential weakness won't prevent me from raising my final score. I just want to make clear whether my understanding is correct.

---

> ### Author Response · Authors · 2023-11-22
> **Response to Reviewer U2hQ (1/2)**
>
> Thank you for your detailed and constructive review. We truly appreciate the time and effort you invested in providing feedback to improve the quality of the manuscript. We would like to carefully address each of your comments and concerns:
>
> ### **Weaknesses**
>
> **W1.** Thanks for the reference, of which we were not aware during submission. We have included the reference in our discussion; in particular, the revised paper discusses the main differences with respect to our introduced homophily notion in Section M of the supplementary materials.
>
> The concepts of Social Equivalence and Social Conformity, as introduced by [Sun, et al., 2023], differs significantly from our definition of homophily. Following a message passing scheme, our definition allows us to examine different time points, attempting to answer how similar a node is (w.r.t. its class) to the nodes it can reach with $t$ steps of message passing. This consideration extends to hyperedges edges as well. Furthermore, the concepts of social environment evolving and social polarization, which instead can be studied dynamically, also differ from our message passing homophily. This distinction arises because these concepts rely on node representations provided by a model, while our message passing homophily remains independent of any model.
>
> **W2.** We respectfully disagree with the reviewer; there have been more than 100+ top-tier publications in the last years focusing on the impact of homophily in graph representation learning (40+ only in the ICLR conference), some of which provide meaningful insights in this regard. For instance, [Ma et al., 2022] explored, both theoretically and empirically, that GCN has the potential to achieve good performance if nodes with the same label share similar neighborhood patterns. Homophilous graphs always satisfy such assumptions, which explains why GCN typically works well for them.
>
> In the graph literature, the concept of homophily is already well studied, but we found that this is not true for hypergraphs. Our work aims to address this by introducing a “natural” way of defining homophily directly for hypergraphs. For example, the proposed homophily measure overcomes the unrealistic assumption of uniform hyperedges needed in the previous definition of [Veldt et al., 2023]. We argue that our homophily concept can become a practical tool for the community to perform insightful analysis of hypergraph models similar to the ones with GNNs. Please see **Supplementary Materials A** for the extended discussion on the interplay of the proposed homophily measure and model performances.
>
> **W3.** This is an interesting aspect. Indeed, in our main table (Table 1 in the paper), we presented the performance of a hypergraph transformer architecture executed over node features enhanced with positional encoding, which is aligned to the 2nd branch mentioned in the comment. Note that the positional encodings are derived from hypergraph connectivity using hyperedge vector centrality [Kovalenko et al., 2022]. On average, these results closely align with those of MLP, indicating that a straightforward use of transformer-based architecture is not immediately beneficial in hypergraph learning. Conversely, the implementation of AllSetTransformer employing the transformer architecture for enhanced message passing, demonstrates consistently strong performance. Note that self-attention is used within every hyperedge over the node features and among hyperedges without any position encodings. Overall, our results show that standard or hybrid message passing architectures are still state-of-the-art on hypergraph datasets, but this can change (potentially) with the exploration of larger and more diverse datasets, with a similar shift happening also in graph neural networks [Liu et al., 2022]. We believe our work, with a clear analysis of different datasets and the effect of mini-batching, is a significant step in that direction.
>
> **W4.** We concur with the reviewer's insight. The observation may be attributed to the homophily level diminishing between steps 0 and step 1 for the dataset under consideration (see Table 5 in the supplementary materials). This implies that executing two steps might yield inferior results compared to opting solely for node aggregation. Perhaps the only exception is when introducing an attention mechanism in the procedure, as shown by the fact that AllSetTransformer consistently outperforms its more simple AllDeepSets counterpart.
>
> We would also like to clarify that our MultiSetMixer implementation does not implement residual connections; looking, for instance, at the results of UniGNN [Huang et al., 2021], which does include them, they suggest that this component does not play a crucial role, as suggested by the reviewer.

---

> ### Author Response · Authors · 2023-11-22
> **Response to Reviewer U2hQ (2/2)**
>
> **(continuation)**
>
> **W5.** We strongly believe that the path towards an AGI solution in this area will rely on a deeper understanding of the domain (graph, hypergraph, topological, geometrical) where data comes from Transformer-based architectures can benefit from deep architectures, but we argue that, whenever relational interactions play a crucial role for the considered task, leveraging these natural inductive biases in the first place can be key to design more scalable solutions.
>
> In this work, we aim to shift the focus of our community towards the study of higher-order network properties and requirements, with the goal of designing concepts and models that can naturally accommodate them (instead of just lifting graph-based counterparts). Improving our understanding on how to analyze and design hypergraph architectures could potentially result in new pipelines that would allow more and larger layers, overcoming the oversmoothing and oversquashing limitations of current methodologies.
>
> Achieving AGI through a more scalable and interpretable framework that can naturally exploit the relational connections present in the data is a powerful research direction for the future, and (as discussed for W3) we believe our results are a step in that direction.
>
> **W6.** We reckon that by addressing the concerns raised in this comment, our work can significantly improve its overall quality; the revised paper has been updated to this end in the following way:
>
> - Added a detailed analysis on homophily vs. performance: See Appendix A, where we show that our  homophily measure correlates better with HGNN models' performance compared to classical homophily measures over the clique-expanded hypergraph. Our findings underscore the crucial role of correctly expressing homophily in HGNNs, emphasizing the potential of our proposed homophily score in capturing higher-order dynamics.
>
> - Updated Evaluation and Conclusions sections to relate the obtained results with our original questions. In particular, our results suggest that:
>     - There is room for improvement in designing hypergraph models that better accommodate higher-order relationships; our MultiSetMixer implementation outperforms other solutions in some datasets (e.g. 20newsgrops) leveraging the natural connectivity distribution shift as described in Section 5.2. Additionally, we demonstrate that our model performs consistently better on 3D object segmentation datasets.
>     - Our analysis on the most used dataset benchmark reveals the need of a new set of datasets if we aim at improving our understanding in modelling relational data in its diverse possibilities. For instance, looking at the homophily analysis we can observe the lack of diversity of datasets in some homophily levels (most of them are in the range of 80-90%). Moreover, in our connectivity-based experiments, we also find that some datasets do not have meaningful hyperedge connections (Zoo, Pubmed, Citeseer).
>
> **W7.** Thank you for your suggestion, we believe that the revised manuscript (Sections 1, Introduction, and Section 6, Discussion) better formulates our motivations and the contributions of our work, and further discusses the challenges addressed by our introduced MultiSet framework.
>
> ### **References**
> [Sun et al., 2023]: Self-supervised Hypergraph Representation Learning for Sociological Analysis. TKDE.2023
>
> [Ma et al., 2022] : Is Homophily a Necessity for Graph Neural Networks?, ICLR 2022
>
> [Veldt et al., 2023]: Combinatorial characterizations and impossibilities for higher-order homophily, Science Advances
>
> [Kovalenko et al., 2022]: Vector Centrality in Hypergraphs, Elsevier'22.
>
> [Liu et al., 2022] Graph Self-Supervised Learning: A Survey, TKDE’22
>
> [Huang et al., 2021]: UniGNN: a Unified Framework for Graph and Hypergraph Neural Networks, IJCAI-21

---

### Official Review · Reviewer_LrvL · 2023-11-01

**Soundness:** 2 fair
**Presentation:** 2 fair
**Contribution:** 2 fair
**Rating:** 5
**Confidence:** 2

**Summary:**

This work studies the hypergraph neural network (HGNN) and proposes several artifacts for developing the HGNN model on top of the homophily concept. In particular, the authors first introduce a novel definition of homophily based on the message-passing scheme. Afterward, they propose a new HGNN architecture, namely MultiSet, to take into account the hyperedge-dependent relations when representing the nodes in a hypergraph. Besides, the MultiSetMixer is devised to implement the layer learning process. In addition, a connectivity-based mini-batching strategy is proposed to handle large hypergraphs. Extensive experiments on benchmark datasets validate the proposed methods.

**Strengths:**

1. How to design an effective HGNN architecture plays a vital role in the graph learning community. The implications of this work could be significant for many downstream applications.
2. Extensive experiments are conducted to evaluate the proposed approach on many benchmark datasets and models.

**Weaknesses:**

1. The presentation and organization of this draft should be improved with many efforts to reach the standard of a top-tier conference like ICLR. For example, the contributions are not highlighted clearly in the introduction, and many background introductions can be found throughout this paper.

2. The technical challenges that motivate this study are unclear. In other words, the technical contributions of this work are limited since the proposed methods are mostly incremental.

**Questions:**

1. What are the evaluation metrics w.r.t. the reported experimental results in Tables 1-4?

---

> ### Author Response · Authors · 2023-11-22
> **Response to Reviewer LrvL**
>
> Thank you for your thorough review. We appreciate the constructive feedback you provided, as it will certainly help enhance our paper. We would like to address the concerns and questions you have raised:
>
> **Weaknesses**
> 1. Thank you for pointing out the need to improve the presentation and organization. To address your concern, we have partially rewritten the main body of the paper. In particular, we have updated Sections 1 (Introduction), 3 (Defining and Measuring Homophily in Hypergraphs), 4 (Methods), 5 (Experimental Results), and 6 (Discussion).
>
> 2. We appreciate your perspective, and following the previous response, we highlighted the challenges addressed by our proposed methods in Section 1. However, we argue that our main technical proposals—i.e., a novel definition of Message Passing homophily for hypergraphs, the MultiSet framework, and the MultiSetMixer implementation with mini-batching—represent valuable contributions to the community. So far, hypergraph modeling has focused on adapting existing graph-based concepts and methodologies by lifting them to higher-order domains. In contrast to this, the technical aspects introduced in our paper represent a shift towards the definition of concepts and methodologies directly over hypergraph-based structures, whose soundness does no longer rely on graph-related notions but rather on novel, efficient ways of analyzing and processing multi-set interactions. The revised manuscript further remarks and clarifies the common technical motivation of our proposals and the potential implications they might bring to our community.
>
> **Questions**
> 1. All datasets used across our evaluation involve a node classification task. Following previous studies and benchmarks (e.g., [Chien et al. 2022]), we consider the test accuracy as the performance metric, representing the percentage of nodes that were successfully classified by each model over the test split. In particular, Tables 1-4 show the mean and standard deviation of the test accuracy achieved by each model over 15 different training, validation, and test splits, considering at each of these splits 2 different model initializations. Therefore, each entry in these tables aggregates the information of 30 different runs of a particular model.
>
> ### **References**
>
> [Chien et al. 2022] You Are AllSet: A Multiset Learning Framework for Hypergraph Neural Networks. ICLR,2022

---

### Author Response · Authors · 2023-11-22
**General Response**

We thank the reviewers for the extensive evaluation of our paper. We are glad they have commented positively on our **extensive experimental evaluation** (all reviewers), on the **novelty and originality of the methods** (reviewers U2hQ, wbgf), and on the **implications for further research** (reviewer LrvL).

We have provided an individual point-by-point answer to each reviewer. We briefly discuss here some points that were raised by reviewers, highlighting the changes that were made to the paper. We strongly believe that the overall quality of the revised paper has been significantly enhanced. All the changes in the revised manuscript appear in blue color.  See a brief summary below:

**Organization and contributions**: the paper has several contributions, including a novel measure of homophily for hypergraphs (that we call *message-passing homophily*), a slight generalization of the AllSets framework, a novel architecture (MultiSetMixer), and an extensive experimental benchmarking. We update the paper to highlight 3 fundamental research questions we address (listed in the abstract as Q1, Q2, and Q3). We reference Q1-3 throughout the paper to clarify which point we are considering for each section / experiment. We have also better connected the different contributions by showing how our message-passing homophily metric correlates with the performance of the different methods (see in particular the novel Appendix A), detailing in turn which properties of the datasets favour a certain architecture (MLP, AllSet, MultiSetMixer).

**Novelty of the framework**: as described above, we have added a large set of experimental evaluations to clarify how the proposed message-passing homophily measure can be used to analyze and even predict the performance of different models. We have also included several papers suggested by the reviewers, clarifying when needed the difference with respect to our paper.

**Experimental results**: we stress that we are not looking for a single hypergraph NN that can outperform all possible models in every conceivable scenario. In fact, we believe our paper clarifies how this is an ill-defined problem since different networks with different biases (e.g., towards homophily) will perform differently on separate datasets. Via our analysis, we have shown that the proposed homophily measure is a powerful tool to analyze complex dynamics inside the hypergraph datasets, allowing us to clarify the strengths and weaknesses of each method in turn. We refer in particular to Sections 1 (Introduction), 5 (Experimental Results), 6 (Discussion), and Supplementary Material A (Interplay of Message Passing Homophily and Models Performances)
Sections 1 (Introduction), 3 (Defining and Measuring Homophily in Hypergraphs), 4 (Methods), 5 (Experimental Results), and 6 (Discussion).

---

### Author Response · Authors · 2023-11-22
**General Response**

We want to extend our sincere apologies for the delayed release of our responses. It was important to us to dedicate the necessary time and effort to thoroughly consider all reviewers and address every concern raised during the review process.

We want to reassure you that, in the remaining time, we are fully committed to clarifying any points that may require additional explanation.

Once again, we appreciate your patience and understanding.

---

### Meta-Review · Area_Chair_333h · 2023-12-18

**Metareview:**

This paper addresses hyper-graph neural networks. The expositions is a bit brusk, skipping over any articulation of what hypergraph neural networks are, what problems they address, and how. The paper explores a hypergraph analogue of the homophily property that has been of interest in the graph neural network literature. The authors introduce a method called MultiSetMixer and perform a number of experiments. Notably even in the GNN literature the discussion of homophily is somewhat fraught and the focus on homophily as an explainer of model performance appears to be somewhat dubious. The reviewers found weaknesses with the exposition, experiments, and believed that the motivation of the setup was not well explained. Unfortunately, the reviewers were not overall very responsive during the discussion period and most did not respond even with post-rebuttal chasing. Still, because the reviewers were somewhat absentee, I took the time to skim the paper, and I agree that the paper does not appear to be in shape for acceptance.

**Justification For Why Not Higher Score:**

Insufficient support from reviewers, unclear motivations and exposition.

**Justification For Why Not Lower Score:**

N/A.

---

### Decision · Program_Chairs · 2024-01-16

Reject